# FEDERATED LEARNING'S BLESSING: FEDAVG HAS LINEAR SPEEDUP

## ABSTRACT

Federated learning (FL) learns a model jointly from a set of participating devices without sharing each other's privately held data. The characteristics of non-*i.i.d.* data across the network, low device participation, high communication costs, and the mandate that data remain private bring challenges in understanding the convergence of FL algorithms, particularly in regards to how convergence scales with the number of participating devices. In this paper, we focus on Federated Averaging (FedAvg)–arguably the most popular and effective FL algorithm class in use today–and provide a unified and comprehensive study of its convergence rate. Although FedAvg has recently been studied by an emerging line of literature, it remains open as to how FedAvg's convergence scales with the number of participating devices in the fully heterogeneous FL setting–a crucial question whose answer would shed light on the performance of FedAvg in large FL systems. We fill this gap by providing a unified analysis that establishes convergence guarantees for FedAvg under three classes of problems: strongly convex smooth, convex smooth, and overparameterized strongly convex smooth problems. We show that FedAvg enjoys linear speedup in each case, although with different convergence rates and communication efficiencies. While there have been linear speedup results from distributed optimization that assumes full participation, ours are the first to establish linear speedup for FedAvg under both statistical and system heterogeneity. For strongly convex and convex problems, we also characterize the corresponding convergence rates for the Nesterov accelerated FedAvg algorithm, which are the first linear speedup guarantees for momentum variants of FedAvg in the convex setting. To provably accelerate FedAvg, we design a new momentum-based FL algorithm that further improves the convergence rate in overparameterized linear regression problems. Empirical studies of the algorithms in various settings have supported our theoretical results.

## 1 INTRODUCTION

Federated learning (FL) is a machine learning paradigm where many clients (e.g., mobile devices or organizations) collaboratively train a model under the orchestration of a central server (e.g., service provider), while keeping the training data decentralized (Smith et al. (2017); Kairouz et al. (2019)). In recent years, FL has swiftly emerged as an important learning paradigm (McMahan et al. (2017); Li et al. (2020a))–one that enjoys widespread success in applications such as personalized recommendation (Chen et al. (2018)), virtual assistant (Lam et al. (2019)), and keyboard prediction (Hard et al. (2018)), to name a few–for at least three reasons: First, the rapid proliferation of smart devices that are equipped with both computing power and data-capturing capabilities provided the infrastructure core for FL. Second, the rising awareness of privacy and the explosive growth of computational power in mobile devices have made it increasingly attractive to push the computation to the edge. Third, the empirical success of communication-efficient FL algorithms has enabled increasingly larger-scale parallel computing and learning with less communication overhead.

Despite its promise and broad applicability in our current era, the potential value FL delivers is coupled with the unique challenges it brings forth. In particular, when FL learns a single statistical model using data from across all the devices while keeping each individual device's data isolated (Kairouz et al. (2019)), it faces two challenges that are absent in centralized optimization and distributed (stochastic) optimization (Zhou & Cong (2018); Stich (2019); Khaled et al. (2019); Liang et al. (2019); Wang & Joshi (2018); Woodworth et al. (2018); Wang et al. (2019); Jiang & Agrawal (2018); Yu et al. (2019b;a); Khaled et al. (2020); Koloskova et al. (2020); Woodworth et al. (2020b;a)):

1) **Data (statistical) heterogeneity:** data distributions in devices are different (and data cannot be shared);

| Objective function / Participation | Strongly Convex | Convex | Overparameterized general case | Overparameterized linear regression |
|---|---|---|---|---|
| Full | $\mathcal{O}(\frac{1}{NT} + \frac{E^2}{T^2})$ | $\mathcal{O}\left(\frac{1}{\sqrt{NT}} + \frac{NE^2}{T}\right)$ | $\mathcal{O}(\exp(-\frac{NT}{E\kappa_1}))$ | $\mathcal{O}(\exp(-\frac{NT}{E\kappa}))^{\dagger}$ |
| Partial | $\mathcal{O}\left(\frac{E^2}{KT} + \frac{E^2}{T^2}\right)$ | $\mathcal{O}\left(\frac{E^2}{\sqrt{KT}} + \frac{KE^2}{T}\right)$ | $\mathcal{O}(\exp(-\frac{KT}{E\kappa_1}))$ | $\mathcal{O}(\exp(-\frac{KT}{E\kappa}))^{\dagger}$ |

Table 1: Our convergence results for FedAvg and accelerated FedAvg in this paper. Throughout the paper, $N$ is the total number of local devices, and $K \leq N$ is the maximal number of devices that are accessible to the central server. $T$ is the total number of stochastic updates performed by each local device, $E$ is the local steps between two consecutive server communications (and hence $T/E$ is the number of communications). $^{\dagger}$ In the linear regression setting, we have $\kappa = \kappa_1$ for FedAvg and $\kappa = \sqrt{\kappa_1 \tilde{\kappa}}$ for momentum accelerated FedAvg (FedMaSS), where $\kappa_1$ and $\sqrt{\kappa_1 \tilde{\kappa}}$ are condition numbers defined in Section G. Since $\kappa_1 \geq \tilde{\kappa}$, this implies a speedup factor of $\sqrt{\frac{\kappa_1}{\tilde{\kappa}}}$ for accelerated FedAvg.

2) **System heterogeneity:** only a subset of devices may access the central server at each time both because the communications bandwidth profiles vary across devices and because there is no central server that has control over when a device is active (the presence of "stragglers").

To address these challenges, Federated Averaging (FedAvg) McMahan et al. (2017) was proposed as a particularly effective heuristic, which has enjoyed great empirical success. This success has since motivated a growing line of research efforts into understanding its theoretical convergence guarantees in various settings. For instance, Haddadpour & Mahdavi (2019) analyzed FedAvg (for non-convex smooth problems satisfying PL conditions) under the assumption that each local device's minimizer is the same as the minimizer of the joint problem (if all devices' data is aggregated together), an overly restrictive assumption that restricts the extent of data heterogeneity. Very recently, Li et al. (2020b) furthered the progress and established an $\mathcal{O}(\frac{1}{T})$ convergence rate for FedAvg for strongly convex smooth Federated learning problems with both data and system heterogeneity. A similar result in the same setting Karimireddy et al. (2019) also established an $\mathcal{O}(\frac{1}{T})$ result that allows for a linear speedup when the number of participating devices is large. At the same time, Huo et al. (2020) studied the Nesterov accelerated FedAvg for non-convex smooth problems and established an $\mathcal{O}(\frac{1}{\sqrt{T}})$ convergence rate to stationary points.

However, despite these very recent fruitful pioneering efforts into understanding the theoretical convergence properties of FedAvg, it remains open as to how the number of devices–particularly the number of devices that participate in the computation–affects the convergence speed. In particular, is linear speedup of FedAvg a universal phenomenon across different settings and for any number of devices? What about when FedAvg is accelerated with momentum updates? Does the presence of both data and system heterogeneity in FL imply different communication complexities and require technical novelties over results in distributed and decentralized optimization? These aspects are currently unexplored or underexplored in FL. We fill in the gaps here by providing affirmative answers.

**Our Contributions** We provide a comprehensive and unified convergence analysis of FedAvg and its accelerated variants in the presence of both data and system heterogeneity. Our contributions are threefold.

First, we establish an $\mathcal{O}(1/KT)$ convergence rate under FedAvg for strongly convex and smooth problems and an $\mathcal{O}(1/\sqrt{KT})$ convergence rate for convex and smooth problems (where $K$ is the number of participating devices), thereby establishing that FedAvg enjoys the desirable linear speedup property in the FL setup. Prior to our work here, the best and the most related convergence analysis is given by Li et al. (2020b) and Karimireddy et al. (2019), which established an $\mathcal{O}(\frac{1}{T})$ convergence rate for strongly convex smooth problems under FedAvg. Our rate matches the same (and optimal) dependence on $T$, but also completes the picture by establishing the linear dependence on $K$, for any $K \leq N$, where $N$ is the total number of devices, whereas Li et al. (2020b) does not have linear speedup analysis, and Karimireddy et al. (2019) only allows linear speedup close to full participation ($K = \mathcal{O}(N)$). As for convex and smooth problems, there was no prior work that established the $\mathcal{O}(\frac{1}{\sqrt{T}})$ rate under both system and data heterogeneity. Our unified analysis highlights the common elements and distinctions between the strongly and convex settings.

Second, we establish the same convergence rates–$\mathcal{O}(1/KT)$ for strongly convex and smooth problems and $\mathcal{O}(1/\sqrt{KT})$ for convex and smooth problems–for Nesterov accelerated FedAvg. We analyze the accelerated version of FedAvg here because empirically it tends to perform better; yet, its theoretical

convergence guarantee is unknown. To the best of our knowledge, these are the first results that provide a linear speedup characterization of Nesterov accelerated FedAvg in those two problem classes (that FedAvg and Nesterov accelerated FedAvg share the same convergence rate is to be expected: this is the case even for centralized stochastic optimization). Prior to our results here, the most relevant results Yu et al. (2019a); Li et al. (2020a); Huo et al. (2020) only concern the non-convex setting, where convergence is measured with respect to stationary points (vanishing of gradient norms, rather than optimality gaps). Our unified analysis of Nesterov FedAvg also illustrates the technical similarities and distinctions compared to the original FedAvg algorithm, whereas prior works (in the non-convex setting) were scattered and used different notations.

Third, we study a subclass of strongly convex smooth problems where the objective is over-parameterized and establish a faster $\mathcal{O}(\exp(-\frac{KT}{\kappa}))$ convergence rate for FedAvg, in contrast to the $\mathcal{O}(\exp(-\frac{T}{\kappa}))$ rate for individual solvers Ma et al. (2018). Within this class, we further consider the linear regression problem and establish an even sharper rate under FedAvg. In addition, we propose a new variant of accelerated FedAvg based on a momentum update of Liu & Belkin (2020)–MaSS accelerated FedAvg–and establish a faster convergence rate (compared to if no acceleration is used). This stands in contrast to generic (strongly) convex stochastic problems where theoretically no rate improvement is obtained when one accelerates FedAvg. The detailed convergence results are summarized in Table 1.

**Connections with Distributed and Decentralized Optimization** Federated learning is closely related to distributed and decentralized optimization, and as such it is important to discuss connections and distinctions between our work and related results from that literature. First, when there is neither system heterogeneity, i.e. all devices participate in parameter averaging during a communication round, nor statistical heterogeneity, i.e. all devices have access to a common set of stochastic gradients, FedAvg coincides with the "Local SGD" of Stich (2019), which showed the linear speedup rate $\mathcal{O}(1/NT)$ for strongly convex and smooth functions. Woodworth et al. (2020b) and Woodworth et al. (2020a) further improved the communication complexity that guarantees the linear speedup rate. When there is only data heterogeneity, some works have continued to use the term Local SGD to refer to FedAvg, while others subsume it in more general frameworks that include decentralized model averaging based on a network topology or a mixing matrix. They have provided linear speedup analyses for strongly convex and convex problems, e.g. Khaled et al. (2020); Koloskova et al. (2020) as well as non-convex problems, e.g. Jiang & Agrawal (2018); Yu et al. (2019b); Wang & Joshi (2018). However, these results do not consider system heterogeneity, i.e. the presence of stragglers in the device network. Even with decentralized model averaging, the assumptions usually imply that model averages over all devices is the same as decentralized model averages based on network topology (e.g. Koloskova et al. (2020) Proposition 1), which precludes system heterogeneity as defined in this paper and prevalent in FL problems. For momentum accelerated FedAvg, Yu et al. (2019a) provided linear speedup analysis for non-convex problems, while results for strongly convex and convex settings are entirely lacking, even without system heterogeneity. Karimireddy et al. (2019) considers both types of heterogeneities for FedAvg, but their rate implies a linear speedup only when the number of stragglers is negligible. In contrast, our linear speedup analyses consider both types of heterogeneity present in the full federated learning setting, and are valid for any number of participating devices. We also highlight a distinction in communication efficiency when system heterogeneity is present. Moreover, our results for Nesterov accelerated FedAvg completes the picture for strongly convex and convex problems. For a detailed comparison with related works, please refer to Table 2 in Appendix Section B.

## 2 SETUP

In this paper, we study the following federated learning problem:

$$\min_{\mathbf{w}} \left\{ F(\mathbf{w}) \triangleq \sum_{k=1}^{N} p_k F_k(\mathbf{w}) \right\}, \tag{1}$$

where $N$ is the number of local devices (users/nodes/workers) and $p_k$ is the $k$-th device's weight satisfying $p_k \geq 0$ and $\sum_{k=1}^{N} p_k = 1$. In the $k$-th local device, there are $n_k$ data points: $\mathbf{x}_k^1, \mathbf{x}_k^2, \ldots, \mathbf{x}_k^{n_k}$. The local objective $F_k(\cdot)$ is defined as: $F_k(\mathbf{w}) \triangleq \frac{1}{n_k} \sum_{j=1}^{n_k} \ell\left(\mathbf{w}; \mathbf{x}_k^j\right)$, where $\ell$ denotes a user-specified loss function. Each device only has access to its local data, which gives rise to its own

local objective $F_k$. Note that we do not make any assumptions on the data distributions of each local device. The local minimum $F_k^* = \min_{\mathbf{w}} F_k(\mathbf{w})$ can be far from the global minimum of Eq (1) (data heterogeneity).

## 2.1 THE FEDERATED AVERAGING (FEDAVG) ALGORITHM

We first introduce the standard Federated Averaging (FedAvg) algorithm which was first proposed by McMahan et al. (2017). FedAvg updates the model in each device by local Stochastic Gradient Descent (SGD) and sends the latest model to the central server every $E$ steps. The central server conducts a weighted average over the model parameters received from active devices and broadcasts the latest averaged model to all devices. Formally, the updates of FedAvg at round $t$ is described as follows:

$$\mathbf{v}_{t+1}^k = \mathbf{w}_t^k - \alpha_t \mathbf{g}_{t,k}, \quad \mathbf{w}_{t+1}^k = \begin{cases} \mathbf{v}_{t+1}^k & \text{if } t+1 \notin \mathcal{I}_E, \\ \sum_{k \in \mathcal{S}_{t+1}} q_k \mathbf{v}_{t+1}^k & \text{if } t+1 \in \mathcal{I}_E, \end{cases} \tag{2}$$

where $\mathbf{w}_t^k$ is the local model parameter maintained in the $k$-th device at the $t$-th iteration and $\mathbf{g}_{t,k} := \nabla F_k(\mathbf{w}_t^k, \xi_t^k)$ is the stochastic gradient based on $\xi_t^k$, the data point sampled from $k$-th device's local data uniformly at random. $\mathcal{I}_E = \{E, 2E, \dots\}$ is the set of global communication steps, when local parameters from a set of active devices are averaged and broadcast to all devices. We use $\mathcal{S}_{t+1}$ to represent the (random) set of active devices at $t+1$. $q_k$ is a set of averaging weights that are specific to the sampling procedure used to obtain the set of active devices $\mathcal{S}_{t+1}$.

Since federated learning usually involves an enormous amount of local devices, it is often more realistic to assume only a subset of local devices is active at each communication round (system heterogeneity). In this work, we consider both the case of **full participation** where the model is averaged over all devices at each communication round, in which case $q_k = p_k$ for all $k$ and $\mathbf{w}_{t+1}^k = \sum_{k=1}^N p_k \mathbf{v}_{t+1}^k$ if $t+1 \in \mathcal{I}_E$, and the case of **partial participation** where $|\mathcal{S}_{t+1}| < N$.

With partial participation, we follow Li et al. (2020a); Karimireddy et al. (2019); Li et al. (2020b) and assume that $\mathcal{S}_{t+1}$ is obtained by one of two types of sampling schemes to simulate practical scenarios. One scheme establishes $\mathcal{S}_{t+1}$ by *i.i.d.* sampling the devices with probability $p_k$ with replacement, and uses $q_k = \frac{1}{K}$, where $K = |\mathcal{S}_{t+1}|$, while the other scheme samples $\mathcal{S}_{t+1}$ uniformly *i.i.d.* from all devices without replacement, and uses $q_k = p_k \frac{N}{K}$. Both schemes guarantee that gradient updates in FedAvg are unbiased stochastic versions of updates in FedAvg with full participation, which is important in the theoretical analysis of convergence. Because the original sampling scheme and weights proposed by McMahan et al. (2017) lacks this nice property, it is not considered in this paper. For more details on the notations and setup as well as properties of the two sampling schemes, please refer to Section A in the appendix.

## 2.2 ASSUMPTIONS

We make the following standard assumptions on the objective function $F_1, \dots, F_N$. Assumptions 1 and 2 are commonly satisfied by a range of popular objective functions, such as $\ell^2$-regularized logistic regression and cross-entropy loss functions.

**Assumption 1** (*L*-smooth). *$F_1, \cdots, F_N$ are all $L$-smooth: for all $\mathbf{v}$ and $\mathbf{w}$, $F_k(\mathbf{v}) \leq F_k(\mathbf{w}) + (\mathbf{v} - \mathbf{w})^T \nabla F_k(\mathbf{w}) + \frac{L}{2} \|\mathbf{v} - \mathbf{w}\|_2^2$.*

**Assumption 2** (Strongly-convex). *$F_1, \cdots, F_N$ are all $\mu$ -strongly convex: for all $\mathbf{v}$ and $\mathbf{w}$, $F_k(\mathbf{v}) \geq F_k(\mathbf{w}) + (\mathbf{v} - \mathbf{w})^T \nabla F_k(\mathbf{w}) + \frac{\mu}{2} \|\mathbf{v} - \mathbf{w}\|_2^2$*

**Assumption 3** (Bounded local variance). *Let $\xi_t^k$ be sampled from the $k$-th device's local data uniformly at random. The variance of stochastic gradients in each device is bounded: $\mathbb{E} \left\| \nabla F_k \left( \mathbf{w}_t^k, \xi_t^k \right) - \nabla F_k \left( \mathbf{w}_t^k \right) \right\|^2 \leq \sigma_k^2$, for $k = 1, \cdots, N$ and any $\mathbf{w}_t^k$. Let $\sigma^2 := \sum_{k=1}^N p_k \sigma_k^2$.*

**Assumption 4** (Bounded local gradient). *The expected squared norm of stochastic gradients is uniformly bounded. i.e., $\mathbb{E} \left\| \nabla F_k \left( \mathbf{w}_t^k, \xi_t^k \right) \right\|^2 \leq G^2$, for all $k = 1, ..., N$ and $t = 0, \dots, T - 1$.*

Assumptions 3 and 4 have been made in many previous works in federated learning, e.g. Yu et al. (2019b); Li et al. (2020b); Stich (2019). We provide further justification for their generality. As model average parameters become closer to $\mathbf{w}^*$, the $L$-smoothness property implies that $\mathbb{E} \|\nabla F_k(\mathbf{w}_t^k, \xi_t^k)\|^2$ and $\mathbb{E} \|\nabla F_k(\mathbf{w}_t^k, \xi_t^k) - \nabla F_k(\mathbf{w}_t^k)\|^2$ approach $\mathbb{E} \|\nabla F_k(\mathbf{w}^*, \xi_t^k)\|^2$ and

$\mathbb{E}\|\nabla F_k(\mathbf{w}^*, \xi_t^k) - \nabla F_k(\mathbf{w}^*)\|^2$. Therefore, there is no substantial difference between these assumptions and assuming the bounds at $\mathbf{w}^*$ only Koloskova et al. (2020). Furthermore, compared to assuming *bounded gradient diversity* as in related work Haddadpour & Mahdavi (2019); Li et al. (2020a), Assumption 4 is much less restrictive. When the optimality gap converges to zero, bounded gradient diversity restricts local objectives to have the same minimizer as the global objective, contradicting the heterogeneous data setting. For detailed discussions of our assumptions, please refer to Appendix Section B.

## 3   LINEAR SPEEDUP ANALYSIS OF FEDAVG

In this section, we provide convergence analyses of FedAvg for convex objectives in the general setting with both heterogeneous data (statistical heterogeneity) and partial participation (system heterogeneity). We show that for strongly convex and smooth objectives, the convergence of the optimality gap of averaged parameters across devices is $\mathcal{O}(1/KT)$, while for convex and smooth objectives, the rate is $\mathcal{O}(1/\sqrt{KT})$. Our results improve upon Li et al. (2020b); Karimireddy et al. (2019) by showing linear speedup for any number of participating devices, and upon Khaled et al. (2020); Koloskova et al. (2020) by allowing system heterogeneity. The proofs also highlight similarities and distinctions between the strongly convex and convex settings. Detailed proofs are deferred to Appendix Section E.

### 3.1   STRONGLY CONVEX AND SMOOTH OBJECTIVES

We first show that FedAvg has an $\mathcal{O}(1/KT)$ convergence rate for $\mu$-strongly convex and $L$-smooth objectives. The result relies on a technical improvement over the analysis in Li et al. (2020b). Moreover, it implies a distinction in communication efficiency that guarantees this linear speedup for FedAvg with full and partial device participation. With full participation, $E$ can be chosen as large as $\mathcal{O}(\sqrt{T/N})$ without degrading the linear speedup in the number of workers. On the other hand, with partial participation, $E$ must be $\mathcal{O}(1)$ to guarantee $\mathcal{O}(1/KT)$ convergence.

**Theorem 1.** *Let $\overline{\mathbf{w}}_T = \sum_{k=1}^N p_k \mathbf{w}_T^k$ in FedAvg, $\nu_{\max} = \max_k N p_k$, and set decaying learning rates $\alpha_t = \frac{4}{\mu(\gamma+t)}$ with $\gamma = \max\{32\kappa, E\}$ and $\kappa = \frac{L}{\mu}$. Then under Assumptions 1 to 4 with full device participation,*

$$\mathbb{E}F(\overline{\mathbf{w}}_T) - F^* = \mathcal{O}\left(\frac{\kappa \nu_{\max}^2 \sigma^2/\mu}{NT} + \frac{\kappa^2 E^2 G^2/\mu}{T^2}\right),$$

*and with partial device participation with at most $K$ sampled devices at each communication round,*

$$\mathbb{E}F(\overline{\mathbf{w}}_T) - F^* = \mathcal{O}\left(\frac{\kappa E^2 G^2/\mu}{KT} + \frac{\kappa \nu_{\max}^2 \sigma^2/\mu}{NT} + \frac{\kappa^2 E^2 G^2/\mu}{T^2}\right).$$

**Proof sketch.** Because our unified analyses of results in the main text follow the same framework with variations in technical details, we first give an outline of proof for Theorem 1 to illustrate the main ideas. For full participation, the main ingredient is a recursive contraction bound

$$\mathbb{E}\|\overline{\mathbf{w}}_{t+1} - \mathbf{w}^*\|^2 \le (1 - \mu\alpha_t)\mathbb{E}\|\overline{\mathbf{w}}_t - \mathbf{w}^*\|^2 + \alpha_t^2 \frac{1}{N}\nu_{max}^2\sigma^2 + 6\alpha_t^3 LE^2 G^2$$

where the $\mathcal{O}(\alpha_t^3 E^2 G^2)$ term is the key improvement over the bound in Li et al. (2020b), which has $\mathcal{O}(\alpha_t^2 E^2 G^2)$ instead. We then use induction to obtain a non-recursive bound on $\mathbb{E}\|\overline{\mathbf{w}}_T - \mathbf{w}^*\|^2$, which is converted to a bound on $\mathbb{E}F(\overline{\mathbf{w}}_T) - F^*$ using $L$-smoothness. For partial participation, an additional term $\mathcal{O}(\frac{1}{K}\alpha_t^2 E^2 G^2)$ of leading order resulting from sampling variance is added to the contraction bound. To facilitate the understanding of our analysis, please refer to a high-level summary in Appendix C.

**Linear speedup.** We compare our bound with that in Li et al. (2020b), which is $\mathcal{O}(\frac{1}{NT} + \frac{E^2}{KT} + \frac{E^2 G^2}{T})$. Because the term $\frac{E^2 G^2}{T}$ is also $\mathcal{O}(1/T)$ without a dependence on $N$, for any choice of $E$ their bound cannot achieve linear speedup. The improvement of our bound comes from the term $\frac{\kappa^2 E^2 G^2/\mu}{T^2}$, which

now is $\mathcal{O}(E^2/T^2)$ and so is not of leading order. As a result, all leading terms scale with $1/N$ in the full device participation setting, and with $1/K$ in the partial participation setting. This implies that in both settings, there is a *linear speedup* in the number of active workers during a communication round. We also emphasize that the reason one cannot recover the full participation bound by setting $K = N$ in the partial participation bound is due to the variance generated by sampling.

**Communication Complexity.** Our bound implies a distinction in the choice of $E$ between the full and partial participation settings. With full participation, the term involving $E$, $\mathcal{O}(E^2/T^2)$, is not of leading order $\mathcal{O}(1/T)$, so we can increase $E$ and reduce the number of communication rounds without degrading the linear speedup in iteration complexity $\mathcal{O}(1/NT)$, as long as $E = \mathcal{O}(\sqrt{T/N})$, since then $\mathcal{O}(E^2/T^2) = \mathcal{O}(1/NT)$ matches the leading term. This corresponds to a communication complexity of $T/E = \mathcal{O}(\sqrt{NT})$. In contrast, the bound in Li et al. (2020b) does not allow $E$ to scale with $\sqrt{T}$ to preserve $\mathcal{O}(1/T)$ rate, even for full participation. On the other hand, with partial participation, $\frac{\kappa E^2 G^2/\mu}{KT}$ is also a leading term, and so $E$ must be $\mathcal{O}(1)$. In this case, our bound still yields a linear speedup in $K$, which is also confirmed by experiments. The requirement that $E = \mathcal{O}(1)$ in order to achieve linear speedup in partial participation cannot be removed for our sampling schemes, as the term $\frac{\kappa E^2 G^2/\mu}{KT}$ comes from variance in the sampling process, which is $\mathcal{O}(E^2/T^2)$. In Proposition 1 in Section E of the appendix, we provide a problem instance where the dependence of the sampling variance on $E$ is tight.

**Comparison with related works.** To better understand the significance of the obtained bound, we compare our rates to the best-known results in related settings. Haddadpour & Mahdavi (2019) proves a linear speedup $\mathcal{O}(1/KT)$ result for strongly convex and smooth objectives[1], with $\mathcal{O}(K^{1/3}T^{2/3})$ communication complexity with non-*i.i.d.* data and partial participation. However, their results build on the bounded gradient diversity assumption, which implies the existence of $\mathbf{w}^*$ that minimizes all local objectives (see discussions in Section 2.2 and Appendix B), effectively removing statistical heterogeneity. The bound in Koloskova et al. (2020) matches our bound in the full participation case, but their framework excludes partial participation (Koloskova et al., 2020, Proposition 1). The result of Karimireddy et al. (2019) applies to the full FL setting, but only has linear speedup when $K = \mathcal{O}(N)$, i.e. close to full participation, whereas our result has linear speedup for any number of participating devices. When there is no data heterogeneity, i.e. in the classical distributed optimization paradigm, communication complexity can be further improved, e.g. Woodworth et al. (2020b;a), but such results are not directly comparable to ours since we consider the setting where individual devices have access to different datasets.

## 3.2 CONVEX SMOOTH OBJECTIVES

Next we provide linear speedup analysis of FedAvg with convex and smooth objectives and show that the optimality gap is $\mathcal{O}(1/\sqrt{KT})$. This result complements the strongly convex case in the previous part, as well as the non-convex smooth setting in Jiang & Agrawal (2018); Yu et al. (2019b); Haddadpour & Mahdavi (2019), where $\mathcal{O}(1/\sqrt{KT})$ results are given in terms of averaged gradient norm, and it also extends the result in Khaled et al. (2020), which has linear speedup in the convex setting, but only for full participation.

**Theorem 2.** *Under Assumptions 1,3,4 and constant learning rate $\alpha_t = \mathcal{O}(\sqrt{\frac{N}{T}})$, FedAvg satisfies*

$$\min_{t \le T} F(\overline{\mathbf{w}}_t) - F(\mathbf{w}^*) = \mathcal{O}\left(\frac{\nu_{\max}^2 \sigma^2}{\sqrt{NT}} + \frac{NE^2 LG^2}{T}\right)$$

*with full participation, and with partial device participation with $K$ sampled devices at each communication round and learning rate $\alpha_t = \mathcal{O}(\sqrt{\frac{K}{T}})$,*

$$\min_{t \le T} F(\overline{\mathbf{w}}_t) - F(\mathbf{w}^*) = \mathcal{O}\left(\frac{\nu_{\max}^2 \sigma^2}{\sqrt{KT}} + \frac{E^2 G^2}{\sqrt{KT}} + \frac{KE^2 LG^2}{T}\right).$$

The analysis again relies on a recursive bound, but without contraction:

$$\mathbb{E}\|\overline{\mathbf{w}}_{t+1} - \mathbf{w}^*\|^2 + \alpha_t(F(\overline{\mathbf{w}}_t) - F(\mathbf{w}^*)) \le \mathbb{E}\|\overline{\mathbf{w}}_t - \mathbf{w}^*\|^2 + \alpha_t^2 \frac{1}{N}\nu_{\max}^2 \sigma^2 + 6\alpha_t^3 E^2 LG^2$$

---

[1]Their result applies to a larger class of non-convex objectives that satisfy the Polyak-Lojasiewicz condition.

which is then summed over time steps to give the desired bound, with $\alpha_t = \mathcal{O}(\sqrt{\frac{N}{T}})$.

**Choice of $E$ and linear speedup.** With full participation, as long as $E = \mathcal{O}(T^{1/4}/N^{3/4})$, the convergence rate is $\mathcal{O}(1/\sqrt{NT})$ with $\mathcal{O}(N^{3/4}T^{3/4})$ communication rounds. In the partial participation setting, $E$ must be $O(1)$ in order to achieve linear speedup of $\mathcal{O}(1/\sqrt{KT})$. This is again due to the fact that the sampling variance $\mathbb{E}\|\overline{\mathbf{w}}_t - \overline{\mathbf{v}}_t\|^2 = \mathcal{O}(\alpha_t^2 E^2 G^2)$ cannot be made independent of $E$, as illustrated by Proposition 1. See also the proof in Section E for how the sampling variance and the term $E^2 G^2/\sqrt{KT}$ are related. Our result again demonstrates the difference in communication complexities between full and partial participation, and is to our knowledge the first result on linear speedup in the general federated learning setting with both heterogeneous data and partial participation for convex objectives.

## 4 LINEAR SPEEDUP ANALYSIS OF NESTEROV ACCELERATED FEDAVG

A natural extension of the FedAvg algorithm is to use momentum-based local updates instead of local SGD updates in order to accelerate FedAvg. As we know from stochastic optimization, Nesterov and other momentum updates fail to provably accelerate over SGD (Liu & Belkin (2020); Kidambi et al. (2018); Liu et al. (2018); Yuan & Ma (2020)). This is in contrast to the classical acceleration result of Nesterov-accelerated gradient descent over GD. Thus in the FL setting, the best provable convergence rate for FedAvg with Nesterov updates is the same as FedAvg with SGD updates. Nevertheless, Nesterov and other momentum updates are frequently used in practice, in both non-FL and FL settings, and are observed to perform better empirically. In fact, previous works such as Stich (2019) on FedAvg with vanilla SGD uses FedAvg with Nesterov or other momentum updates in their experiments to achieve target accuracy. Because of the popularity of Nesterov and other momentum-based methods, understanding the linear speedup behavior of FedAvg with such local updates is important. To our knowledge, the only convergence analyses of FedAvg with momentum-based stochastic updates focus on the non-convex smooth case Huo et al. (2020); Yu et al. (2019a); Li et al. (2020a), and no results existed in the convex smooth settings. In this section, we complete the picture by providing the first $\mathcal{O}(1/KT)$ and $\mathcal{O}(1/\sqrt{KT})$ convergence results for Nesterov-accelerated FedAvg for convex objectives that match the rates for FedAvg with SGD updates. Detailed proofs of convergence results in this section are deferred to Appendix Section F.

### 4.1 STRONGLY CONVEX AND SMOOTH OBJECTIVES

The Nesterov Accelerated FedAvg algorithm follows the updates:

$$\mathbf{v}_{t+1}^k = \mathbf{w}_t^k - \alpha_t \mathbf{g}_{t,k}, \quad \mathbf{w}_{t+1}^k = \begin{cases} \mathbf{v}_{t+1}^k + \beta_t(\mathbf{v}_{t+1}^k - \mathbf{v}_t^k) & \text{if } t+1 \notin \mathcal{I}_E, \\ \sum_{k \in \mathcal{S}_{t+1}} q_k \left[ \mathbf{v}_{t+1}^k + \beta_t(\mathbf{v}_{t+1}^k - \mathbf{v}_t^k) \right] & \text{if } t+1 \in \mathcal{I}_E, \end{cases}$$

where $\mathbf{g}_{t,k} := \nabla F_k(\mathbf{w}_t^k, \xi_t^k)$ is the stochastic gradient sampled on the $k$-th device at time $t$, and $q_k$ again depends on participation and sampling schemes.

**Theorem 3.** *Let $\overline{\mathbf{v}}_T = \sum_{k=1}^N p_k \mathbf{v}_T^k$ in Nesterov accelerated FedAvg, and set learning rates $\alpha_t = \frac{6}{\mu}\frac{1}{t+\gamma}$, $\beta_{t-1} = \frac{3}{14(t+\gamma)(1-\frac{6}{t+\gamma})\max\{\mu,1\}}$. Then under Assumptions 1,2,3,4 with full device participation,*

$$\mathbb{E}F(\overline{\mathbf{v}}_T) - F^* = \mathcal{O}\left( \frac{\kappa\nu_{\max}^2\sigma^2/\mu}{NT} + \frac{\kappa^2 E^2 G^2/\mu}{T^2} \right),$$

*and with partial device participation with $K$ sampled devices at each communication round,*

$$\mathbb{E}F(\overline{\mathbf{v}}_T) - F^* = \mathcal{O}\left( \frac{\kappa\nu_{\max}^2\sigma^2/\mu}{NT} + \frac{\kappa E^2 G^2/\mu}{KT} + \frac{\kappa^2 E^2 G^2/\mu}{T^2} \right).$$

Similar to FedAvg, the key step in the proof of this result is a recursive contraction bound, but different in that it involves three time steps, due to the update format of Nesterov SGD (see Lemma 7 in Appendix F.1). Then we can again use induction and $L$-smoothness to obtain the desired bound. To our knowledge, this is the first convergence result for Nesterov accelerated FedAvg in the strongly convex and smooth setting. The same discussion about linear speedup of FedAvg applies to the Nesterov accelerated variant. In particular, to achieve $\mathcal{O}(1/NT)$ linear speedup, $T$ iterations of the algorithm require only $\mathcal{O}(\sqrt{NT})$ communication rounds with full participation.

## 4.2 CONVEX SMOOTH OBJECTIVES

We now show that the optimality gap of Nesterov-accelerated FedAvg has $\mathcal{O}(1/\sqrt{KT})$ rate for convex and smooth objectives. This result complements the strongly convex case in the previous part, as well as the non-convex smooth setting in Huo et al. (2020); Yu et al. (2019a); Li et al. (2020a), where a similar $\mathcal{O}(1/\sqrt{KT})$ rate is given in terms of averaged gradient norm.

**Theorem 4.** *Set learning rates* $\alpha_t = \beta_t = \mathcal{O}(\sqrt{\frac{N}{T}})$. *Then under Assumptions 1,3,4 Nesterov accelerated FedAvg with full device participation has rate*

$$\min_{t \leq T} F(\overline{\mathbf{v}}_t) - F^* = \mathcal{O}\left(\frac{\nu_{\max}^2 \sigma^2}{\sqrt{NT}} + \frac{NE^2 LG^2}{T}\right),$$

*and with partial device participation with $K$ sampled devices at each communication round,*

$$\min_{t \leq T} F(\overline{\mathbf{v}}_t) - F^* = \mathcal{O}\left(\frac{\nu_{\max}^2 \sigma^2}{\sqrt{KT}} + \frac{E^2 G^2}{\sqrt{KT}} + \frac{KE^2 LG^2}{T}\right).$$

We emphasize again that in the stochastic optimization setting, the optimal convergence rate that FedAvg wtih Nesterov udpates can achieve is the same as FedAvg with SGD updates. However, due to the popularity and superior performance of momentum methods in practice, it is still important to understand the linear speedup behavior of such FedAvg variants. Our results in this section fill exactly this gap.

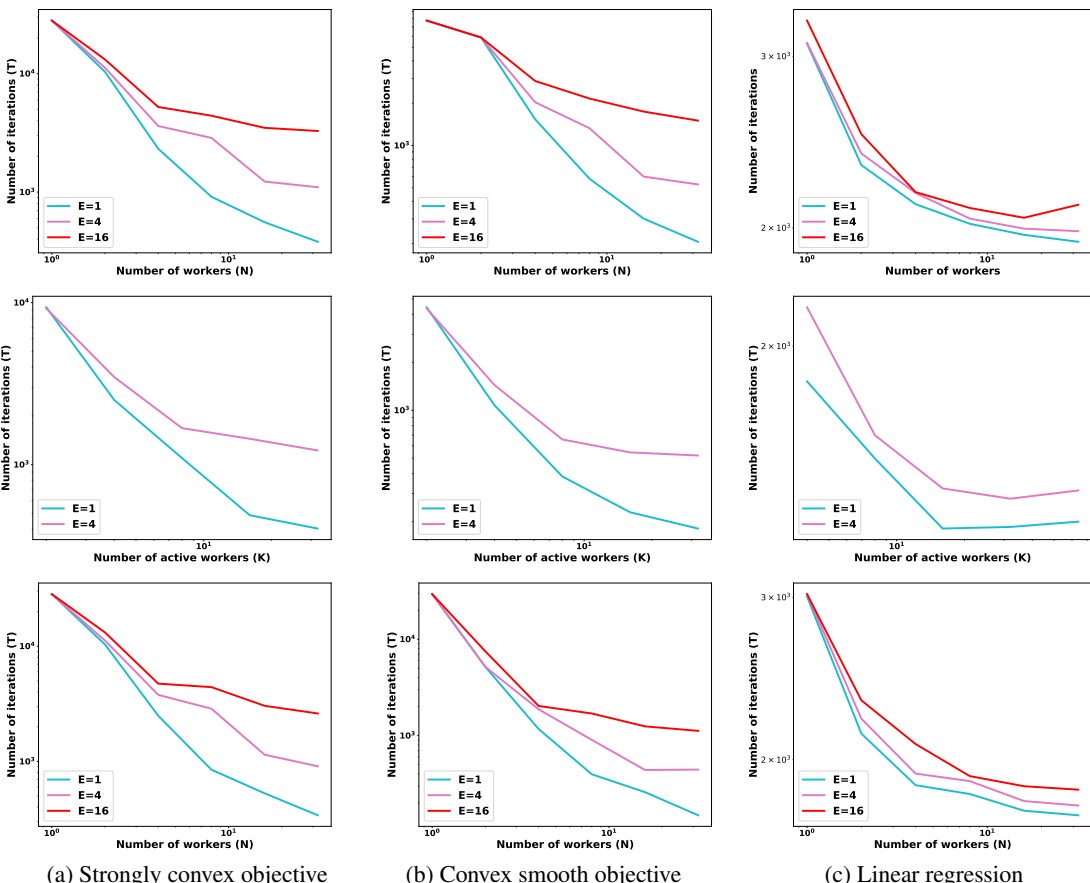

|  (a) Strongly convex objective | (b) Convex smooth objective | (c) Linear regression |

Figure 1: The linear speedup of FedAvg in full participation, partial participation, and the linear speedup of Nesterov accelerated FedAvg, respectively.

## 5 NUMERICAL EXPERIMENTS

In this section, we empirically examine the linear speedup convergence of FedAvg and Nesterov accelerated FedAvg in various settings, including strongly convex function, convex smooth function, and overparameterized objectives, as analyzed in previous sections.

**Setup.** Following the experimental setting in Stich (2019), we conduct experiments on both synthetic datasets and real-world dataset w8a Platt (1998) ($d = 300, n = 49749$). We consider the distributed objectives $F(\mathbf{w}) = \sum_{k=1}^{N} p_k F_k(\mathbf{w})$, and the objective function on the $k$-th local device includes three cases: 1) **Strongly convex objective**: the regularized binary logistic regression problem, $F_k(\mathbf{w}) = \frac{1}{N_k} \sum_{i=1}^{N_k} \log(1 + \exp(-y_i^k \mathbf{w}^T \mathbf{x}_i^k) + \frac{\lambda}{2} \|\mathbf{w}\|^2$. The regularization parameter is set to $\lambda = 1/n \approx 2e - 5$. 2) **Convex smooth objective**: the binary logistic regression problem without regularization. 3) **Overparameterized setting**: the linear regression problem without adding noise to the label, $F_k(\mathbf{w}) = \frac{1}{N_k} \sum_{i=1}^{N_k} (\mathbf{w}^T \mathbf{x}_i^k + b - y_i^k)^2$.

**Linear speedup of FedAvg and Nesterov accelerated FedAvg.** To verify the linear speedup convergence as shown in Theorems 1 2 3 4, we evaluate the number of iterations needed to reach $\epsilon$-accuracy in three objectives. We initialize all runs with $\mathbf{w}_0 = \mathbf{0}_d$ and measure the number of iterations to reach the target accuracy $\epsilon$. For each configuration $(E, K)$, we extensively search the learning rate from $\min(\eta_0, \frac{nc}{1+t})$, where $\eta_0 \in \{0.1, 0.12, 1, 32\}$ according to different problems and $c$ can take the values $c = 2^i \ \forall i \in \mathbb{Z}$. As the results shown in Figure 1, the number of iterations decreases as the number of (active) workers increasing, which is consistent for FedAvg and Nesterov accelerated FedAvg across all scenarios. For additional experiments on the impact of $E$, detailed experimental setup, and hyperparameter setting, please refer to the Appendix Section I.

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

## A  ADDITIONAL NOTATIONS AND BOUNDS FOR SAMPLING SCHEMES

In this section, we introduce additional notations that are used throughout the proofs. Following common practice, e.g. Stich (2019); Li et al. (2020b), we define two virtual sequences $\overline{\mathbf{v}}_t = \sum_{k=1}^N p_k \mathbf{v}_t^k$ and $\overline{\mathbf{w}}_t = \sum_{k=1}^N p_k \mathbf{w}_t^k$, where we recall the FedAvg updates from (2):

$$\mathbf{v}_{t+1}^k = \mathbf{w}_t^k - \alpha_t \mathbf{g}_{t,k}, \quad \mathbf{w}_{t+1}^k = \begin{cases} \mathbf{v}_{t+1}^k & \text{if } t+1 \notin \mathcal{I}_E, \\ \sum_{k \in \mathcal{S}_{t+1}} q_k \mathbf{v}_{t+1}^k & \text{if } t+1 \in \mathcal{I}_E. \end{cases}$$

The following observations apply to FedAvg updates, while Nesterov accelerated FedAvg requires modifications. For full device participation or partial participation with $t \notin \mathcal{I}_E$, note that $\overline{\mathbf{v}}_t = \overline{\mathbf{w}}_t = \sum_{k=1}^N p_k \mathbf{v}_t^k$. For partial participation with $t \in \mathcal{I}_E$, $\overline{\mathbf{w}}_t \neq \overline{\mathbf{v}}_t$ since $\overline{\mathbf{v}}_t = \sum_{k=1}^N p_k \mathbf{v}_t^k$ while $\overline{\mathbf{w}}_t = \sum_{k \in \mathcal{S}_t} q_k \mathbf{w}_t^k$. However, we can use unbiased sampling strategies such that $\mathbb{E}_{\mathcal{S}_t} \overline{\mathbf{w}}_t = \overline{\mathbf{v}}_t$. Note that $\overline{\mathbf{v}}_{t+1}$ is one-step SGD from $\overline{\mathbf{w}}_t$.

$$\overline{\mathbf{v}}_{t+1} = \overline{\mathbf{w}}_t - \alpha_t \mathbf{g}_t, \tag{3}$$

where $\mathbf{g}_t = \sum_{k=1}^N p_k \mathbf{g}_{t,k}$ is the one-step stochastic gradient averaged over all devices.

$$\mathbf{g}_{t,k} = \nabla F_k\left(\mathbf{w}_t^k, \xi_t^k\right),$$

Similarly, we denote the expected one-step gradient $\overline{\mathbf{g}}_t = \mathbb{E}_{\xi_t}[\mathbf{g}_t] = \sum_{k=1}^N p_k \mathbb{E}_{\xi_t^k} \mathbf{g}_{t,k}$, where

$$\mathbb{E}_{\xi_t^k} \mathbf{g}_{t,k} = \nabla F_k\left(\mathbf{w}_t^k\right), \tag{4}$$

and $\xi_t = \{\xi_t^k\}_{k=1}^N$ denotes random samples at all devices at time step $t$.

Since in this work we also consider the case of partial participation, the sampling strategy to approximate the system heterogeneity can also affect the convergence. Here we follow the prior works Li et al. (2020b) and Li et al. (2020a) and consider two types of sampling schemes that guarantee $\mathbb{E}_{\mathcal{S}_t} \overline{\mathbf{w}}_t = \overline{\mathbf{v}}_t$. The sampling scheme I establishes $\mathcal{S}_{t+1}$ by *i.i.d.* sampling the devices according to probabilities $p_k$ with replacement, and setting $q_k = \frac{1}{K}$. In this case the upper bound of expected square norm of $\overline{\mathbf{w}}_{t+1} - \overline{\mathbf{v}}_{t+1}$ is given by (Li et al., 2020b, Lemma 5):

$$\mathbb{E}_{\mathcal{S}_{t+1}} \|\overline{\mathbf{w}}_{t+1} - \overline{\mathbf{v}}_{t+1}\|^2 \leq \frac{4}{K} \alpha_t^2 E^2 G^2. \tag{5}$$

The sampling scheme II establishes $\mathcal{S}_{t+1}$ by uniformly sampling all devices without replacement and setting $q_k = p_k \frac{N}{K}$, in which case we have

$$\mathbb{E}_{\mathcal{S}_{t+1}} \|\overline{\mathbf{w}}_{t+1} - \overline{\mathbf{v}}_{t+1}\|^2 \leq \frac{4(N-K)}{K(N-1)} \alpha_t^2 E^2 G^2. \tag{6}$$

We summarize these upper bounds as follows:

$$\mathbb{E}_{\mathcal{S}_{t+1}} \|\overline{\mathbf{w}}_{t+1} - \overline{\mathbf{v}}_{t+1}\|^2 \leq \frac{4}{K} \alpha_t^2 E^2 G^2. \tag{7}$$

and this bound will be used in the convergence proof of the partial participation result.

## B  COMPARISON OF CONVERGENCE RATES WITH RELATED WORKS

In this section, we compare our convergence rate with the best-known results in the literature (see Table 2). In Haddadpour & Mahdavi (2019), the authors provide $\mathcal{O}(1/NT)$ convergence rate of non-convex problems under Polyak-Łojasiewicz (PL) condition, which means their results can directly apply to the strongly convex problems. However, their assumption is based on bounded gradient diversity, defined as follows:

$$\Lambda(\mathbf{w}) = \frac{\sum_k p_k \|\nabla F_k(\mathbf{w})\|_2^2}{\|\sum_k p_k \nabla F_k(\mathbf{w})\|_2^2} \leq B$$

This is a more restrictive assumption comparing to assuming bounded gradient under the case of target accuracy $\epsilon \to 0$ and PL condition. To see this, consider the gradient diversity at the global optimal

$\mathbf{w}^*$, i.e., $\Lambda(\mathbf{w}^*) = \frac{\sum_k p_k \|\nabla F_k(\mathbf{w})\|_2^2}{\|\sum_k p_k \nabla F_k(\mathbf{w})\|_2^2}$. For $\Lambda(\mathbf{w}^*)$ to be bounded, it requires $\|\nabla F_k(\mathbf{w}^*)\|_2^2 = 0, \forall k$. This indicates $\mathbf{w}^*$ is also the minimizer of each local objective, which contradicts to the practical setting of heterogeneous data. Therefore, their bound is not effective for arbitrary small $\epsilon$-accuracy under general heterogeneous data while our convergence results still hold in this case.

In Karimireddy et al. (2019), the linear speedup convergence rate of FedAvg are provided for strongly convex, general convex, and non-convex problems under full participation setting. However, their rate does not enjoy linear speedup for any number of devices while our results apply to any valid $K \leq N$. For example, they provides an optimality gap of $\mathcal{O}\left((1 - \frac{K}{N})E/T\right)$ for the strongly convex case (Karimireddy et al., 2019, Theorem V). With partial participation, and when $K = O(1)$, their convergence rate is $\mathcal{O}(E/T)$ which does not have linear speedup. Under partial participation, the FedAvg analyses in Karimireddy et al. (2019) requires $E = \mathcal{O}(1)$. For example, the strongly convex result $\mathcal{O}((1 - \frac{K}{N})E/T)$ in Theorem V is $\mathcal{O}(E/T)$ when $K = O(1)$ and is $\mathcal{O}(E/NT)$ when $K = \mathcal{O}(N)$. In either case, to achieve a $\mathcal{O}(1/T)$ convergence rate, it requires $E = \mathcal{O}(1)$ as well. Similar conclusion also holds for the general convex problem.

| Reference | Convergence rate | E | NonIID | Participation | Extra Assumptions | Setting |
|---|---|---|---|---|---|---|
| FedAvgLi et al. (2020b) | $\mathcal{O}(\frac{E^2}{T})$ | $\mathcal{O}(1)$ | ✓ | Partial | Bounded gradient | Strongly convex |
| FedAvgHaddadpour & Mahdavi (2019) | $\mathcal{O}(\frac{1}{KT})$ | $\mathcal{O}(K^{-1/3}T^{2/3})^\dagger$ | ✓[‡‡] | Partial | Bounded gradient diversity | Strongly convex[§] |
| FedAvgKoloskova et al. (2020) | $\mathcal{O}(\frac{1}{NT})$ | $\mathcal{O}(N^{-1/2}T^{1/2})$ | ✓ | Full | Bounded gradient | Strongly convex |
| FedAvgKarimireddy et al. (2019) | $\mathcal{O}(\frac{1}{NT})^{\dagger\dagger}$ | $\mathcal{O}(N^{-1/2}T^{1/2})^{\dagger\dagger}$ | ✓ | Partial | Bounded gradient dissimilarity | Strongly convex |
| FedAvg/N-FedAvg (our work) | $\mathcal{O}(\frac{1}{KT})$ | $\mathcal{O}(N^{-1/2}T^{1/2})^\ddagger$ | ✓ | Partial | Bounded gradient | Strongly convex |
| FedAvgKhaled et al. (2020) | $\mathcal{O}(\frac{1}{\sqrt{NT}})$ | $\mathcal{O}(N^{-3/2}T^{1/2})$ | ✓ | Full | Bounded gradient | Convex |
| FedAvgKoloskova et al. (2020) | $\mathcal{O}(\frac{1}{\sqrt{NT}})$ | $\mathcal{O}(N^{-3/4}T^{1/4})$ | ✓ | Full | Bounded gradient | Convex |
| FedAvgKarimireddy et al. (2019) | $\mathcal{O}(\frac{1}{\sqrt{NT}})^{\dagger\dagger}$ | $\mathcal{O}(N^{-3/4}T^{1/4})^{\dagger\dagger}$ | ✓ | Partial | Bounded gradient dissimilarity | Convex |
| FedAvg/N-FedAvg (our work) | $\mathcal{O}\left(\frac{1}{\sqrt{KT}}\right)$ | $\mathcal{O}(N^{-3/4}T^{1/4})^\ddagger$ | ✓ | Partial | Bounded gradient | Convex |
| FedAvg | $\mathcal{O}\left(\exp(-\frac{NT}{E\kappa_1})\right)$ | $\mathcal{O}(T^\beta)$ | ✓ | Partial | Bounded gradient | Overparameterized LR |
| FedMass | $\mathcal{O}\left(\exp(-\frac{NT}{E\sqrt{\kappa_1\tilde{\kappa}}})\right)$ | $\mathcal{O}(T^\beta)$ | ✓ | Partial | Bounded gradient | Overparameterized LR |

Table 2: A high-level summary of the convergence results in this paper compared to prior state-of-the-art FL algorithms. This table only highlights the dependence on $T$ (number of iterations), $E$ (the maximal number of local steps), $N$ (the total number of devices), and $K \leq N$ the number of participated devices. $\kappa$ is the condition number of the system and $\beta \in (0, 1)$. We denote Nesterov accelerated FedAvg as N-FedAvg in this table.
$\dagger$ This $E$ is obtained under i.i.d. setting.
$\ddagger$ This $E$ is obtained under full participation setting.
$\S$ In Haddadpour & Mahdavi (2019), the convergence rate is for non-convex smooth problems with PL condition, which also applies to strongly convex problems. Therefore, we compare it with our strongly convex results here.
$\ddagger\ddagger$ The bounded gradient diversity assumption is not applicable for general heterogeneous data when converging to arbitrarily small $\epsilon$-accuracy (see discussions in Sec B).
$\dagger\dagger$ Although the results in Karimireddy et al. (2019) is applicable for partial participation setting, their results only achieve linear speedup under full participation setting $K = N$ while we show linear speedup convergence for $K \leq N$ (see discussions in Sec B). The $E$ in the table is obtained under full participation. Under partial participation, the communication complexity is $E = \mathcal{O}(1)$.

## C  A HIGH-LEVEL SUMMARY OF FEDAVG ANALYSIS

To facilitate the understanding of our analysis and highlight the improvement of our work comparing to prior arts, we summarize the general steps used in the proofs across the various settings. In this section, we take the strongly convex case as an example to illustrate our analysis. The corresponding proof for general convex functions follows the same framework.

**One step progress bound**
This step establishes the progress of distance ($\|\overline{\mathbf{w}}_t - \mathbf{w}^*\|^2$) to optimal solution after one step SGD update (see line 9, Alg 1), as the following equation shows:

$$\mathbb{E}\|\overline{\mathbf{w}}_{t+1} - \mathbf{w}^*\|^2 \leq \mathcal{O}(\eta_t \mathbb{E}\|\overline{\mathbf{w}}_t - \mathbf{w}^*\|^2 + \alpha_t^2 \sigma^2/N + \alpha_t^3 E^2 G^2).$$

---

**Algorithm 1** FEDAVG: Federated Averaging
---
1: **Server input:** initial model $\mathbf{w}_0$, initial step size $\alpha_0$, local steps $E$.
2: **Client input:**
3: **for** each round $r = 0, 1, ..., R$, where $r = t * E$ **do**
4:     Sample clients $\mathcal{S}_t \subseteq \{1, ..., N\}$
5:     Broadcast $\mathbf{w}$ to all clients $k \in \mathcal{S}_t$
6:     **for** each client $k \subseteq \mathcal{S}_t$ **do**
7:        initialize local model $\mathbf{w}_t^k = \mathbf{w}$
8:        **for** $t = r * E + 1, \dots, (r + 1) * E$ **do**
9:          $\mathbf{w}_{t+1}^k = \mathbf{w}_t^k - \alpha_t \mathbf{g}_{t,k}$
10:       **end for**
11:     **end for**
12:     Average the local models at server end: $\overline{\mathbf{w}}_t = \sum_{k \in \mathcal{S}_t} \mathbf{w}_t^k$.
13: **end for**

---

The above bound consists of three main ingredients, the distance to optima in previous step (with $\eta_t \in (0, 1)$ to obtained a contraction bound), the variance of stochastic gradients in local clients (second term), the variance across different clients (third term). Notice that the third term in this bound is the primary source of improvement in the rate. Comparing to the bound in Li et al. (2020b), we improve the third term from $\mathcal{O}(\alpha_t^2 E^2 G^2)$ to $\mathcal{O}(\alpha_t^2 E^2 G^2)$, which enables the linear speedup in the convergence rate.

**Iterative deduction**
This step uses the *one step progress bound* iteratively to connect the the current distance to optimal solution with the initial distance ($\|\overline{\mathbf{w}}_0 - \mathbf{w}^*\|^2$), as follows:

$$\mathbb{E}\|\overline{\mathbf{w}}_{t+1} - \mathbf{w}^*\|^2 \leq \mathcal{O}(\mathbb{E}\|\overline{\mathbf{w}}_0 - \mathbf{w}^*\|^2 \frac{1}{T}).$$

Then we can use the distance to optima to upper bound the optimality gap ($F(\mathbf{w}_t) - F^* \leq \mathcal{O}(1/T)$), as follows:

$$\mathbb{E}(F(\overline{\mathbf{w}}_t)) - F^* \leq \mathcal{O}(\mathbb{E}\|\overline{\mathbf{w}}_t - \mathbf{w}^*\|^2).$$

The convergence rate of the optimality gap is equally obtained as the convergence rate of the distance to optima.

**From full participation to partial participation**
There are three sources of variances that affect the convergence rate. The first two sources come from the variances of within local clients and across clients (second and third term in one step progress bound). The partial participation, which involves a sampling procedure, is the third source of variance. Therefore, comparing to the rate in full participation, this will add another term of variance into the convergence rate, where we follow a similar derivation as in Li et al. (2020b).

## D    TECHNICAL LEMMAS

To facilitate reading, we first summarize some basic properties of $L$-smooth and $\mu$-strongly convex functions, found in e.g. Rockafellar (1970), which are used in various steps of proofs in the appendix.

**Lemma 1.** *Let $F$ be a convex $L$-smooth function. Then we have the following inequalities:*

*1. Quadratic upper bound: $0 \leq F(\mathbf{w}) - F(\mathbf{w}') - \langle \nabla F(\mathbf{w}'), \mathbf{w} - \mathbf{w}' \rangle \leq \frac{L}{2}\|\mathbf{w} - \mathbf{w}'\|^2$.*

*2. Coercivity: $\frac{1}{L}\|\nabla F(\mathbf{w}) - \nabla F(\mathbf{w}')\|^2 \leq \langle \nabla F(\mathbf{w}) - \nabla F(\mathbf{w}'), \mathbf{w} - \mathbf{w}' \rangle$.*

*3. Lower bound: $F(\mathbf{w}) \geq F(\mathbf{w}') + \langle \nabla F(\mathbf{w}'), \mathbf{w} - \mathbf{w}' \rangle + \frac{1}{2L}\|\nabla F(\mathbf{w}) - \nabla F(\mathbf{w}')\|^2$. In particular, $\|\nabla F(\mathbf{w})\|^2 \leq 2L(F(\mathbf{w}) - F(\mathbf{w}^*))$.*

*4. Optimality gap: $F(\mathbf{w}) - F(\mathbf{w}^*) \leq \langle \nabla F(\mathbf{w}), \mathbf{w} - \mathbf{w}^* \rangle$.*

**Lemma 2.** *Let $F$ be a $\mu$-strongly convex function. Then*

$$F(\mathbf{w}) \leq F(\mathbf{w}') + \langle \nabla F(\mathbf{w}'), \mathbf{w} - \mathbf{w}' \rangle + \frac{1}{2\mu}\|\nabla F(\mathbf{w}) - \nabla F(\mathbf{w}')\|^2$$

$$F(\mathbf{w}) - F(\mathbf{w}^*) \le \frac{1}{2\mu}\|\nabla F(\mathbf{w})\|^2$$

# E  Proof of Convergence Results for FedAvg

## E.1  Strongly Convex Smooth Objectives

To organize our proofs more effectively and highlight the significance of our results compared to prior works, we first state the following key lemmas used in proofs of main results and defer their proofs to later.

**Lemma 3** (**One step progress, strongly convex**). *Let $\overline{\mathbf{w}}_t = \sum_{k=1}^{N} p_k \mathbf{w}_t^k$, and suppose our functions satisfy Assumptions 1,2,3,4, and set step size $\alpha_t = \frac{4}{\mu(\gamma+t)}$ with $\gamma = \max\{32\kappa, E\}$ and $\kappa = \frac{L}{\mu}$, then the updates of FedAvg with full participation satisfy*

$$\mathbb{E}\|\overline{\mathbf{w}}_{t+1} - \mathbf{w}^*\|^2 \le (1 - \mu\alpha_t)\mathbb{E}\|\overline{\mathbf{w}}_t - \mathbf{w}^*\|^2 + \alpha_t^2 \frac{1}{N}\nu_{max}^2\sigma^2 + 6E^2 L\alpha_t^3 G^2.$$

We emphasize that the above lemma is the key step that allows us to obtain a bound that improves on the convergence result of Li et al. (2020b) with linear speedup. Its proof will make use of the following two results.

**Lemma 4** (**Bounding gradient variance (Lemma 2 Li et al. (2020b))**). *Given Assumption 3, the upper bound of gradient variance is given as follows,*

$$\mathbb{E}\|\mathbf{g}_t - \overline{\mathbf{g}}_t\|^2 \le \sum_{k=1}^{N} p_k^2 \sigma_k^2.$$

**Lemma 5** (**Bounding the divergence of $\mathbf{w}_t^k$ (Lemma 3 Li et al. (2020b))**). *Given Assumption 4, and assume that $\alpha_t$ is non-increasing and $\alpha_t \le 2\alpha_{t+E}$ for all $t \ge 0$, we have*

$$\mathbb{E}\left[\sum_{k=1}^{N} p_k\|\overline{\mathbf{w}}_t - \mathbf{w}_t^k\|^2\right] \le 4E^2\alpha_t^2 G^2.$$

We now restate Theorem 1 from the main text and then prove it using Lemma 3.

**Theorem 1.** *Let $\overline{\mathbf{w}}_T = \sum_{k=1}^{N} p_k \mathbf{w}_T^k$ in FedAvg, $\nu_{\max} = \max_k Np_k$, and set decaying learning rates $\alpha_t = \frac{4}{\mu(\gamma+t)}$ with $\gamma = \max\{32\kappa, E\}$ and $\kappa = \frac{L}{\mu}$. Then under Assumptions 1,2,3,4 with full device participation,*

$$\mathbb{E}F(\overline{\mathbf{w}}_T) - F^* = \mathcal{O}\left(\frac{\kappa\nu_{\max}^2\sigma^2/\mu}{NT} + \frac{\kappa^2 E^2 G^2/\mu}{T^2}\right)$$

*and with partial device participation with at most $K$ sampled devices at each communication round,*

$$\mathbb{E}F(\overline{\mathbf{w}}_T) - F^* = \mathcal{O}\left(\frac{\kappa E^2 G^2/\mu}{KT} + \frac{\kappa\nu_{\max}^2\sigma^2/\mu}{NT} + \frac{\kappa^2 E^2 G^2/\mu}{T^2}\right)$$

*Proof.* The road map of the proof for full device participation contains three steps. First, we establish a recursive relationship between $\mathbb{E}\|\overline{\mathbf{w}}_{t+1} - \mathbf{w}^*\|^2$ and $\mathbb{E}\|\overline{\mathbf{w}}_t - \mathbf{w}^*\|^2$, upper bounding the progress of FedAvg from step $t$ to step $t+1$. Second, we show that $\mathbb{E}\|\overline{\mathbf{w}}_t - \mathbf{w}^*\|^2 = \mathcal{O}(\frac{\nu_{max}^2\sigma^2/\mu}{tN} + \frac{E^2 LG^2/\mu^2}{t^2})$ by induction using the recursive relationship from the previous step. Third, we use the property of $L$-smoothness to bound the optimality gap by $\mathbb{E}\|\overline{\mathbf{w}}_t - \mathbf{w}^*\|^2$.

By Lemma 3, we have the following upper bound for the one step progress:

$$\mathbb{E}\|\overline{\mathbf{w}}_{t+1} - \mathbf{w}^*\|^2 \le (1 - \mu\alpha_t)\mathbb{E}\|\overline{\mathbf{w}}_t - \mathbf{w}^*\|^2 + \alpha_t^2 \frac{1}{N}\nu_{max}^2\sigma^2 + 6E^2 L\alpha_t^3 G^2.$$

We show next that $\mathbb{E}\|\overline{\mathbf{w}}_t - \mathbf{w}^*\|^2 = O(\frac{\nu_{max}^2\sigma^2/\mu}{tN} + \frac{E^2LG^2/\mu^2}{t^2})$ using induction. To simplify the presentation, we denote $C \equiv 6E^2LG^2$ and $D \equiv \frac{1}{N}\nu_{max}^2\sigma^2$. Suppose that we have the bound $\mathbb{E}\|\overline{\mathbf{w}}_t - \mathbf{w}^*\|^2 \leq b \cdot (\alpha_t D + \alpha_t^2 C)$ for some constant $b$ and learning rates $\alpha_t$. Then the one step progress from Lemma 3 becomes:

$$\mathbb{E}\|\overline{\mathbf{w}}_{t+1} - \mathbf{w}^*\|^2 \leq (b(1-\mu\alpha_t) + \alpha_t)\alpha_t D + (b(1-\mu\alpha_t) + \alpha_t)\alpha_t^2 C$$

To establish the result at step $t+1$, it remains to choose $\alpha_t$ and $b$ such that $(b(1-\mu\alpha_t)+\alpha_t)\alpha_t \leq b\alpha_{t+1}$ and $(b(1-\mu\alpha_t)+\alpha_t)\alpha_t^2 \leq b\alpha_{t+1}^2$. If we let $\alpha_t = \frac{4}{\mu(t+\gamma)}$ where $\gamma = \max\{E, 32\kappa\}$ (choice of $\gamma$ required to guarantee the one step progress) and set $b = \frac{4}{\mu}$, we have:

$$(b(1-\mu\alpha_t) + \alpha_t)\alpha_t = \left(b(1 - \frac{4}{t+\gamma}) + \frac{4}{\mu(t+\gamma)}\right)\frac{4}{\mu(t+\gamma)} \leq b\frac{4}{\mu(t+\gamma+1)} = b\alpha_{t+1}$$

$$(b(1-\mu\alpha_t) + \alpha_t)\alpha_t^2 = b(\frac{t+\gamma-2}{t+\gamma})\frac{16}{\mu^2(t+\gamma)^2} \leq b\frac{16}{\mu^2(t+\gamma+1)^2} = b\alpha_{t+1}^2$$

where we have used the following inequalities:

$$\frac{t+\gamma-1}{(t+\gamma)^2} \leq \frac{1}{(t+\gamma+1)} \qquad \frac{t+\gamma-2}{(t+\gamma)^3} \leq \frac{1}{(t+\gamma+1)^2} \qquad \forall\, \gamma \geq 1$$

Thus we have established the result at step $t+1$ assuming the result is correct at step $t$:

$$\mathbb{E}\|\overline{\mathbf{w}}_{t+1} - \mathbf{w}^*\|^2 \leq b \cdot (\alpha_{t+1}D + \alpha_{t+1}^2 C)$$

At step $t = 0$, we can ensure the following inequality by scaling $b$ with $c\|\mathbf{w}_0 - \mathbf{w}^*\|^2$ for a sufficiently large constant $c$:

$$\|\mathbf{w}_0 - \mathbf{w}^*\|^2 \leq b \cdot (\alpha_0 D + \alpha_0^2 C) = b \cdot (\frac{4}{\mu\gamma}D + \frac{16}{\mu^2\gamma^2}C)$$

It follows that

$$\mathbb{E}\|\overline{\mathbf{w}}_t - \mathbf{w}^*\|^2 \leq c\|\mathbf{w}_0 - \mathbf{w}^*\|^2 \frac{4}{\mu}(D\alpha_t + C\alpha_t^2) \tag{8}$$

for all $t \geq 0$.

Finally, the $L$-smoothness of $F$ implies

$$\begin{aligned}
\mathbb{E}(F(\overline{\mathbf{w}}_T)) - F^* &\leq \frac{L}{2}\mathbb{E}\|\overline{\mathbf{w}}_T - \mathbf{w}^*\|^2 \\
&\leq \frac{L}{2}c\|\mathbf{w}_0 - \mathbf{w}^*\|^2 \frac{4}{\mu}(D\alpha_T + C\alpha_T^2) \\
&= 2c\|\mathbf{w}_0 - \mathbf{w}^*\|^2\kappa(D\alpha_T + C\alpha_T^2) \\
&\leq 2c\|\mathbf{w}_0 - \mathbf{w}^*\|^2\kappa\left[\frac{4}{\mu(T+\gamma)} \cdot \frac{1}{N}\nu_{max}^2\sigma^2 + 6E^2LG^2 \cdot (\frac{4}{\mu(T+\gamma)})^2\right] \\
&= \mathcal{O}(\frac{\kappa}{\mu}\frac{1}{N}\nu_{max}^2\sigma^2 \cdot \frac{1}{T} + \frac{\kappa^2}{\mu}E^2G^2 \cdot \frac{1}{T^2})
\end{aligned}$$

where in the first line, we use the property of $L$-smooth function (see Lemma 1), and in the second line, we use the conclusion in Eq (8).

With partial participation, the update at each communication round is now given by weighted averages over a subset of sampled devices. When $t+1 \notin \mathcal{I}_E$, $\overline{\mathbf{v}}_{t+1} = \overline{\mathbf{w}}_{t+1}$, while when $t+1 \in \mathcal{I}_E$, we have $\mathbb{E}\overline{\mathbf{w}}_{t+1} = \overline{\mathbf{v}}_{t+1}$ by design of the sampling schemes (Li et al. (2020b), Lemma 4), so that

$$\begin{aligned}
\mathbb{E}\|\overline{\mathbf{w}}_{t+1} - \mathbf{w}^*\|^2 &= \mathbb{E}\|\overline{\mathbf{w}}_{t+1} - \overline{\mathbf{v}}_{t+1} + \overline{\mathbf{v}}_{t+1} - \mathbf{w}^*\|^2 \\
&= \mathbb{E}\|\overline{\mathbf{w}}_{t+1} - \overline{\mathbf{v}}_{t+1}\|^2 + \mathbb{E}\|\overline{\mathbf{v}}_{t+1} - \mathbf{w}^*\|^2
\end{aligned}$$

This in particular implies $\mathbb{E}\|\overline{\mathbf{v}}_t - \mathbf{w}^*\|^2 \leq \mathbb{E}\|\overline{\mathbf{w}}_t - \mathbf{w}^*\|^2$ for all $t$. Since $\overline{\mathbf{v}}_t = \sum_{k=1}^N p_k \mathbf{v}_t^k$ always averages over all devices, the full participation one step progress result Lemma 3 applied to $\overline{\mathbf{v}}_t$ implies

$$\mathbb{E}\|\overline{\mathbf{v}}_{t+1} - \mathbf{w}^*\|^2 \leq \mathbb{E}(1-\mu\alpha_t)\|\overline{\mathbf{v}}_t - \mathbf{w}^*\|^2 + 6E^2L\alpha_t^3 G^2 + \alpha_t^2\frac{1}{N}\nu_{max}^2\sigma^2$$

$$\leq \mathbb{E}(1 - \mu\alpha_t)\|\overline{\mathbf{w}}_t - \mathbf{w}^*\|^2 + 6E^2 L\alpha_t^3 G^2 + \alpha_t^2 \frac{1}{N}\nu_{max}^2\sigma^2$$

The bound for $\mathbb{E}\|\overline{\mathbf{w}}_{t+1} - \overline{\mathbf{v}}_{t+1}\|^2$ for the two sampling schemes we consider is provided in Eq (7), and applying it we can write the one step progress for partial participation as

$$\mathbb{E}\|\overline{\mathbf{w}}_{t+1} - \mathbf{w}^*\|^2 \leq (1 - \mu\alpha_t)\mathbb{E}\|\overline{\mathbf{w}}_t - \mathbf{w}^*\|^2 + \alpha_t^2 \frac{1}{N}\nu_{max}^2\sigma^2 + \frac{4}{K}\alpha_t^2 E^2 G^2 + 6E^2 L\alpha_t^3 G^2, \quad (9)$$

and the same arguments using induction and $L$-smoothness as the full participation case implies

$$\mathbb{E}F(\overline{\mathbf{w}}_T) - F^* = \mathcal{O}(\frac{\kappa\nu_{\max}^2\sigma^2/\mu}{NT} + \frac{\kappa E^2 G^2/\mu}{KT} + \frac{\kappa^2 E^2 G^2/\mu}{T^2})$$

□

### E.1.1 DEFERRED PROOFS OF KEY LEMMAS

Here we first rewrite the proofs of lemmas 4 and 5 from Li et al. (2020b) with slight modifications for the consistency and completeness of this work, since later we will use modified versions of these results in the convergence proof for Nesterov accelerated FedAvg.

*Proof of lemma 4.*

$$\mathbb{E}\|\mathbf{g}_t - \overline{\mathbf{g}}_t\|^2 = \mathbb{E}\|\mathbf{g}_t - \mathbb{E}\mathbf{g}_t\|^2 = \sum_{k=1}^{N} p_k^2\|\mathbf{g}_{t,k} - \mathbb{E}\mathbf{g}_{t,k}\|^2 \leq \sum_{k=1}^{N} p_k^2\sigma_k^2$$

□

*Proof of lemma 5.* Now we bound $\mathbb{E}\sum_{k=1}^{N} p_k\|\overline{\mathbf{w}}_t - \mathbf{w}_t^k\|^2$ following Li et al. (2020b). Since communication is done every $E$ steps, for any $t \geq 0$, we can find a $t_0 \leq t$ such that $t - t_0 \leq E - 1$ and $\mathbf{w}_{t_0}^k = \overline{\mathbf{w}}_{t_0}$ for all $k$. Moreover, using $\alpha_t$ is non-increasing and $\alpha_{t_0} \leq 2\alpha_t$ for any $t - t_0 \leq E - 1$, we have

$$\mathbb{E}\sum_{k=1}^{N} p_k\|\overline{\mathbf{w}}_t - \mathbf{w}_t^k\|^2$$

$$=\mathbb{E}\sum_{k=1}^{N} p_k\|\mathbf{w}_t^k - \overline{\mathbf{w}}_{t_0} - (\overline{\mathbf{w}}_t - \overline{\mathbf{w}}_{t_0})\|^2$$

$$\leq\mathbb{E}\sum_{k=1}^{N} p_k\|\mathbf{w}_t^k - \overline{\mathbf{w}}_{t_0}\|^2$$

$$=\mathbb{E}\sum_{k=1}^{N} p_k\|\mathbf{w}_t^k - \mathbf{w}_{t_0}^k\|^2$$

$$=\mathbb{E}\sum_{k=1}^{N} p_k\| - \sum_{i=t_0}^{t-1} \alpha_i\mathbf{g}_{i,k}\|^2$$

$$\leq 2\sum_{k=1}^{N} p_k\mathbb{E}\sum_{i=t_0}^{t-1} E\alpha_i^2\|\mathbf{g}_{i,k}\|^2$$

$$\leq 2\sum_{k=1}^{N} p_k E^2\alpha_{t_0}^2 G^2$$

$$\leq 4E^2\alpha_t^2 G^2$$

□

Based on the results of Lemma 4, 5, we now prove the upper bound of one step SGD progress. This proof improves on the previous work Li et al. (2020b) and is the first to reveal the linear speedup of convergence of FedAvg.

*Proof of lemma 3.* We have

$$\|\overline{\mathbf{w}}_{t+1} - \mathbf{w}^*\|^2 = \|(\overline{\mathbf{w}}_t - \alpha_t \mathbf{g}_t) - \mathbf{w}^*\|^2 = \|(\overline{\mathbf{w}}_t - \alpha_t \overline{\mathbf{g}}_t - \mathbf{w}^*) - \alpha_t(\mathbf{g}_t - \overline{\mathbf{g}}_t)\|^2$$
$$= \underbrace{\|\overline{\mathbf{w}}_t - \mathbf{w}^* - \alpha_t \overline{\mathbf{g}}_t\|^2}_{A_1} + \underbrace{2\alpha_t\langle\overline{\mathbf{w}}_t - \mathbf{w}^* - \alpha_t\overline{\mathbf{g}}_t, \overline{\mathbf{g}}_t - \mathbf{g}_t\rangle}_{A_2} + \underbrace{\alpha_t^2\|\mathbf{g}_t - \overline{\mathbf{g}}_t\|^2}_{A_3}$$

where we denote:

$$A_1 = \|\overline{\mathbf{w}}_t - \mathbf{w}^* - \alpha_t\overline{\mathbf{g}}_t\|^2$$
$$A_2 = 2\alpha_t\langle\overline{\mathbf{w}}_t - \mathbf{w}^* - \alpha_t\overline{\mathbf{g}}_t, \overline{\mathbf{g}}_t - \mathbf{g}_t\rangle$$
$$A_3 = \alpha_t^2\|\mathbf{g}_t - \overline{\mathbf{g}}_t\|^2$$

By definition of $\mathbf{g}_t$ and $\overline{\mathbf{g}}_t$ (see Eq (4)), we have $\mathbb{E}A_2 = 0$. For $A_3$, we have the following upper bound (see Lemma 4):

$$\alpha_t^2\mathbb{E}\|\mathbf{g}_t - \overline{\mathbf{g}}_t\|^2 \le \alpha_t^2\sum_{k=1}^{N} p_k^2\sigma_k^2$$

Next we bound $A_1$:

$$\|\overline{\mathbf{w}}_t - \mathbf{w}^* - \alpha_t\overline{\mathbf{g}}_t\|^2 = \|\overline{\mathbf{w}}_t - \mathbf{w}^*\|^2 + 2\langle\overline{\mathbf{w}}_t - \mathbf{w}^*, -\alpha_t\overline{\mathbf{g}}_t\rangle + \|\alpha_t\overline{\mathbf{g}}_t\|^2$$

and we will show that the third term $\|\alpha_t\overline{\mathbf{g}}_t\|^2$ can be canceled by an upper bound of the second term, which is one of major improvement comparing to prior art Li et al. (2020b). The upper bound of second term can be derived as follows, using the strong convexity and $L$-smoothness of $F_k$:

$$-2\alpha_t\langle\overline{\mathbf{w}}_t - \mathbf{w}^*, \overline{\mathbf{g}}_t\rangle$$
$$= -2\alpha_t\sum_{k=1}^{N} p_k\langle\overline{\mathbf{w}}_t - \mathbf{w}^*, \nabla F_k(\mathbf{w}_t^k)\rangle$$
$$= -2\alpha_t\sum_{k=1}^{N} p_k\langle\overline{\mathbf{w}}_t - \mathbf{w}_t^k, \nabla F_k(\mathbf{w}_t^k)\rangle - 2\alpha_t\sum_{k=1}^{N} p_k\langle\mathbf{w}_t^k - \mathbf{w}^*, \nabla F_k(\mathbf{w}_t^k)\rangle$$
$$\le -2\alpha_t\sum_{k=1}^{N} p_k\langle\overline{\mathbf{w}}_t - \mathbf{w}_t^k, \nabla F_k(\mathbf{w}_t^k)\rangle + 2\alpha_t\sum_{k=1}^{N} p_k(F_k(\mathbf{w}^*) - F_k(\mathbf{w}_t^k)) - \alpha_t\mu\sum_{k=1}^{N} p_k\|\mathbf{w}_t^k - \mathbf{w}^*\|^2$$
$$\le 2\alpha_t\sum_{k=1}^{N} p_k\left[F_k(\mathbf{w}_t^k) - F_k(\overline{\mathbf{w}}_t) + \frac{L}{2}\|\overline{\mathbf{w}}_t - \mathbf{w}_t^k\|^2 + F_k(\mathbf{w}^*) - F_k(\mathbf{w}_t^k)\right] - \alpha_t\mu\sum_{k=1}^{N} p_k\|\mathbf{w}_t^k - \mathbf{w}^*\|^2$$
$$= \alpha_t L\sum_{k=1}^{N} p_k\|\overline{\mathbf{w}}_t - \mathbf{w}_t^k\|^2 + 2\alpha_t\sum_{k=1}^{N} p_k[F_k(\mathbf{w}^*) - F_k(\overline{\mathbf{w}}_t)] - \alpha_t\mu\|\overline{\mathbf{w}}_t - \mathbf{w}^*\|^2$$

We record the bound we have obtained so far, as it will also be used in the proof for convex case:

$$\mathbb{E}\|\overline{\mathbf{w}}_{t+1} - \mathbf{w}^*\|^2 \le \mathbb{E}(1 - \mu\alpha_t)\|\overline{\mathbf{w}}_t - \mathbf{w}^*\|^2 + \alpha_t L\sum_{k=1}^{N} p_k\|\overline{\mathbf{w}}_t - \mathbf{w}_t^k\|^2$$
$$+ 2\alpha_t\sum_{k=1}^{N} p_k[F_k(\mathbf{w}^*) - F_k(\overline{\mathbf{w}}_t)] + \alpha_t^2\sum_{k=1}^{N} p_k^2\sigma_k^2 + \alpha_t^2\|\overline{\mathbf{g}}_t\|^2 \quad (10)$$

For the term $2\alpha_t\sum_{k=1}^{N} p_k[F_k(\mathbf{w}^*) - F_k(\overline{\mathbf{w}}_t)]$, which is negative, we can ignore it, but this yields a suboptimal bound that fails to provide the desired linear speedup. Instead, we upper bound it using the following derivation:

$$2\alpha_t\sum_{k=1}^{N} p_k[F_k(\mathbf{w}^*) - F_k(\overline{\mathbf{w}}_t)]$$

$$\leq 2\alpha_t \left[F(\overline{\mathbf{w}}_{t+1}) - F(\overline{\mathbf{w}}_t)\right]$$

$$\leq 2\alpha_t \mathbb{E}\langle \nabla F(\overline{\mathbf{w}}_t), \overline{\mathbf{w}}_{t+1} - \overline{\mathbf{w}}_t \rangle + \alpha_t L \mathbb{E}\|\overline{\mathbf{w}}_{t+1} - \overline{\mathbf{w}}_t\|^2$$

$$= -2\alpha_t^2 \mathbb{E}\langle \nabla F(\overline{\mathbf{w}}_t), \mathbf{g}_t \rangle + \alpha_t^3 L \mathbb{E}\|\mathbf{g}_t\|^2$$

$$= -2\alpha_t^2 \mathbb{E}\langle \nabla F(\overline{\mathbf{w}}_t), \overline{\mathbf{g}}_t \rangle + \alpha_t^3 L \mathbb{E}\|\mathbf{g}_t\|^2$$

$$= -\alpha_t^2 \left[\|\nabla F(\overline{\mathbf{w}}_t)\|^2 + \|\overline{\mathbf{g}}_t\|^2 - \|\nabla F(\overline{\mathbf{w}}_t) - \overline{\mathbf{g}}_t\|^2\right] + \alpha_t^3 L \mathbb{E}\|\mathbf{g}_t\|^2$$

$$= -\alpha_t^2 \left[\|\nabla F(\overline{\mathbf{w}}_t)\|^2 + \|\overline{\mathbf{g}}_t\|^2 - \|\nabla F(\overline{\mathbf{w}}_t) - \sum_k p_k \nabla F(\mathbf{w}_t^k)\|^2\right] + \alpha_t^3 L \mathbb{E}\|\mathbf{g}_t\|^2$$

$$\leq -\alpha_t^2 \left[\|\nabla F(\overline{\mathbf{w}}_t)\|^2 + \|\overline{\mathbf{g}}_t\|^2 - \sum_k p_k \|\nabla F(\overline{\mathbf{w}}_t) - \nabla F(\mathbf{w}_t^k)\|^2\right] + \alpha_t^3 L \mathbb{E}\|\mathbf{g}_t\|^2$$

$$\leq -\alpha_t^2 \left[\|\nabla F(\overline{\mathbf{w}}_t)\|^2 + \|\overline{\mathbf{g}}_t\|^2 - L^2 \sum_k p_k \|\overline{\mathbf{w}}_t - \mathbf{w}_t^k\|^2\right] + \alpha_t^3 L \mathbb{E}\|\mathbf{g}_t\|^2$$

$$\leq -\alpha_t^2 \|\overline{\mathbf{g}}_t\|^2 + \alpha_t^2 L^2 \sum_k p_k \|\overline{\mathbf{w}}_t - \mathbf{w}_t^k\|^2 + \alpha_t^3 L \mathbb{E}\|\mathbf{g}_t\|^2 - \alpha_t^2 \|\nabla F(\overline{\mathbf{w}}_t)\|^2$$

where we have used the smoothness of $F$ twice.

Note that the term $-\alpha_t^2 \|\overline{\mathbf{g}}_t\|^2$ exactly cancels the $\alpha_t^2 \|\overline{\mathbf{g}}_t\|^2$ in the bound in Eq (10), so that plugging in the bound for $-2\alpha_t \langle \overline{\mathbf{w}}_t - \mathbf{w}^*, \overline{\mathbf{g}}_t \rangle$, we have so far proved

$$\mathbb{E}\|\overline{\mathbf{w}}_{t+1} - \mathbf{w}^*\|^2 \leq \mathbb{E}(1 - \mu\alpha_t)\|\overline{\mathbf{w}}_t - \mathbf{w}^*\|^2 + \alpha_t L \sum_{k=1}^N p_k \|\overline{\mathbf{w}}_t - \mathbf{w}_t^k\|^2 + \alpha_t^2 \sum_{k=1}^N p_k^2 \sigma_k^2$$

$$+ \alpha_t^2 L^2 \sum_{k=1}^N p_k \|\overline{\mathbf{w}}_t - \mathbf{w}_t^k\|^2 + \alpha_t^3 L \mathbb{E}\|\mathbf{g}_t\|^2 - \alpha_t^2 \|\nabla F(\overline{\mathbf{w}}_t)\|^2 \qquad (11)$$

Under Assumption 4, we have $\mathbb{E}\|\mathbf{g}_t\|^2 \leq G^2$. Furthermore, we can check that our choice of $\alpha_t$ satisfies $\alpha_t$ is non-increasing and $\alpha_t \leq 2\alpha_{t+E}$, so we may plug in the bound $\mathbb{E}\sum_{k=1}^N p_k \|\overline{\mathbf{w}}_t - \mathbf{w}_t^k\|^2 \leq 4E^2 \alpha_t^2 G^2$ to the above inequality (see Lemma 5).

Therefore, we can conclude that, with $\nu_{max} := N \cdot \max_k p_k$ and $\nu_{min} := N \cdot \min_k p_k$,

$$\mathbb{E}\|\overline{\mathbf{w}}_{t+1} - \mathbf{w}^*\|^2$$

$$\leq \mathbb{E}(1 - \mu\alpha_t)\|\overline{\mathbf{w}}_t - \mathbf{w}^*\|^2 + 4E^2 L\alpha_t^3 G^2 + 4E^2 L^2 \alpha_t^4 G^2 + \alpha_t^2 \sum_{k=1}^N p_k^2 \sigma_k^2 + \alpha_t^3 L G^2$$

$$= \mathbb{E}(1 - \mu\alpha_t)\|\overline{\mathbf{w}}_t - \mathbf{w}^*\|^2 + 4E^2 L\alpha_t^3 G^2 + 4E^2 L^2 \alpha_t^4 G^2 + \alpha_t^2 \frac{1}{N^2} \sum_{k=1}^N (p_k N)^2 \sigma_k^2 + \alpha_t^3 L G^2$$

$$\leq \mathbb{E}(1 - \mu\alpha_t)\|\overline{\mathbf{w}}_t - \mathbf{w}^*\|^2 + 4E^2 L\alpha_t^3 G^2 + 4E^2 L^2 \alpha_t^4 G^2 + \alpha_t^2 \frac{1}{N^2} \nu_{max}^2 \sum_{k=1}^N \sigma_k^2 + \alpha_t^3 L G^2$$

$$\leq \mathbb{E}(1 - \mu\alpha_t)\|\overline{\mathbf{w}}_t - \mathbf{w}^*\|^2 + 6E^2 L\alpha_t^3 G^2 + \alpha_t^2 \frac{1}{N} \nu_{max}^2 \sigma^2$$

where in the last inequality we use $\sigma^2 = \sum_{k=1}^N p_k \sigma_k^2$, and that by construction $\alpha_t$ satisfies $L\alpha_t \leq \frac{1}{8}$.

$\square$

One may ask whether the dependence on $E$ in the term $\frac{\kappa E^2 G^2/\mu}{KT}$ can be removed, or equivalently whether $\sum_k p_k \|\mathbf{w}_t^k - \overline{\mathbf{w}}_t\|^2 = \mathcal{O}(1/T^2)$ can be independent of $E$. We provide a simple counterexample that shows that this is not possible in general.

**Proposition 1.** *There exists a dataset such that if $E = \mathcal{O}(T^\beta)$ for any $\beta > 0$ then $\sum_k p_k \|\mathbf{w}_t^k - \overline{\mathbf{w}}_t\|^2 = \Omega(\frac{1}{T^{2-2\beta}})$.*

*Proof.* Suppose that we have an even number of devices and each $F_k(\mathbf{w}) = \frac{1}{n_k}\sum_{j=1}^{n_k}(\mathbf{x}_k^j - \mathbf{w})^2$ contains data points $\mathbf{x}_k^j = \mathbf{w}^{*,k}$, with $n_k \equiv n$. Moreover, the $\mathbf{w}^{*,k}$'s come in pairs around the origin. As a result, the global objective $F$ is minimized at $\mathbf{w}^* = 0$. Moreover, if we start from $\overline{\mathbf{w}}_0 = 0$, then by design of the dataset the updates in local steps exactly cancel each other at each iteration, resulting in $\overline{\mathbf{w}}_t = 0$ for all $t$. On the other hand, if $E = T^\beta$, then starting from any $t = \mathcal{O}(T)$ with constant step size $\mathcal{O}(\frac{1}{T})$, after $E$ iterations of local steps, the local parameters are updated towards $\mathbf{w}^{*,k}$ with $\|\mathbf{w}_{t+E}^k\|^2 = \Omega((T^\beta \cdot \frac{1}{T})^2) = \Omega(\frac{1}{T^{2-2\beta}})$. This implies that

$$\sum_k p_k \|\mathbf{w}_{t+E}^k - \overline{\mathbf{w}}_{t+E}\|^2 = \sum_k p_k \|\mathbf{w}_{t+E}^k\|^2$$

$$= \Omega(\frac{1}{T^{2-2\beta}})$$

which is at a slower rate than $\frac{1}{T^2}$ for any $\beta > 0$. Thus the sampling variance $\mathbb{E}\|\overline{\mathbf{w}}_{t+1} - \overline{\mathbf{v}}_{t+1}\|^2 = \Omega(\sum_k p_k \mathbb{E}\|\mathbf{w}_{t+1}^k - \overline{\mathbf{w}}_{t+1}\|^2)$ decays at a slower rate than $\frac{1}{T^2}$, resulting in a convergence rate slower than $\mathcal{O}(\frac{1}{T})$ with partial participation. $\qquad\square$

### E.2 Convex Smooth Objectives

In this section we provide the proof of the convergence result for FedAvg with convex and smooth objectives. The key step is a one step progress result analogous to that in the strongly convex case, and their proofs share identical components as well.

**Lemma 6 (One step progress, convex case).** *Let $\overline{\mathbf{w}}_t = \sum_{k=1}^N p_k \mathbf{w}_t^k$ in FedAvg. Under assumptions 1,3,4, the following bound holds for all $t$:*

$$\mathbb{E}\|\overline{\mathbf{w}}_{t+1} - \mathbf{w}^*\|^2 + \alpha_t(F(\overline{\mathbf{w}}_t) - F(\mathbf{w}^*)) \leq \mathbb{E}\|\overline{\mathbf{w}}_t - \mathbf{w}^*\|^2 + \alpha_t^2 \frac{1}{N}\nu_{\max}^2\sigma^2 + 6\alpha_t^3 E^2 L G^2$$

*Proof.* The first part of the proof follows directly from Eq (10) in the proof of Lemma 3. Setting $\mu = 0$ in Eq (10) (since we are in the convex setting instead of strongly convex), we obtain

$$\|\overline{\mathbf{w}}_{t+1} - \mathbf{w}^*\|^2 \leq \|\overline{\mathbf{w}}_t - \mathbf{w}^*\|^2 + \alpha_t L \sum_{k=1}^N p_k \|\overline{\mathbf{w}}_t - \mathbf{w}_t^k\|^2$$

$$+ 2\alpha_t \sum_{k=1}^N p_k [F_k(\mathbf{w}^*) - F_k(\overline{\mathbf{w}}_t)] + \alpha_t^2\|\overline{\mathbf{g}}_t\|^2 + \alpha_t^2 \sum_{k=1}^N p_k^2 \sigma_k^2$$

The difference of this bound with that in the strongly convex case is that we no longer have a contraction factor of $1 - \mu\alpha_t$ in front of $\|\overline{\mathbf{w}}_t - \mathbf{w}^*\|^2$. In the strongly convex case, we were able to cancel $\alpha_t^2\|\overline{\mathbf{g}}_t\|^2$ with $2\alpha_t \sum_{k=1}^N p_k [F_k(\mathbf{w}^*) - F_k(\overline{\mathbf{w}}_t)]$ and obtain only lower order terms. In the convex case, we use a different strategy and preserve $\sum_{k=1}^N p_k [F_k(\mathbf{w}^*) - F_k(\overline{\mathbf{w}}_t)]$ in order to obtain the desired optimality gap.

We have

$$\|\overline{\mathbf{g}}_t\|^2 = \|\sum_k p_k \nabla F_k(\mathbf{w}_t^k)\|^2$$

$$= \|\sum_k p_k \nabla F_k(\mathbf{w}_t^k) - \sum_k p_k \nabla F_k(\overline{\mathbf{w}}_t) + \sum_k p_k \nabla F_k(\overline{\mathbf{w}}_t)\|^2$$

$$\leq 2\|\sum_k p_k \nabla F_k(\mathbf{w}_t^k) - \sum_k p_k \nabla F_k(\overline{\mathbf{w}}_t)\|^2 + 2\|\sum_k p_k \nabla F_k(\overline{\mathbf{w}}_t)\|^2$$

$$\leq 2L^2 \sum_k p_k \|\mathbf{w}_t^k - \overline{\mathbf{w}}_t\|^2 + 2\|\sum_k p_k \nabla F_k(\overline{\mathbf{w}}_t)\|^2$$

$$= 2L^2 \sum_k p_k \|\mathbf{w}_t^k - \overline{\mathbf{w}}_t\|^2 + 2\|\nabla F(\overline{\mathbf{w}}_t)\|^2$$

using $\nabla F(\mathbf{w}^*) = 0$. Now using the $L$ smoothness of $F$, we have $\|\nabla F(\overline{\mathbf{w}}_t)\|^2 \leq 2L(F(\overline{\mathbf{w}}_t) - F(\mathbf{w}^*))$, so that

$$\|\overline{\mathbf{w}}_{t+1} - \mathbf{w}^*\|^2$$

$$\leq \|\overline{\mathbf{w}}_t - \mathbf{w}^*\|^2 + \alpha_t L \sum_{k=1}^N p_k \|\overline{\mathbf{w}}_t - \mathbf{w}_t^k\|^2 + 2\alpha_t \sum_{k=1}^N p_k \left[F_k(\mathbf{w}^*) - F_k(\overline{\mathbf{w}}_t)\right]$$

$$+ 2\alpha_t^2 L^2 \sum_k p_k \|\mathbf{w}_t^k - \overline{\mathbf{w}}_t\|^2 + 4\alpha_t^2 L(F(\overline{\mathbf{w}}_t) - F(\mathbf{w}^*)) + \alpha_t^2 \sum_{k=1}^N p_k^2 \sigma_k^2$$

$$= \|\overline{\mathbf{w}}_t - \mathbf{w}^*\|^2 + (2\alpha_t^2 L^2 + \alpha_t L) \sum_{k=1}^N p_k \|\overline{\mathbf{w}}_t - \mathbf{w}_t^k\|^2 + \alpha_t \sum_{k=1}^N p_k \left[F_k(\mathbf{w}^*) - F_k(\overline{\mathbf{w}}_t)\right]$$

$$+ \alpha_t^2 \sum_{k=1}^N p_k^2 \sigma_k^2 + \alpha_t(1 - 4\alpha_t L)(F(\mathbf{w}^*) - F(\overline{\mathbf{w}}_t))$$

Since $F(\mathbf{w}^*) \leq F(\overline{\mathbf{w}}_t)$, as long as $4\alpha_t L \leq 1$, we can ignore the last term, and rearrange the inequality to obtain

$$\|\overline{\mathbf{w}}_{t+1} - \mathbf{w}^*\|^2 + \alpha_t(F(\overline{\mathbf{w}}_t) - F(\mathbf{w}^*))$$

$$\leq \|\overline{\mathbf{w}}_t - \mathbf{w}^*\|^2 + (2\alpha_t^2 L^2 + \alpha_t L) \sum_{k=1}^N p_k \|\overline{\mathbf{w}}_t - \mathbf{w}_t^k\|^2 + \alpha_t^2 \sum_{k=1}^N p_k^2 \sigma_k^2$$

$$\leq \|\overline{\mathbf{w}}_t - \mathbf{w}^*\|^2 + \frac{3}{2}\alpha_t L \sum_{k=1}^N p_k \|\overline{\mathbf{w}}_t - \mathbf{w}_t^k\|^2 + \alpha_t^2 \sum_{k=1}^N p_k^2 \sigma_k^2$$

The same argument as before yields $\mathbb{E} \sum_{k=1}^N p_k \|\overline{\mathbf{w}}_t - \mathbf{w}_t^k\|^2 \leq 4E^2 \alpha_t^2 G^2$ which gives

$$\|\overline{\mathbf{w}}_{t+1} - \mathbf{w}^*\|^2 + \alpha_t(F(\overline{\mathbf{w}}_t) - F(\mathbf{w}^*)) \leq \|\overline{\mathbf{w}}_t - \mathbf{w}^*\|^2 + \alpha_t^2 \sum_{k=1}^N p_k^2 \sigma_k^2 + 6\alpha_t^3 E^2 L G^2$$

$$\leq \|\overline{\mathbf{w}}_t - \mathbf{w}^*\|^2 + \alpha_t^2 \frac{1}{N} \nu_{\max}^2 \sigma^2 + 6\alpha_t^3 E^2 L G^2$$

$\square$

With the one step progress result, we can now prove the convergence result in the convex setting, which we restate below.

**Theorem 2.** *Under assumptions 1,3,4 and constant learning rate $\alpha_t = \mathcal{O}(\sqrt{\frac{N}{T}})$, FedAvg satisfies*

$$\min_{t \leq T} F(\overline{\mathbf{w}}_t) - F(\mathbf{w}^*) = \mathcal{O}\left(\frac{\nu_{\max}\sigma^2}{\sqrt{NT}} + \frac{NE^2 L G^2}{T}\right)$$

*with full participation, and with partial device participation with $K$ sampled devices at each communication round and learning rate $\alpha_t = \mathcal{O}(\sqrt{\frac{K}{T}})$,*

$$\min_{t \leq T} F(\overline{\mathbf{w}}_t) - F(\mathbf{w}^*) = \mathcal{O}\left(\frac{\nu_{\max}\sigma^2}{\sqrt{KT}} + \frac{E^2 G^2}{\sqrt{KT}} + \frac{KE^2 L G^2}{T}\right)$$

*Proof.* We first prove the bound for full participation. Applying Lemma 6, we have

$$\|\overline{\mathbf{w}}_{t+1} - \mathbf{w}^*\|^2 + \alpha_t(F(\overline{\mathbf{w}}_t) - F(\mathbf{w}^*)) \leq \|\overline{\mathbf{w}}_t - \mathbf{w}^*\|^2 + \alpha_t^2 \frac{1}{N} \nu_{\max}^2 \sigma^2 + 6\alpha_t^3 E^2 L G^2$$

Summing the inequalities from $t = 0$ to $t = T$, we obtain

$$\sum_{t=0}^T \alpha_t(F(\overline{\mathbf{w}}_t) - F(\mathbf{w}^*)) \leq \|\mathbf{w}_0 - \mathbf{w}^*\|^2 + \sum_{t=0}^T \alpha_t^2 \cdot \frac{1}{N} \nu_{\max}^2 \sigma^2 + \sum_{t=0}^T \alpha_t^3 \cdot 6E^2 L G^2$$

so that

$$\min_{t \le T} F(\overline{\mathbf{w}}_t) - F(\mathbf{w}^*) \le \frac{1}{\sum_{t=0}^T \alpha_t} \left( \|\mathbf{w}_0 - \mathbf{w}^*\|^2 + \sum_{t=0}^T \alpha_t^2 \cdot \frac{1}{N} \nu_{\max}^2 \sigma^2 + \sum_{t=0}^T \alpha_t^3 \cdot 6E^2 LG^2 \right)$$

By setting the constant learning rate $\alpha_t \equiv \sqrt{\frac{N}{T}}$, we have

$$
\begin{aligned}
\min_{t \le T} F(\overline{\mathbf{w}}_t) - F(\mathbf{w}^*) &\le \frac{1}{\sqrt{NT}} \cdot \|\mathbf{w}_0 - \mathbf{w}^*\|^2 + \frac{1}{\sqrt{NT}} T \cdot \frac{N}{T} \cdot \frac{1}{N} \nu_{\max}^2 \sigma^2 + \frac{1}{\sqrt{NT}} T (\sqrt{\frac{N}{T}})^3 6E^2 LG^2 \\
&\le \frac{1}{\sqrt{NT}} \cdot \|\mathbf{w}_0 - \mathbf{w}^*\|^2 + \frac{1}{\sqrt{NT}} T \cdot \frac{N}{T} \cdot \frac{1}{N} \nu_{\max}^2 \sigma^2 + \frac{N}{T} 6E^2 LG^2 \\
&= (\|\mathbf{w}_0 - \mathbf{w}^*\|^2 + \nu_{\max}^2 \sigma^2) \frac{1}{\sqrt{NT}} + \frac{N}{T} 6E^2 LG^2 \\
&= \mathcal{O}(\frac{\nu_{\max}^2 \sigma^2}{\sqrt{NT}} + \frac{NE^2 LG^2}{T})
\end{aligned}
$$

For partial participation, the one step progress bound in Lemma 6 is updated in a similar manner as the strongly convex case in (9) to incorporate the sampling variance. More precisely, with partial participation,

$$
\begin{aligned}
\mathbb{E}\|\overline{\mathbf{w}}_{t+1} - \mathbf{w}^*\|^2 &= \mathbb{E}\|\overline{\mathbf{w}}_{t+1} - \overline{\mathbf{v}}_{t+1} + \overline{\mathbf{v}}_{t+1} - \mathbf{w}^*\|^2 \\
&= \mathbb{E}\|\overline{\mathbf{w}}_{t+1} - \overline{\mathbf{v}}_{t+1}\|^2 + \mathbb{E}\|\overline{\mathbf{v}}_{t+1} - \mathbf{w}^*\|^2,
\end{aligned}
$$

where $\mathbb{E}\overline{\mathbf{w}}_{t+1} = \overline{\mathbf{v}}_{t+1}$ for all $t$, by the unbiasedness of our sampling schemes. Since $\overline{\mathbf{v}}_t = \sum_{k=1}^N p_k \mathbf{v}_t^k$ always averages over all devices, the full participation one step progress bound in Lemma 6 applied to $\overline{\mathbf{v}}_t$ implies

$$
\begin{aligned}
\mathbb{E}\|\overline{\mathbf{v}}_{t+1} - \mathbf{w}^*\|^2 + \alpha_t(F(\overline{\mathbf{v}}_t) - F(\mathbf{w}^*)) &\le \mathbb{E}\|\overline{\mathbf{v}}_t - \mathbf{w}^*\|^2 + \alpha_t^2 \frac{1}{N} \nu_{\max}^2 \sigma^2 + 6\alpha_t^3 E^2 LG^2 \\
&\le \mathbb{E}\|\overline{\mathbf{w}}_t - \mathbf{w}^*\|^2 + \alpha_t^2 \frac{1}{N} \nu_{\max}^2 \sigma^2 + 6\alpha_t^3 E^2 LG^2
\end{aligned}
$$

The bound for $\mathbb{E}\|\overline{\mathbf{w}}_{t+1} - \overline{\mathbf{v}}_{t+1}\|^2$ for the two sampling schemes we consider is provided in Eq (7), and applying it to the above bound we can write the one step progress for partial participation as

$$\mathbb{E}\|\overline{\mathbf{w}}_{t+1} - \mathbf{w}^*\|^2 + \alpha_t(F(\overline{\mathbf{w}}_t) - F(\mathbf{w}^*)) \le \mathbb{E}\|\overline{\mathbf{w}}_t - \mathbf{w}^*\|^2 + \alpha_t^2 (\frac{1}{N} \nu_{max}^2 \sigma^2 + C) + 6E^2 L\alpha_t^3 G^2,$$

where $C = \frac{4}{K} E^2 G^2$ or $\frac{N-K}{N-1} \frac{4}{K} E^2 G^2$ depending on the sampling scheme.

Summing up the one-step progress over $t$,

$$\min_{t \le T} F(\overline{\mathbf{w}}_t) - F(\mathbf{w}^*) \le \frac{1}{\sum_{t=0}^T \alpha_t} \left( \|\mathbf{w}_0 - \mathbf{w}^*\|^2 + \sum_{t=0}^T \alpha_t^2 \cdot (\frac{1}{N} \nu_{\max} \sigma^2 + C) + \sum_{t=0}^T \alpha_t^3 \cdot 6E^2 LG^2 \right),$$

so that with $\alpha_t = \sqrt{\frac{K}{T}}$, we have

$$\min_{t \le T} F(\overline{\mathbf{w}}_t) - F(\mathbf{w}^*) = \mathcal{O}(\frac{\nu_{\max} \sigma^2}{\sqrt{KT}} + \frac{E^2 G^2}{\sqrt{KT}} + \frac{KE^2 LG^2}{T}).$$

$\square$

## F  PROOF OF CONVERGENCE RESULTS FOR NESTEROV ACCELERATED FEDAVG

### F.1  STRONGLY CONVEX SMOOTH OBJECTIVES

Recall that the Nesterov accelerated FedAvg follows the updates

$$\mathbf{v}_{t+1}^k = \mathbf{w}_t^k - \alpha_t \mathbf{g}_{t,k}, \quad \mathbf{w}_{t+1}^k = \begin{cases} \mathbf{v}_{t+1}^k + \beta_t(\mathbf{v}_{t+1}^k - \mathbf{v}_t^k) & \text{if } t+1 \notin \mathcal{I}_E, \\ \sum_{k \in \mathcal{S}_{t+1}} q_k \left[ \mathbf{v}_{t+1}^k + \beta_t(\mathbf{v}_{t+1}^k - \mathbf{v}_t^k) \right] & \text{if } t+1 \in \mathcal{I}_E. \end{cases}$$

The proofs of convergence results for Nesterov Accelerated FedAvg consists of components that are direct analogues of the FedAvg case. We first state these analogue results before proving the main theorem. Like before, the proofs of the lemmas are deferred to after the main proof.

**Lemma 7 (One step progress, Nesterov).** *Let $\overline{\mathbf{v}}_t = \sum_{k=1}^N p_k \mathbf{v}_t^k$ in Nesterov accelerated FedAvg, and suppose our functions satisfy Assumptions 1,2,3,4, and set step sizes $\alpha_t = \frac{6}{\mu}\frac{1}{t+\gamma}$, $\beta_{t-1} = \frac{3}{14(t+\gamma)(1-\frac{6}{t+\gamma})\max\{\mu,1\}}$ with $\gamma = \max\{32\kappa, E\}$ and $\kappa = \frac{L}{\mu}$, the updates of Nesterov accelerated FedAvg satisfy*

$$\mathbb{E}\|\overline{\mathbf{v}}_{t+1} - \mathbf{w}^*\|^2 \leq \mathbb{E}(1 - \mu\alpha_t)(1 + \beta_{t-1})^2\|\overline{\mathbf{v}}_t - \mathbf{w}^*\|^2 + 20E^2 L\alpha_t^3 G^2 + (1 - \alpha_t\mu)\beta_{t-1}^2\|(\overline{\mathbf{v}}_{t-1} - \mathbf{w}^*)\|^2$$
$$+ \alpha_t^2 \frac{1}{N}\nu_{\max}\sigma^2 + 2\beta_{t-1}(1 + \beta_{t-1})(1 - \alpha_t\mu)\|\overline{\mathbf{v}}_t - \mathbf{w}^*\| \cdot \|\overline{\mathbf{v}}_{t-1} - \mathbf{w}^*\|$$

The one step progress result makes use of the same bound on the gradient variance in Lemma 4, as well as a divergence bound analogous to Lemma 5, which we state below.

**Lemma 8 (Bounding the divergence of $\mathbf{w}_t^k$, Nesterov).** *Given Assumption 4, and assume that $\alpha_t$ is non-increasing, $\alpha_t \leq 2\alpha_{t+E}$, and $2\beta_{t-1}^2 + 2\alpha_t^2 \leq 1/2$ for all $t \geq 0$, $\overline{\mathbf{w}}_t = \sum_{k=1}^N p_k \mathbf{w}_t^k$ in Nesterov accelerated FedAvg satisfies*

$$\mathbb{E}\left[\sum_{k=1}^N p_k\|\overline{\mathbf{w}}_t - \mathbf{w}_t^k\|^2\right] \leq 16(E-1)^2\alpha_t^2 G^2.$$

**Theorem 3.** *Let $\overline{\mathbf{v}}_T = \sum_{k=1}^N p_k \mathbf{v}_T^k$ in Nesterov accelerated FedAvg and set learning rates $\alpha_t = \frac{6}{\mu}\frac{1}{t+\gamma}$, $\beta_{t-1} = \frac{3}{14(t+\gamma)(1-\frac{6}{t+\gamma})\max\{\mu,1\}}$. Then under Assumptions 1,2,3,4 with full device participation,*

$$\mathbb{E}F(\overline{\mathbf{v}}_T) - F^* = \mathcal{O}\left(\frac{\kappa\nu_{\max}\sigma^2/\mu}{NT} + \frac{\kappa^2 E^2 G^2/\mu}{T^2}\right),$$

*and with partial device participation with $K$ sampled devices at each communication round,*

$$\mathbb{E}F(\overline{\mathbf{v}}_T) - F^* = \mathcal{O}\left(\frac{\kappa\nu_{\max}\sigma^2/\mu}{NT} + \frac{\kappa E^2 G^2/\mu}{KT} + \frac{\kappa^2 E^2 G^2/\mu}{T^2}\right).$$

*Proof.* We first prove the result for full participation. Applying the one step progress bound in Lemma 7, we have

$$\mathbb{E}\|\overline{\mathbf{v}}_{t+1} - \mathbf{w}^*\|^2 \leq \mathbb{E}(1 - \mu\alpha_t)(1 + \beta_{t-1})^2\|\overline{\mathbf{v}}_t - \mathbf{w}^*\|^2 + 20E^2 L\alpha_t^3 G^2 + (1 - \alpha_t\mu)\beta_{t-1}^2\|(\overline{\mathbf{v}}_{t-1} - \mathbf{w}^*)\|^2$$
$$+ \alpha_t^2 \frac{1}{N}\nu_{\max}\sigma^2 + 2\beta_{t-1}(1 + \beta_{t-1})(1 - \alpha_t\mu)\|\overline{\mathbf{v}}_t - \mathbf{w}^*\| \cdot \|\overline{\mathbf{v}}_{t-1} - \mathbf{w}^*\|$$

Recall that we require $\alpha_{t_0} \leq 2\alpha_t$ for any $t - t_0 \leq E - 1$, $L\alpha_t \leq \frac{1}{5}$, and $2\beta_{t-1}^2 + 2\alpha_t^2 \leq 1/2$ in order for Lemmas 8 and 7 to hold, which we can check by definition of $\alpha_t$ and $\beta_t$.

We show next that $\mathbb{E}\|\overline{\mathbf{v}}_t - \mathbf{w}^*\|^2 = \mathcal{O}(\frac{\nu_{max}^2\sigma^2/\mu}{tN} + \frac{E^2 LG^2/\mu^2}{t^2})$ by induction. Assume that we have shown

$$\mathbb{E}\|\overline{\mathbf{v}}_t - \mathbf{w}^*\|^2 \leq b(C\alpha_t^2 + D\alpha_t)$$

for all iterations until $t$, where $C = 20E^2 LG^2$, $D = \frac{1}{N}\nu_{max}^2\sigma^2$, and $b$ is some constant to be chosen later. For step sizes recall that we choose $\alpha_t = \frac{6}{\mu}\frac{1}{t+\gamma}$ and $\beta_{t-1} = \frac{3}{14(t+\gamma)(1-\frac{6}{t+\gamma})\max\{\mu,1\}}$ where $\gamma = \max\{32\kappa, E\}$, so that $\beta_{t-1} \leq \alpha_t$ and

$$(1 - \mu\alpha_t)(1 + 14\beta_{t-1}) \leq (1 - \frac{6}{t+\gamma})(1 + \frac{3}{(t+\gamma)(1-\frac{6}{t+\gamma})})$$
$$= 1 - \frac{6}{t+\gamma} + \frac{3}{t+\gamma} = 1 - \frac{3}{t+\gamma} = 1 - \frac{\mu\alpha_t}{2}$$

Moreover, $\mathbb{E}\|\overline{\mathbf{v}}_{t-1} - \mathbf{w}^*\|^2 \le b(C\alpha_{t-1}^2 + D\alpha_{t-1}) \le 4b(C\alpha_t^2 + D\alpha_t)$ with the chosen step sizes. Therefore the bound for $\mathbb{E}\|\overline{\mathbf{v}}_{t+1} - \mathbf{w}^*\|^2$ can be further simplified with

$$2\beta_{t-1}(1 + \beta_{t-1})(1 - \alpha_t\mu)\mathbb{E}\|\overline{\mathbf{v}}_t - \mathbf{w}^*\| \cdot \|\overline{\mathbf{v}}_{t-1} - \mathbf{w}^*\| \le 4\beta_{t-1}(1 + \beta_{t-1})(1 - \alpha_t\mu) \cdot b(C\alpha_t^2 + D\alpha_t)$$

and

$$(1 - \alpha_t\mu)\beta_{t-1}^2 \mathbb{E}\|(\overline{\mathbf{v}}_{t-1} - \mathbf{w}^*)\|^2 \le 4(1 - \alpha_t\mu)\beta_{t-1}^2 \cdot b(C\alpha_t^2 + D\alpha_t)$$

so that

$$\begin{aligned}
\mathbb{E}\|\overline{\mathbf{v}}_{t+1} - \mathbf{w}^*\|^2 &\le (1 - \mu\alpha_t)((1 + \beta_{t-1})^2 + 4\beta_{t-1}(1 + \beta_{t-1}) + 4\beta_{t-1}^2) \cdot b(C\alpha_t^2 + D\alpha_t) \\
&\quad + 20E^2 L\alpha_t^3 G^2 + \alpha_t^2 \frac{1}{N}\nu_{\max}\sigma^2 \\
&\le \mathbb{E}(1 - \mu\alpha_t)(1 + 14\beta_{t-1}) \cdot b(C\alpha_t^2 + D\alpha_t) + 20E^2 L\alpha_t^3 G^2 + \alpha_t^2 \frac{1}{N}\nu_{\max}\sigma^2 \\
&\le b(1 - \frac{\mu\alpha_t}{2})(C\alpha_t^2 + D\alpha_t) + C\alpha_t^3 + D\alpha_t^2 \\
&= (b(1 - \frac{\mu\alpha_t}{2}) + \alpha_t)\alpha_t^2 C + (b(1 - \frac{\mu\alpha_t}{2}) + \alpha_t)\alpha_t D
\end{aligned}$$

and so it remains to choose $b$ such that

$$(b(1 - \frac{\mu\alpha_t}{2}) + \alpha_t)\alpha_t \le b\alpha_{t+1}$$
$$(b(1 - \frac{\mu\alpha_t}{2}) + \alpha_t)\alpha_t^2 \le b\alpha_{t+1}^2$$

from which we can conclude $\mathbb{E}\|\overline{\mathbf{v}}_{t+1} - \mathbf{w}^*\|^2 \le \alpha_{t+1}^2 C + \alpha_{t+1} D$.

With $b = \frac{6}{\mu}$, we have

$$\begin{aligned}
(b(1 - \frac{\mu\alpha_t}{2}) + \alpha_t)\alpha_t &= (b(1 - (\frac{3}{t + \gamma}) + \frac{6}{\mu(t + \gamma)})\frac{6}{\mu(t + \gamma)} \\
&= (b\frac{t + \gamma - 3}{t + \gamma} + \frac{6}{\mu(t + \gamma)})\frac{6}{\mu(t + \gamma)} \\
&\le b(\frac{t + \gamma - 1}{t + \gamma})\frac{6}{\mu(t + \gamma)} \\
&\le b\frac{6}{\mu(t + \gamma + 1)} = b\alpha_{t+1}
\end{aligned}$$

where we have used $\frac{t + \gamma - 1}{(t + \gamma)^2} \le \frac{1}{t + \gamma + 1}$.

Similarly

$$\begin{aligned}
(b(1 - \frac{\mu\alpha_t}{2}) + \alpha_t)\alpha_t^2 &= (b(1 - (\frac{3}{t + \gamma}) + \frac{6}{\mu(t + \gamma)})(\frac{6}{\mu(t + \gamma)})^2 \\
&= (b\frac{t + \gamma - 3}{t + \gamma} + \frac{6}{\mu(t + \gamma)})(\frac{6}{\mu(t + \gamma)})^2 \\
&= b(\frac{t + \gamma - 2}{t + \gamma})(\frac{6}{\mu(t + \gamma)})^2 \\
&\le b\frac{36}{\mu^2(t + \gamma + 1)^2} = b\alpha_{t+1}^2
\end{aligned}$$

where we have used $\frac{t + \gamma - 2}{(t + \gamma)^3} \le \frac{1}{(t + \gamma + 1)^2}$.

Finally, to ensure $\|\mathbf{v}_0 - \mathbf{w}^*\|^2 \le b(C\alpha_0^2 + D\alpha_0)$, we can rescale $b$ by $c\|\mathbf{v}_0 - \mathbf{w}^*\|^2$ for some $c$. It follows that $\mathbb{E}\|\overline{\mathbf{v}}_t - \mathbf{w}^*\|^2 \le b(C\alpha_t^2 + D\alpha_t)$ for all $t \ge 0$. Using the $L$-smooothness of $F$,

$$\begin{aligned}
\mathbb{E}(F(\overline{\mathbf{v}}_T)) - F^* &= \mathbb{E}(F(\overline{\mathbf{v}}_T) - F(\mathbf{w}^*)) \\
&\le \frac{L}{2}\mathbb{E}\|\overline{\mathbf{v}}_T - \mathbf{w}^*\|^2 \le \frac{L}{2}c\|\mathbf{v}_0 - \mathbf{w}^*\|^2 \frac{6}{\mu}(D\alpha_T + C\alpha_T^2)
\end{aligned}$$

$$= 3c\|\mathbf{v}_0 - \mathbf{w}^*\|^2 \kappa (D\alpha_T + C\alpha_T^2)$$

$$\leq 3c\|\mathbf{v}_0 - \mathbf{w}^*\|^2 \kappa \left[ \frac{6}{\mu(T+\gamma)} \cdot \frac{1}{N} \nu_{\max}\sigma^2 + 20E^2 LG^2 \cdot (\frac{6}{\mu(T+\gamma)})^2 \right]$$

$$= \mathcal{O}(\frac{\kappa}{\mu}\frac{1}{N}\nu_{\max}\sigma^2 \cdot \frac{1}{T} + \frac{\kappa^2}{\mu}E^2 G^2 \cdot \frac{1}{T^2})$$

With partial participation, the same argument with an added term for sampling error yields

$$\mathbb{E}F(\overline{\mathbf{w}}_T) - F^* = \mathcal{O}(\frac{\kappa\nu_{\max}\sigma^2/\mu}{NT} + \frac{\kappa E^2 G^2/\mu}{KT} + \frac{\kappa^2 E^2 G^2/\mu}{T^2})$$

$\square$

### F.1.1 DEFERRED PROOFS OF KEY LEMMAS

*Proof of lemma 8.* The proof of bound for $\mathbb{E}\sum_{k=1}^{N} p_k\|\overline{\mathbf{w}}_t - \mathbf{w}_t^k\|^2$ in the Nesterov accelerated FedAvg follows a similar logic as in Lemma 5, but requires extra reasoning. Since communication is done every $E$ steps, for any $t \geq 0$, we can find a $t_0 \leq t$ such that $t - t_0 \leq E - 1$ and $w_{t_0}^k = \overline{\mathbf{w}}_{t_0}$ for all $k$. Moreover, using $\alpha_t$ is non-increasing, $\alpha_{t_0} \leq 2\alpha_t$, and $\beta_t \leq \alpha_t$ for any $t - t_0 \leq E - 1$, we have

$$\mathbb{E}\sum_{k=1}^{N} p_k\|\overline{\mathbf{w}}_t - \mathbf{w}_t^k\|^2 = \mathbb{E}\sum_{k=1}^{N} p_k\|\mathbf{w}_t^k - \overline{\mathbf{w}}_{t_0} - (\overline{\mathbf{w}}_t - \overline{\mathbf{w}}_{t_0})\|^2$$

$$\leq \mathbb{E}\sum_{k=1}^{N} p_k\|\mathbf{w}_t^k - \overline{\mathbf{w}}_{t_0}\|^2$$

$$= \mathbb{E}\sum_{k=1}^{N} p_k\|\mathbf{w}_t^k - \mathbf{w}_{t_0}^k\|^2$$

$$= \mathbb{E}\sum_{k=1}^{N} p_k\| \sum_{i=t_0}^{t-1} \beta_i(\mathbf{v}_{i+1}^k - \mathbf{v}_i^k) - \sum_{i=t_0}^{t-1} \alpha_i \mathbf{g}_{i,k}\|^2$$

$$\leq 2\sum_{k=1}^{N} p_k\mathbb{E}\sum_{i=t_0}^{t-1}(E-1)\alpha_i^2\|\mathbf{g}_{i,k}\|^2 + 2\sum_{k=1}^{N} p_k\mathbb{E}\sum_{i=t_0}^{t-1}(E-1)\beta_i^2\|(\mathbf{v}_{i+1}^k - \mathbf{v}_i^k)\|^2$$

$$\leq 2\sum_{k=1}^{N} p_k\mathbb{E}\sum_{i=t_0}^{t-1}(E-1)\alpha_i^2(\|\mathbf{g}_{i,k}\|^2 + \|(\mathbf{v}_{i+1}^k - \mathbf{v}_i^k)\|^2)$$

$$\leq 4\sum_{k=1}^{N} p_k\mathbb{E}\sum_{i=t_0}^{t-1}(E-1)\alpha_i^2 G^2$$

$$\leq 4(E-1)^2\alpha_{t_0}^2 G^2 \leq 16(E-1)^2\alpha_t^2 G^2$$

where we have used $\mathbb{E}\|\mathbf{v}_t^k - \mathbf{v}_{t-1}^k\|^2 \leq G^2$. To see this identity for appropriate $\alpha_t, \beta_t$, note the recursion

$$\mathbf{v}_{t+1}^k - \mathbf{v}_t^k = \mathbf{w}_t^k - \mathbf{w}_{t-1}^k - (\alpha_t\mathbf{g}_{t,k} - \alpha_{t-1}\mathbf{g}_{t-1,k})$$
$$\mathbf{w}_{t+1}^k - \mathbf{w}_t^k = -\alpha_t\mathbf{g}_{t,k} + \beta_t(\mathbf{v}_{t+1}^k - \mathbf{v}_t^k)$$

so that

$$\mathbf{v}_{t+1}^k - \mathbf{v}_t^k = -\alpha_{t-1}\mathbf{g}_{t-1,k} + \beta_{t-1}(\mathbf{v}_t^k - \mathbf{v}_{t-1}^k) - (\alpha_t\mathbf{g}_{t,k} - \alpha_{t-1}\mathbf{g}_{t-1,k})$$
$$= \beta_{t-1}(\mathbf{v}_t^k - \mathbf{v}_{t-1}^k) - \alpha_t\mathbf{g}_{t,k}$$

Since the identity $\mathbf{v}_{t+1}^k - \mathbf{v}_t^k = \beta_{t-1}(\mathbf{v}_t^k - \mathbf{v}_{t-1}^k) - \alpha_t\mathbf{g}_{t,k}$ implies

$$\mathbb{E}\|\mathbf{v}_{t+1}^k - \mathbf{v}_t^k\|^2 \leq 2\beta_{t-1}^2\mathbb{E}\|\mathbf{v}_t^k - \mathbf{v}_{t-1}^k\|^2 + 2\alpha_t^2 G^2$$

as long as $\alpha_t, \beta_{t-1}$ satisfy $2\beta_{t-1}^2 + 2\alpha_t^2 \leq 1/2$, we can guarantee that $\mathbb{E}\|\mathbf{v}_t^k - \mathbf{v}_{t-1}^k\|^2 \leq G^2$ for all $k$ by induction. This together with Jensen's inequality also gives $\mathbb{E}\|\overline{\mathbf{v}}_t - \overline{\mathbf{v}}_{t-1}\|^2 \leq G^2$ for all $t$. $\square$

Now we are ready to prove the one step progress result for Nesterov accelerated FedAvg. The first part of the proof is identical to that of the FedAvg case, while the main recursion takes a different form.

*Proof of lemma 7.* We again have

$$\|\overline{\mathbf{v}}_{t+1} - \mathbf{w}^*\|^2 = \|(\overline{\mathbf{w}}_t - \alpha_t \mathbf{g}_t) - \mathbf{w}^*\|^2$$

and using exactly the same derivation as the FedAvg case, we can obtain the following bound (same as Eq (11) in the proof of Lemma 3):

$$\mathbb{E}\|\overline{\mathbf{w}}_{t+1} - \mathbf{w}^*\|^2 \leq \mathbb{E}(1 - \mu\alpha_t)\|\overline{\mathbf{w}}_t - \mathbf{w}^*\|^2 + \alpha_t L \sum_{k=1}^{N} p_k \|\overline{\mathbf{w}}_t - \mathbf{w}_t^k\|^2 + \alpha_t^2 \sum_{k=1}^{N} p_k^2 \sigma_k^2$$

$$+ \alpha_t^2 L^2 \sum_{k=1}^{N} p_k \|\overline{\mathbf{w}}_t - \mathbf{w}_t^k\|^2 + \alpha_t^3 L \mathbb{E}\|\mathbf{g}_t\|^2 - \alpha_t^2 \|\nabla F(\overline{\mathbf{w}}_t)\|^2$$

Different from the FedAvg case, we no longer have $\overline{\mathbf{w}}_t = \overline{\mathbf{v}}_t$. Instead,

$$\|\overline{\mathbf{w}}_t - \mathbf{w}^*\|^2 = \|\overline{\mathbf{v}}_t + \beta_{t-1}(\overline{\mathbf{v}}_t - \overline{\mathbf{v}}_{t-1}) - \mathbf{w}^*\|^2$$
$$= \|(1 + \beta_{t-1})(\overline{\mathbf{v}}_t - \mathbf{w}^*) - \beta_{t-1}(\overline{\mathbf{v}}_{t-1} - \mathbf{w}^*)\|^2$$
$$= (1 + \beta_{t-1})^2 \|\overline{\mathbf{v}}_t - \mathbf{w}^*\|^2 - 2\beta_{t-1}(1 + \beta_{t-1})\langle \overline{\mathbf{v}}_t - \mathbf{w}^*, \overline{\mathbf{v}}_{t-1} - \mathbf{w}^*\rangle + \beta_{t-1}^2 \|(\overline{\mathbf{v}}_{t-1} - \mathbf{w}^*)\|^2$$
$$\leq (1 + \beta_{t-1})^2 \|\overline{\mathbf{v}}_t - \mathbf{w}^*\|^2 + 2\beta_{t-1}(1 + \beta_{t-1})\|\overline{\mathbf{v}}_t - \mathbf{w}^*\| \cdot \|\overline{\mathbf{v}}_{t-1} - \mathbf{w}^*\| + \beta_{t-1}^2 \|(\overline{\mathbf{v}}_{t-1} - \mathbf{w}^*)\|^2$$

which gives a recursion involving both $\overline{\mathbf{v}}_t$ and $\overline{\mathbf{v}}_{t-1}$:

$$\|\overline{\mathbf{v}}_{t+1} - \mathbf{w}^*\|^2 \leq (1 - \alpha_t\mu)(1 + \beta_{t-1})^2 \|\overline{\mathbf{v}}_t - \mathbf{w}^*\|^2 + 2(1 - \alpha_t\mu)\beta_{t-1}(1 + \beta_{t-1})\|\overline{\mathbf{v}}_t - \mathbf{w}^*\| \cdot \|\overline{\mathbf{v}}_{t-1} - \mathbf{w}^*\| + \alpha_t^2 \sum_{k=1}^{N} p_k^2 \sigma_k^2$$

$$+ \beta_{t-1}^2(1 - \alpha_t\mu)\|(\overline{\mathbf{v}}_{t-1} - \mathbf{w}^*)\|^2 + \alpha_t L \sum_{k=1}^{N} p_k \|\overline{\mathbf{w}}_t - \mathbf{w}_t^k\|^2 + \alpha_t^2 L^2 \sum_{k} p_k \|\overline{\mathbf{w}}_t - \mathbf{w}_t^k\|^2 + \alpha_t^3 L G^2$$

and we will using this recursive relation to obtain the desired bound.

We can check that our choice of $\alpha_t$ and $\beta_t$ satisfy $\alpha_t$ is non-increasing, $\alpha_t \leq 2\alpha_{t+E}$, and $2\beta_{t-1}^2 + 2\alpha_t^2 \leq 1/2$ for all $t \geq 0$, so that we can apply the bound from Lemma 8 on $\mathbb{E}\sum_{k=1}^{N} p_k \|\overline{\mathbf{w}}_t - \mathbf{w}_t^k\|^2$ to conclude that, with $\nu_{\max} := N \cdot \max_k p_k$,

$$\mathbb{E}\|\overline{\mathbf{v}}_{t+1} - \mathbf{w}^*\|^2 \leq \mathbb{E}(1 - \mu\alpha_t)(1 + \beta_{t-1})^2 \|\overline{\mathbf{v}}_t - \mathbf{w}^*\|^2 + 16E^2 L\alpha_t^3 G^2 + 16E^2 L^2 \alpha_t^4 G^2 + \alpha_t^3 L G^2$$

$$+ (1 - \alpha_t\mu)\beta_{t-1}^2 \|(\overline{\mathbf{v}}_{t-1} - \mathbf{w}^*)\|^2 + \alpha_t^2 \sum_{k=1}^{N} p_k^2 \sigma_k^2 + 2\beta_{t-1}(1 + \beta_{t-1})(1 - \alpha_t\mu)\|\overline{\mathbf{v}}_t - \mathbf{w}^*\| \cdot \|\overline{\mathbf{v}}_{t-1} - \mathbf{w}^*\|$$

$$\leq \mathbb{E}(1 - \mu\alpha_t)(1 + \beta_{t-1})^2 \|\overline{\mathbf{v}}_t - \mathbf{w}^*\|^2 + 20E^2 L\alpha_t^3 G^2 + (1 - \alpha_t\mu)\beta_{t-1}^2 \|(\overline{\mathbf{v}}_{t-1} - \mathbf{w}^*)\|^2$$

$$+ \alpha_t^2 \frac{1}{N} \nu_{\max}\sigma^2 + 2\beta_{t-1}(1 + \beta_{t-1})(1 - \alpha_t\mu)\|\overline{\mathbf{v}}_t - \mathbf{w}^*\| \cdot \|\overline{\mathbf{v}}_{t-1} - \mathbf{w}^*\|$$

where we have used $\sigma^2 = \sum_k p_k \sigma_k^2$, and by construction our $\alpha_t$ satisfies $L\alpha_t \leq \frac{1}{5}$. $\qquad\square$

## F.2 CONVEX SMOOTH OBJECTIVES

In this section we provide proof of the convergence result for Nesterov accelerated FedAvg with convex and smooth objectives. Unlike with the FedAvg algorithm, where convex and strongly convex results share identical components, the proof for the convergence result in the convex setting for Nesterov FedAvg uses a change of variables, although the general ideas are in the same vein: we have a one step progress bound for $\mathbb{E}\|\overline{\mathbf{w}}_{t+1} - \mathbf{w}^*\|^2 + \eta_t(F(\overline{\mathbf{w}}_t) - F(\mathbf{w}^*))$, which is then used to form a telescoping sum that gives an upper bound on $\min_{t \leq T} F(\overline{\mathbf{w}}_t) - F(\mathbf{w}^*)$.

**Lemma 9** (One step progress, convex case, Nesterov). *Let $\overline{\mathbf{w}}_t = \sum_{k=1}^{N} p_k \mathbf{w}_t^k$ in Nesterov accelerated FedAvg, and define $\eta_t = \frac{\alpha_t}{1-\beta_t}$. Under assumptions 1,3,4, the following bound holds for all $t$:*

$$\mathbb{E}\|\overline{\mathbf{w}}_{t+1} - \mathbf{w}^*\|^2 + \eta_t(F(\overline{\mathbf{w}}_t) - F(\mathbf{w}^*)) \leq \mathbb{E}\|\overline{\mathbf{w}}_t - \mathbf{w}^*\|^2 + 32LE^2\alpha_t^2\eta_t G^2 + \eta_t^2\nu_{\max}\frac{1}{N}\sigma^2 + 2\eta_t\frac{\beta_t^2}{1-\beta_t}G^2$$

**Theorem 4.** *Set learning rates $\alpha_t = \beta_t = \mathcal{O}(\sqrt{\frac{N}{T}})$. Then under Assumptions 1,3,4 Nesterov accelerated FedAvg with full device participation has rate*

$$\min_{t \leq T} F(\overline{\mathbf{w}}_t) - F^* = \mathcal{O}\left(\frac{\nu_{\max}\sigma^2}{\sqrt{NT}} + \frac{NE^2LG^2}{T}\right),$$

*and with partial device participation with $K$ sampled devices at each communication round,*

$$\min_{t \leq T} F(\overline{\mathbf{w}}_t) - F^* = \mathcal{O}\left(\frac{\nu_{\max}\sigma^2}{\sqrt{KT}} + \frac{E^2G^2}{\sqrt{KT}} + \frac{KE^2LG^2}{T}\right).$$

*Proof.* Applying the bound from Lemma 9, with $\eta_t = \frac{\alpha_t}{1-\beta_t}$ we have

$$\mathbb{E}\|\overline{\mathbf{w}}_{t+1} - \mathbf{w}^*\|^2 + \eta_t(F(\overline{\mathbf{w}}_t) - F(\mathbf{w}^*)) \leq \mathbb{E}\|\overline{\mathbf{w}}_t - \mathbf{w}^*\|^2 + 32LE^2\alpha_t^2\eta_t G^2 + \eta_t^2\nu_{\max}\frac{1}{N}\sigma^2 + 2\eta_t\frac{\beta_t^2}{1-\beta_t}G^2$$

Summing the inequalities from $t = 0$ to $t = T$, we obtain

$$\sum_{t=0}^{T} \eta_t(F(\overline{\mathbf{w}}_t) - F(\mathbf{w}^*)) \leq \|\mathbf{w}_0 - \mathbf{w}^*\|^2 + \sum_{t=0}^{T} \eta_t^2 \cdot \frac{1}{N}\nu_{\max}\sigma^2 + \sum_{t=0}^{T} \eta_t\alpha_t^2 \cdot 32LE^2G^2 + \sum_{t=0}^{T} 2\eta_t\frac{\beta_t^2}{1-\beta_t}G^2$$

so that

$$\min_{t \leq T} F(\overline{\mathbf{w}}_t) - F(\mathbf{w}^*) \leq \frac{1}{\sum_{t=0}^{T}\eta_t}\left(\|\mathbf{w}_0 - \mathbf{w}^*\|^2 + \sum_{t=0}^{T} \eta_t^2 \cdot \frac{1}{N}\nu_{\max}\sigma^2 + \sum_{t=0}^{T} \eta_t\alpha_t^2 \cdot 32LE^2G^2 + \sum_{t=0}^{T} 2\eta_t\frac{\beta_t^2}{1-\beta_t}G^2\right)$$

By setting the constant learning rates $\alpha_t \equiv \sqrt{\frac{N}{T}}$ and $\beta_t \equiv c\sqrt{\frac{N}{T}}$ so that $\eta_t = \frac{\alpha_t}{1-\beta_t} = \frac{\sqrt{\frac{N}{T}}}{1-c\sqrt{\frac{N}{T}}} \leq 2\sqrt{\frac{N}{T}}$, we have

$$\min_{t \leq T} F(\overline{\mathbf{w}}_t) - F(\mathbf{w}^*)$$

$$\leq \frac{1}{2\sqrt{NT}} \cdot \|\mathbf{w}_0 - \mathbf{w}^*\|^2 + \frac{2}{\sqrt{NT}}T \cdot \frac{N}{T} \cdot \frac{1}{N}\nu_{\max}\sigma^2 + \frac{1}{\sqrt{NT}}T(\sqrt{\frac{N}{T}})^3 32LE^2G^2 + \frac{2}{\sqrt{NT}}T(\sqrt{\frac{N}{T}})^3 G^2$$

$$= (\frac{1}{2}\|\mathbf{w}_0 - \mathbf{w}^*\|^2 + 2\nu_{\max}\sigma^2)\frac{1}{\sqrt{NT}} + \frac{N}{T}(32LE^2G^2 + 2G^2)$$

$$= O(\frac{\nu_{\max}\sigma^2}{\sqrt{NT}} + \frac{NE^2LG^2}{T})$$

Similarly, for partial participation, we have

$$\min_{t \leq T} F(\overline{\mathbf{w}}_t) - F(\mathbf{w}^*) \leq \frac{1}{\sum_{t=0}^{T}\alpha_t}\left(\|\mathbf{w}_0 - \mathbf{w}^*\|^2 + \sum_{t=0}^{T} \alpha_t^2 \cdot (\frac{1}{N}\nu_{\max}\sigma^2 + C) + \sum_{t=0}^{T} \alpha_t^3 \cdot 6E^2LG^2\right)$$

where $C = \frac{4}{K}E^2G^2$ or $\frac{N-K}{N-1}\frac{4}{K}E^2G^2$, so that with $\alpha_t \equiv \sqrt{\frac{K}{T}}$ and $\beta_t \equiv c\sqrt{\frac{K}{T}}$, we have

$$\min_{t \leq T} F(\overline{\mathbf{w}}_t) - F(\mathbf{w}^*) = \mathcal{O}(\frac{\nu_{\max}\sigma^2}{\sqrt{KT}} + \frac{E^2G^2}{\sqrt{KT}} + \frac{KE^2LG^2}{T})$$

$\square$

### F.2.1 DEFERRED PROOFS OF KEY LEMMAS

*Proof of lemma 9.* Define $\overline{\mathbf{p}}_t := \frac{\beta_t}{1-\beta_t}[\overline{\mathbf{w}}_t - \overline{\mathbf{w}}_{t-1} + \alpha_t\mathbf{g}_{t-1}] = \frac{\beta_t^2}{1-\beta_t}(\overline{\mathbf{v}}_t - \overline{\mathbf{v}}_{t-1})$ for $t \geq 1$ and $0$ for $t = 0$. We can check that

$$\overline{\mathbf{w}}_{t+1} + \overline{\mathbf{p}}_{t+1} = \overline{\mathbf{w}}_t + \overline{\mathbf{p}}_t - \frac{\alpha_t}{1-\beta_t}\mathbf{g}_t$$

Now we define $\overline{\mathbf{z}}_t := \overline{\mathbf{w}}_t + \overline{\mathbf{p}}_t$ and $\eta_t = \frac{\alpha_t}{1-\beta_t}$ for all $t$, so that we have the recursive relation

$$\overline{\mathbf{z}}_{t+1} = \overline{\mathbf{z}}_t - \eta_t\mathbf{g}_t$$

Now

$$\begin{aligned}
\|\overline{\mathbf{z}}_{t+1} - \mathbf{w}^*\|^2 &= \|(\overline{\mathbf{z}}_t - \eta_t\mathbf{g}_t) - \mathbf{w}^*\|^2 \\
&= \|(\overline{\mathbf{z}}_t - \eta_t\overline{\mathbf{g}}_t - \mathbf{w}^*) - \eta_t(\mathbf{g}_t - \overline{\mathbf{g}}_t)\|^2 \\
&= A_1 + A_2 + A_3
\end{aligned}$$

where

$$\begin{aligned}
A_1 &= \|\overline{\mathbf{z}}_t - \mathbf{w}^* - \eta_t\overline{\mathbf{g}}_t\|^2 \\
A_2 &= 2\eta_t\langle\overline{\mathbf{z}}_t - \mathbf{w}^* - \eta_t\overline{\mathbf{g}}_t, \overline{\mathbf{g}}_t - \mathbf{g}_t\rangle \\
A_3 &= \eta_t^2\|\mathbf{g}_t - \overline{\mathbf{g}}_t\|^2
\end{aligned}$$

where again $\mathbb{E}A_2 = 0$ and $\mathbb{E}A_3 \leq \eta_t^2\sum_k p_k^2\sigma_k^2$. For $A_1$ we have

$$\|\overline{\mathbf{z}}_t - \mathbf{w}^* - \eta_t\overline{\mathbf{g}}_t\|^2 = \|\overline{\mathbf{z}}_t - \mathbf{w}^*\|^2 + 2\langle\overline{\mathbf{z}}_t - \mathbf{w}^*, -\eta_t\overline{\mathbf{g}}_t\rangle + \|\eta_t\overline{\mathbf{g}}_t\|^2$$

Using the convexity and $L$-smoothness of $F_k$,

$$\begin{aligned}
&-2\eta_t\langle\overline{\mathbf{z}}_t - \mathbf{w}^*, \overline{\mathbf{g}}_t\rangle \\
&= -2\eta_t\sum_{k=1}^{N} p_k\langle\overline{\mathbf{z}}_t - \mathbf{w}^*, \nabla F_k(\mathbf{w}_t^k)\rangle \\
&= -2\eta_t\sum_{k=1}^{N} p_k\langle\overline{\mathbf{z}}_t - \mathbf{w}_t^k, \nabla F_k(\mathbf{w}_t^k)\rangle - 2\eta_t\sum_{k=1}^{N} p_k\langle\mathbf{w}_t^k - \mathbf{w}^*, \nabla F_k(\mathbf{w}_t^k)\rangle \\
&= -2\eta_t\sum_{k=1}^{N} p_k\langle\overline{\mathbf{z}}_t - \overline{\mathbf{w}}_t, \nabla F_k(\mathbf{w}_t^k)\rangle - 2\eta_t\sum_{k=1}^{N} p_k\langle\overline{\mathbf{w}}_t - \mathbf{w}_t^k, \nabla F_k(\mathbf{w}_t^k)\rangle - 2\eta_t\sum_{k=1}^{N} p_k\langle\mathbf{w}_t^k - \mathbf{w}^*, \nabla F_k(\mathbf{w}_t^k)\rangle \\
&\leq -2\eta_t\sum_{k=1}^{N} p_k\langle\overline{\mathbf{z}}_t - \overline{\mathbf{w}}_t, \nabla F_k(\mathbf{w}_t^k)\rangle - 2\eta_t\sum_{k=1}^{N} p_k\langle\overline{\mathbf{w}}_t - \mathbf{w}_t^k, \nabla F_k(\mathbf{w}_t^k)\rangle + 2\eta_t\sum_{k=1}^{N} p_k(F_k(\mathbf{w}^*) - F_k(\mathbf{w}_t^k)) \\
&\leq 2\eta_t\sum_{k=1}^{N} p_k\left[F_k(\mathbf{w}_t^k) - F_k(\overline{\mathbf{w}}_t) + \frac{L}{2}\|\overline{\mathbf{w}}_t - \mathbf{w}_t^k\|^2 + F_k(\mathbf{w}^*) - F_k(\mathbf{w}_t^k)\right] \\
&\quad - 2\eta_t\sum_{k=1}^{N} p_k\langle\overline{\mathbf{z}}_t - \overline{\mathbf{w}}_t, \nabla F_k(\mathbf{w}_t^k)\rangle \\
&= \eta_t L\sum_{k=1}^{N} p_k\|\overline{\mathbf{w}}_t - \mathbf{w}_t^k\|^2 + 2\eta_t\sum_{k=1}^{N} p_k[F_k(\mathbf{w}^*) - F_k(\overline{\mathbf{w}}_t)] - 2\eta_t\sum_{k=1}^{N} p_k\langle\overline{\mathbf{z}}_t - \overline{\mathbf{w}}_t, \nabla F_k(\mathbf{w}_t^k)\rangle
\end{aligned}$$

which results in

$$\begin{aligned}
\mathbb{E}\|\overline{\mathbf{w}}_{t+1} - \mathbf{w}^*\|^2 &\leq \mathbb{E}\|\overline{\mathbf{w}}_t - \mathbf{w}^*\|^2 + \eta_t L\sum_{k=1}^{N} p_k\|\overline{\mathbf{w}}_t - \mathbf{w}_t^k\|^2 + 2\eta_t\sum_{k=1}^{N} p_k[F_k(\mathbf{w}^*) - F_k(\overline{\mathbf{w}}_t)] \\
&\quad + \eta_t^2\|\overline{\mathbf{g}}_t\|^2 + \eta_t^2\sum_{k=1}^{N} p_k^2\sigma_k^2 - 2\eta_t\sum_{k=1}^{N} p_k\langle\overline{\mathbf{z}}_t - \overline{\mathbf{w}}_t, \nabla F_k(\mathbf{w}_t^k)\rangle
\end{aligned}$$

As before, $\|\overline{\mathbf{g}}_t\|^2 \leq 2L^2 \sum_k p_k \|\mathbf{w}_t^k - \overline{\mathbf{w}}_t\|^2 + 4L(F(\overline{\mathbf{w}}_t) - F(\mathbf{w}^*))$, so that

$$\eta_t^2\|\overline{\mathbf{g}}_t\|^2 + \eta_t \sum_{k=1}^N p_k \left[F_k(\mathbf{w}^*) - F_k(\overline{\mathbf{w}}_t)\right] \leq 2L^2\eta_t^2 \sum_k p_k\|\mathbf{w}_t^k - \overline{\mathbf{w}}_t\|^2 + \eta_t(1 - 4\eta_t L)(F(\mathbf{w}^*) - F(\overline{\mathbf{w}}_t))$$

$$\leq 2L^2\eta_t^2 \sum_k p_k\|\mathbf{w}_t^k - \overline{\mathbf{w}}_t\|^2$$

for $\eta_t \leq 1/4L$. Using $\sum_{k=1}^N p_k\|\overline{\mathbf{w}}_t - \mathbf{w}_t^k\|^2 \leq 16E^2\alpha_t^2 G^2$ and $\sum_{k=1}^N p_k^2\sigma_k^2 \leq \nu_{\max}\frac{1}{N}\sigma^2$, it follows that

$$\mathbb{E}\|\overline{\mathbf{w}}_{t+1} - \mathbf{w}^*\|^2 + \eta_t(F(\overline{\mathbf{w}}_t) - F(\mathbf{w}^*)) \leq \mathbb{E}\|\overline{\mathbf{w}}_t - \mathbf{w}^*\|^2 + (\eta_t L + 2L^2\eta_t^2)\sum_{k=1}^N p_k\|\overline{\mathbf{w}}_t - \mathbf{w}_t^k\|^2 + \eta_t^2 \sum_{k=1}^N p_k^2\sigma_k^2$$

$$- 2\eta_t \sum_{k=1}^N p_k\langle \overline{\mathbf{z}}_t - \overline{\mathbf{w}}_t, \nabla F_k(\mathbf{w}_t^k)\rangle$$

$$\leq \mathbb{E}\|\overline{\mathbf{w}}_t - \mathbf{w}^*\|^2 + 32LE^2\alpha_t^2\eta_t G^2 + \eta_t^2\nu_{\max}\frac{1}{N}\sigma^2$$

$$- 2\eta_t \sum_{k=1}^N p_k\langle \overline{\mathbf{z}}_t - \overline{\mathbf{w}}_t, \nabla F_k(\mathbf{w}_t^k)\rangle$$

if $\eta_t \leq \frac{1}{2L}$. It remains to bound $\mathbb{E}\sum_{k=1}^N p_k\langle \overline{\mathbf{z}}_t - \overline{\mathbf{w}}_t, \nabla F_k(\mathbf{w}_t^k)\rangle$. Recall that $\overline{\mathbf{z}}_t - \overline{\mathbf{w}}_t = \frac{\beta_t}{1-\beta_t}\left[\overline{\mathbf{w}}_t - \overline{\mathbf{w}}_{t-1} + \alpha_t\mathbf{g}_{t-1}\right] = \frac{\beta_t^2}{1-\beta_t}(\overline{\mathbf{v}}_t - \overline{\mathbf{v}}_{t-1})$ and $\mathbb{E}\|\overline{\mathbf{v}}_t - \overline{\mathbf{v}}_{t-1}\|^2 \leq G^2$, $\mathbb{E}\|\nabla F_k(\mathbf{w}_t^k)\|^2 \leq G^2$. Cauchy-Schwarz gives

$$\mathbb{E}\sum_{k=1}^N p_k\langle \overline{\mathbf{z}}_t - \overline{\mathbf{w}}_t, \nabla F_k(\mathbf{w}_t^k)\rangle \leq \sum_{k=1}^N p_k\sqrt{\mathbb{E}\|\overline{\mathbf{z}}_t - \overline{\mathbf{w}}_t\|^2} \cdot \sqrt{\mathbb{E}\|\nabla F_k(\mathbf{w}_t^k)\|^2}$$

$$\leq \frac{\beta_t^2}{1-\beta_t}G^2$$

Thus

$$\mathbb{E}\|\overline{\mathbf{w}}_{t+1} - \mathbf{w}^*\|^2 + \eta_t(F(\overline{\mathbf{w}}_t) - F(\mathbf{w}^*)) \leq \mathbb{E}\|\overline{\mathbf{w}}_t - \mathbf{w}^*\|^2 + 32LE^2\alpha_t^2\eta_t G^2 + \eta_t^2\nu_{\max}\frac{1}{N}\sigma^2 + 2\eta_t\frac{\beta_t^2}{1-\beta_t}G^2$$

$$\square$$

## G   GEOMETRIC CONVERGENCE OF FEDAVG IN THE OVERPARAMETERIZED SETTING

Overparameterization is a prevalent machine learning setting where the statistical model has much more parameters than the number of training samples and the existence of parameter choices with zero training loss is ensured Allen-Zhu et al. (2018); Zhang et al. (2016). Due to the property of *automatic variance reduction* in overparameterization, a line of recent works proved that SGD and accelerated methods achieve geometric convergence Ma et al. (2018); Moulines & Bach (2011); Needell et al. (2014); Schmidt & Roux (2013); Strohmer & Vershynin (2009). A natural question is whether such a result still holds in the federated learning setting. In this section, we provide the first geometric convergence rate of FedAvg for the overparameterized strongly convex and smooth problems, and show that it preserves linear speedup at the same time. We then sharpen this result in the special case of linear regression. Inspired by recent advances in accelerating SGD Liu et al. (2020); Jain et al. (2017), we further propose a novel momentum-based FedAvg algorithm, which enjoys an improved convergence rate over FedAvg. Detailed proofs are deferred to Appendix Section H. In particular, we do not need Assumptions 3 and 4 and use modified versions of Assumptions 1 and 2 detailed in this section.

## G.1 GEOMETRIC CONVERGENCE OF FEDAVG IN THE OVERPARAMETERIZED SETTING

Recall the FL problem $\min_w \sum_{k=1}^{N} p_k F_k(\mathbf{w})$ with $F_k(\mathbf{w}) = \frac{1}{n_k} \sum_{j=1}^{n_k} \ell(\mathbf{w}; \mathbf{x}_k^j)$. In this section, we consider the standard Empirical Risk Minimization (ERM) setting where $\ell$ is non-negative, $l$-smooth, and convex, and as before, each $F_k(\mathbf{w})$ is $L$-smooth and $\mu$-strongly convex. Note that $l \geq L$. This setup includes many important problems in practice. In the overparameterized setting, there exists $\mathbf{w}^* \in \arg\min_w \sum_{k=1}^{N} p_k F_k(\mathbf{w})$ such that $\ell(\mathbf{w}^*; \mathbf{x}_k^j) = 0$ for all $\mathbf{x}_k^j$. We first show that FedAvg achieves geometric convergence with linear speedup in the number of workers.

**Theorem 5.** *In the overparameterized setting, FedAvg with communication every $E$ iterations and constant step size $\overline{\alpha} = \mathcal{O}(\frac{1}{E} \frac{N}{l\nu_{\max} + L(N - \nu_{\min})})$ has geometric convergence:*

$$\mathbb{E}F(\overline{\mathbf{w}}_T) \leq \frac{L}{2}(1 - \overline{\alpha})^T \|\mathbf{w}_0 - \mathbf{w}^*\|^2 = \mathcal{O}\left(L \exp\left(-\frac{\mu}{E} \frac{NT}{l\nu_{\max} + L(N - \nu_{\min})}\right) \cdot \|\mathbf{w}_0 - \mathbf{w}^*\|^2\right).$$

**Linear speedup and Communication Complexity** The linear speedup factor is on the order of $\mathcal{O}(N/E)$ for $N \leq \mathcal{O}(\frac{l}{L})$, i.e. FedAvg with $N$ workers and communication every $E$ iterations provides a geometric convergence speedup factor of $\mathcal{O}(N/E)$, for $N \leq \mathcal{O}(\frac{l}{L})$. When $N$ is above this threshold, however, the speedup is almost constant in the number of workers. This matches the findings in Ma et al. (2018). Our result also illustrates that $E$ can be taken $\mathcal{O}(T^\beta)$ for any $\beta < 1$ to achieve geometric convergence, achieving better communication efficiency than the standard FL setting. We emphasize again that compared to the single-server results in Ma et al. (2018), the difference of our result lies in the factor of $N$ in the speedup, which cannot be obtained if one simply applied the single-server result to each device in our problem.

## G.2 OVERPARAMETERIZED LINEAR REGRESSION PROBLEMS

We now turn to quadratic problems and show that the bound in Theorem 5 can be improved to $\mathcal{O}(\exp(-\frac{N}{E\kappa_1}t))$ for a larger range of $N$. We then propose a variant of FedAvg that has provable acceleration over FedAvg with SGD updates. The local device objectives are now given by the sum of squares $F_k(\mathbf{w}) = \frac{1}{2n_k} \sum_{j=1}^{n_k} (\mathbf{w}^T \mathbf{x}_k^j - z_k^j)^2$, and there exists $\mathbf{w}^*$ such that $F(\mathbf{w}^*) \equiv 0$. Two notions of condition number are important in our results: $\kappa_1$ which is based on local Hessians, and $\tilde{\kappa}$, which is termed the statistical condition number Liu & Belkin (2020); Jain et al. (2017). For their detailed definitions, please refer to Appendix Section H. Here we use the fact $\tilde{\kappa} \leq \kappa_1$. Recall $\nu_{\max} = \max_k p_k N$ and $\nu_{\min} = \min_k p_k N$.

**Theorem 6.** *For the overparamterized linear regression problem, FedAvg with communication every $E$ iterations with constant step size $\overline{\alpha} = \mathcal{O}(\frac{1}{E} \frac{N}{l\nu_{\max} + \mu(N - \nu_{\min})})$ has geometric convergence:*

$$\mathbb{E}F(\overline{\mathbf{w}}_T) \leq \mathcal{O}\left(L \exp(-\frac{NT}{E(\nu_{\max}\kappa_1 + (N - \nu_{\min}))})\|\mathbf{w}_0 - \mathbf{w}^*\|^2\right).$$

When $N = \mathcal{O}(\kappa_1)$, the convergence rate is $\mathcal{O}((1 - \frac{N}{E\kappa_1})^T) = \mathcal{O}(\exp(-\frac{NT}{E\kappa_1}))$, which exhibits linear speedup in the number of workers, as well as a $1/\kappa_1$ dependence on the condition number $\kappa_1$. Inspired by Liu & Belkin (2020), we propose the **MaSS accelerated FedAvg algorithm** (FedMaSS):

$$\mathbf{w}_{t+1}^k = \begin{cases} \mathbf{u}_t^k - \eta_1^k \mathbf{g}_{t,k} & \text{if } t + 1 \notin \mathcal{I}_E, \\ \sum_{k \in \mathcal{S}_{t+1}} [\mathbf{u}_t^k - \eta_1^k \mathbf{g}_{t,k}] & \text{if } t + 1 \in \mathcal{I}_E, \end{cases}$$

$$\mathbf{u}_{t+1}^k = \mathbf{w}_{t+1}^k + \gamma^k(\mathbf{w}_{t+1}^k - \mathbf{w}_t^k) + \eta_2^k \mathbf{g}_{t,k}.$$

When $\eta_2^k \equiv 0$, this algorithm reduces to the Nesterov accelerated FedAvg algorithm. In the next theorem, we demonstrate that FedMaSS improves the convergence to $\mathcal{O}(\exp(-\frac{NT}{E\sqrt{\kappa_1 \tilde{\kappa}}}))$. To our knowledge, this is the first acceleration result of FedAvg with momentum updates over SGD updates.

**Theorem 7.** *For the overparamterized linear regression problem, FedMaSS with communication every $E$ iterations and constant step sizes $\overline{\eta}_1 = \mathcal{O}(\frac{1}{E} \frac{N}{l\nu_{\max} + \mu(N - \nu_{\min})}), \overline{\eta}_2 = \frac{\overline{\eta}_1(1 - \frac{1}{\tilde{\kappa}})}{1 + \frac{1}{\sqrt{\kappa_1 \tilde{\kappa}}}}, \overline{\gamma} = \frac{1 - \frac{1}{\sqrt{\kappa_1 \tilde{\kappa}}}}{1 + \frac{1}{\sqrt{\kappa_1 \tilde{\kappa}}}}$*

*has geometric convergence:*

$$\mathbb{E}F(\overline{\mathbf{w}}_T) \leq \mathcal{O}\left(L \exp(-\frac{NT}{E(\nu_{\max}\sqrt{\kappa_1 \tilde{\kappa}} + (N - \nu_{\min}))})\|\mathbf{w}_0 - \mathbf{w}^*\|^2\right).$$

**Speedup of FedMaSS over FedAvg** To better understand the significance of the above result, we briefly discuss related works on accelerating SGD. Nesterov and Heavy Ball updates are known to fail to accelerate over SGD in both the overparameterized and convex settings Liu & Belkin (2020); Kidambi et al. (2018); Liu et al. (2018); Yuan et al. (2016). Thus in general one cannot hope to obtain acceleration results for the FedAvg algorithm with Nesterov and Heavy Ball updates. Luckily, recent works in SGD Jain et al. (2017); Liu & Belkin (2020) introduced an additional compensation term to the Nesterov updates to address the non-acceleration issue. Surprisingly, we show the same approach can effectively improve the rate of FedAvg. Comparing the convergence rate of FedMass (Theorem 7) and FedAvg (Theorem 6), when $N = \mathcal{O}(\sqrt{\kappa_1 \tilde{\kappa}})$, the convergence rate is $\mathcal{O}((1 - \frac{N}{E\sqrt{\kappa_1 \tilde{\kappa}}})^T) = \mathcal{O}(\exp(-\frac{NT}{E\sqrt{\kappa_1 \tilde{\kappa}}}))$ as opposed to $\mathcal{O}(\exp(-\frac{NT}{E\kappa_1}))$. Since $\kappa_1 \geq \tilde{\kappa}$, this implies a speedup factor of $\sqrt{\frac{\kappa_1}{\tilde{\kappa}}}$ for FedMaSS. On the other hand, the same linear speedup in the number of workers holds for $N$ in a smaller range of values.

## H  PROOF OF GEOMETRIC CONVERGENCE RESULTS FOR OVERPARAMETERIZED PROBLEMS

### H.1  GEOMETRIC CONVERGENCE OF FEDAVG FOR GENERAL STRONGLY CONVEX AND SMOOTH OBJECTIVES

**Theorem 5.** *For the overparameterized setting with general strongly convex and smooth objectives, FedAvg with local SGD updates and communication every $E$ iterations with constant step size $\overline{\alpha} = \frac{1}{2E}\frac{N}{l\nu_{\max} + L(N - \nu_{\min})}$ gives the exponential convergence guarantee*

$$\mathbb{E}F(\overline{\mathbf{w}}_t) \leq \frac{L}{2}(1 - \mu\overline{\alpha})^t \|\mathbf{w}_0 - \mathbf{w}^*\|^2 = O(\exp(-\frac{\mu}{2E}\frac{N}{l\nu_{\max} + L(N - \nu_{\min})}t) \cdot \|\mathbf{w}_0 - \mathbf{w}^*\|^2)$$

*Proof.* To illustrate the main ideas of the proof, we first present the proof for $E = 2$. Let $t - 1$ be a communication round, so that $\mathbf{w}_{t-1}^k = \overline{\mathbf{w}}_{t-1}$. We show that

$$\|\overline{\mathbf{w}}_{t+1} - \mathbf{w}^*\|^2 \leq (1 - \alpha_t\mu)(1 - \alpha_{t-1}\mu)\|\overline{\mathbf{w}}_{t-1} - \mathbf{w}^*\|^2$$

for appropriately chosen constant step sizes $\alpha_t, \alpha_{t-1}$. We have

$$
\begin{aligned}
\|\overline{\mathbf{w}}_{t+1} - \mathbf{w}^*\|^2 &= \|(\overline{\mathbf{w}}_t - \alpha_t\mathbf{g}_t) - \mathbf{w}^*\|^2 \\
&= \|\overline{\mathbf{w}}_t - \mathbf{w}^*\|^2 - 2\alpha_t\langle\overline{\mathbf{w}}_t - \mathbf{w}^*, \mathbf{g}_t\rangle + \alpha_t^2\|\mathbf{g}_t\|^2
\end{aligned}
$$

and the cross term can be bounded as usual using $\mu$-convexity and $L$-smoothness of $F_k$:

$$
\begin{aligned}
&-2\alpha_t\mathbb{E}_t\langle\overline{\mathbf{w}}_t - \mathbf{w}^*, \mathbf{g}_t\rangle \\
&= -2\alpha_t\sum_{k=1}^{N} p_k\langle\overline{\mathbf{w}}_t - \mathbf{w}^*, \nabla F_k(\mathbf{w}_t^k)\rangle \\
&= -2\alpha_t\sum_{k=1}^{N} p_k\langle\overline{\mathbf{w}}_t - \mathbf{w}_t^k, \nabla F_k(\mathbf{w}_t^k)\rangle - 2\alpha_t\sum_{k=1}^{N} p_k\langle\mathbf{w}_t^k - \mathbf{w}^*, \nabla F_k(\mathbf{w}_t^k)\rangle \\
&\leq -2\alpha_t\sum_{k=1}^{N} p_k\langle\overline{\mathbf{w}}_t - \mathbf{w}_t^k, \nabla F_k(\mathbf{w}_t^k)\rangle + 2\alpha_t\sum_{k=1}^{N} p_k(F_k(\mathbf{w}^*) - F_k(\mathbf{w}_t^k)) - \alpha_t\mu\sum_{k=1}^{N} p_k\|\mathbf{w}_t^k - \mathbf{w}^*\|^2 \\
&\leq 2\alpha_t\sum_{k=1}^{N} p_k\left[F_k(\mathbf{w}_t^k) - F_k(\overline{\mathbf{w}}_t) + \frac{L}{2}\|\overline{\mathbf{w}}_t - \mathbf{w}_t^k\|^2 + F_k(\mathbf{w}^*) - F_k(\mathbf{w}_t^k)\right] - \alpha_t\mu\|\sum_{k=1}^{N} p_k(\mathbf{w}_t^k - \mathbf{w}^*)\|^2 \\
&= \alpha_t L\sum_{k=1}^{N} p_k\|\overline{\mathbf{w}}_t - \mathbf{w}_t^k\|^2 + 2\alpha_t\sum_{k=1}^{N} p_k\left[F_k(\mathbf{w}^*) - F_k(\overline{\mathbf{w}}_t)\right] - \alpha_t\mu\|\overline{\mathbf{w}}_t - \mathbf{w}^*\|^2
\end{aligned}
$$

$$= \alpha_t L \sum_{k=1}^{N} p_k \|\overline{\mathbf{w}}_t - \mathbf{w}_t^k\|^2 - 2\alpha_t \sum_{k=1}^{N} p_k F_k(\overline{\mathbf{w}}_t) - \alpha_t \mu \|\overline{\mathbf{w}}_t - \mathbf{w}^*\|^2$$

and so

$$\mathbb{E}\|\overline{\mathbf{w}}_{t+1} - \mathbf{w}^*\|^2 \leq \mathbb{E}(1 - \alpha_t \mu)\|\overline{\mathbf{w}}_t - \mathbf{w}^*\|^2 - 2\alpha_t F(\overline{\mathbf{w}}_t) + \alpha_t^2 \|\mathbf{g}_t\|^2 + \alpha_t L \sum_{k=1}^{N} p_k \|\overline{\mathbf{w}}_t - \mathbf{w}_t^k\|^2$$

Applying this recursive relation to $\|\overline{\mathbf{w}}_t - \mathbf{w}^*\|^2$ and using $\|\overline{\mathbf{w}}_{t-1} - \mathbf{w}_{t-1}^k\|^2 \equiv 0$, we further obtain

$$\mathbb{E}\|\overline{\mathbf{w}}_{t+1} - \mathbf{w}^*\|^2 \leq \mathbb{E}(1 - \alpha_t \mu)\left((1 - \alpha_{t-1}\mu)\|\overline{\mathbf{w}}_{t-1} - \mathbf{w}^*\|^2 - 2\alpha_{t-1}F(\overline{\mathbf{w}}_{t-1}) + \alpha_{t-1}^2 \|\mathbf{g}_{t-1}\|^2\right)$$

$$- 2\alpha_t F(\overline{\mathbf{w}}_t) + \alpha_t^2 \|\mathbf{g}_t\|^2 + \alpha_t L \sum_{k=1}^{N} p_k \|\overline{\mathbf{w}}_t - \mathbf{w}_t^k\|^2$$

Now instead of bounding $\sum_{k=1}^{N} p_k \|\overline{\mathbf{w}}_t - \mathbf{w}_t^k\|^2$ using the arguments in the general convex case, we follow Ma et al. (2018) and use the fact that in the overparameterized setting, $\mathbf{w}^*$ is a minimizer of each $\ell(\mathbf{w}, x_k^j)$ and that each $\ell$ is $l$-smooth to obtain $\|\nabla F_k(\overline{\mathbf{w}}_{t-1}, \xi_{t-1}^k)\|^2 \leq 2l(F_k(\overline{\mathbf{w}}_{t-1}, \xi_{t-1}^k) - F_k(\mathbf{w}^*, \xi_{t-1}^k))$, where recall $F_k(\mathbf{w}, \xi_{t-1}^k) = \ell(\mathbf{w}, \xi_{t-1}^k)$, so that

$$\sum_{k=1}^{N} p_k \|\overline{\mathbf{w}}_t - \mathbf{w}_t^k\|^2 = \sum_{k=1}^{N} p_k \|\overline{\mathbf{w}}_{t-1} - \alpha_{t-1}\mathbf{g}_{t-1} - \mathbf{w}_{t-1}^k + \alpha_{t-1}\mathbf{g}_{t-1,k}\|^2$$

$$= \sum_{k=1}^{N} p_k \alpha_{t-1}^2 \|\mathbf{g}_{t-1} - \mathbf{g}_{t-1,k}\|^2$$

$$= \alpha_{t-1}^2 \sum_{k=1}^{N} p_k (\|\mathbf{g}_{t-1,k}\|^2 - \|\mathbf{g}_{t-1}\|^2)$$

$$= \alpha_{t-1}^2 \sum_{k=1}^{N} p_k \|\nabla F_k(\overline{\mathbf{w}}_{t-1}, \xi_{t-1}^k)\|^2 - \alpha_{t-1}^2 \|\mathbf{g}_{t-1}\|^2$$

$$\leq \alpha_{t-1}^2 \sum_{k=1}^{N} p_k 2l(F_k(\overline{\mathbf{w}}_{t-1}, \xi_{t-1}^k) - F_k(\mathbf{w}^*, \xi_{t-1}^k)) - \alpha_{t-1}^2 \|\mathbf{g}_{t-1}\|^2$$

again using $\overline{\mathbf{w}}_{t-1} = \mathbf{w}_{t-1}^k$. Taking expectation with respect to $\xi_{t-1}^k$'s and using the fact that $F(\mathbf{w}^*) = 0$, we have

$$\mathbb{E}_{t-1} \sum_{k=1}^{N} p_k \|\overline{\mathbf{w}}_t - \mathbf{w}_t^k\|^2 \leq 2l\alpha_{t-1}^2 \sum_{k=1}^{N} p_k F_k(\overline{\mathbf{w}}_{t-1}) - \alpha_{t-1}^2 \|\mathbf{g}_{t-1}\|^2$$

$$= 2l\alpha_{t-1}^2 F(\overline{\mathbf{w}}_{t-1}) - \alpha_{t-1}^2 \|\mathbf{g}_{t-1}\|^2$$

Note also that

$$\|\mathbf{g}_{t-1}\|^2 = \|\sum_{k=1}^{N} p_k \nabla F_k(\overline{\mathbf{w}}_{t-1}, \xi_{t-1}^k)\|^2$$

while

$$\|\mathbf{g}_t\|^2 = \|\sum_{k=1}^{N} p_k \nabla F_k(\mathbf{w}_t^k, \xi_t^k)\|^2 \leq 2\|\sum_{k=1}^{N} p_k \nabla F_k(\overline{\mathbf{w}}_t, \xi_t^k)\|^2 + 2\|\sum_{k=1}^{N} p_k(\nabla F_k(\overline{\mathbf{w}}_t, \xi_t^k) - \nabla F_k(\mathbf{w}_t^k, \xi_t^k))\|^2$$

$$\leq 2\|\sum_{k=1}^{N} p_k \nabla F_k(\overline{\mathbf{w}}_t, \xi_t^k)\|^2 + 2\sum_{k=1}^{N} p_k l^2 \|\overline{\mathbf{w}}_t - \mathbf{w}_t^k\|^2$$

Substituting these into the bound for $\|\overline{\mathbf{w}}_{t+1} - \mathbf{w}^*\|^2$, we have

$$\mathbb{E}\|\overline{\mathbf{w}}_{t+1} - \mathbf{w}^*\|^2 \leq \mathbb{E}(1 - \alpha_t\mu)((1 - \alpha_{t-1}\mu)\|\overline{\mathbf{w}}_{t-1} - \mathbf{w}^*\|^2 - 2\alpha_{t-1}F(\overline{\mathbf{w}}_{t-1}) + \alpha_{t-1}^2\|\mathbf{g}_{t-1}\|^2)$$

$$- 2\alpha_t F(\overline{\mathbf{w}}_t) + 2\alpha_t^2\|\sum_{k=1}^{N} p_k \nabla F_k(\overline{\mathbf{w}}_t, \xi_t^k)\|^2 + \left(2l^2\alpha_{t-1}^2\alpha_t^2 + \alpha_t\alpha_{t-1}^2 L\right)\left(2lF(\overline{\mathbf{w}}_{t-1}) - \|\mathbf{g}_{t-1}\|^2\right)$$

$$= \mathbb{E}(1 - \alpha_t\mu)(1 - \alpha_{t-1}\mu)\|\overline{\mathbf{w}}_{t-1} - \mathbf{w}^*\|^2$$

$$- 2\alpha_t(F(\overline{\mathbf{w}}_t) - \alpha_t\|\sum_{k=1}^{N} p_k \nabla F_k(\overline{\mathbf{w}}_t, \xi_t^k)\|^2)$$

$$- 2\alpha_{t-1}(1 - \alpha_t\mu)\left((1 - \frac{l\alpha_{t-1}(2l^2\alpha_t^2 + \alpha_t L)}{1 - \alpha_t\mu})F(\overline{\mathbf{w}}_{t-1}) - \frac{\alpha_{t-1}}{2}\|\sum_{k=1}^{N} p_k \nabla F_k(\overline{\mathbf{w}}_{t-1}, \xi_{t-1}^k)\|^2\right)$$

from which we can conclude that

$$\mathbb{E}\|\overline{\mathbf{w}}_{t+1} - \mathbf{w}^*\|^2 \leq (1 - \alpha_t\mu)(1 - \alpha_{t-1}\mu)\mathbb{E}\|\overline{\mathbf{w}}_{t-1} - \mathbf{w}^*\|^2$$

if we can choose $\alpha_t, \alpha_{t-1}$ to guarantee

$$\mathbb{E}(F(\overline{\mathbf{w}}_t) - \alpha_t\|\sum_{k=1}^{N} p_k \nabla F_k(\overline{\mathbf{w}}_t, \xi_t^k)\|^2) \geq 0$$

$$\mathbb{E}\left((1 - \frac{l\alpha_{t-1}(2l^2\alpha_t^2 + \alpha_t L)}{1 - \alpha_t\mu})F(\overline{\mathbf{w}}_{t-1}) - \frac{\alpha_{t-1}}{2}\|\sum_{k=1}^{N} p_k \nabla F_k(\overline{\mathbf{w}}_{t-1}, \xi_{t-1}^k)\|^2\right) \geq 0$$

Note that

$$\mathbb{E}_t\|\sum_{k=1}^{N} p_k \nabla F_k(\overline{\mathbf{w}}_t, \xi_t^k)\|^2 = \mathbb{E}_t\langle\sum_{k=1}^{N} p_k \nabla F_k(\overline{\mathbf{w}}_t, \xi_t^k), \sum_{k=1}^{N} p_k \nabla F_k(\overline{\mathbf{w}}_t, \xi_t^k)\rangle$$

$$= \sum_{k=1}^{N} p_k^2 \mathbb{E}_t\|\nabla F_k(\overline{\mathbf{w}}_t, \xi_t^k)\|^2 + \sum_{k=1}^{N}\sum_{j\neq k} p_j p_k \mathbb{E}_t\langle\nabla F_k(\overline{\mathbf{w}}_t, \xi_t^k), \nabla F_j(\overline{\mathbf{w}}_t, \xi_t^j)\rangle$$

$$= \sum_{k=1}^{N} p_k^2 \mathbb{E}_t\|\nabla F_k(\overline{\mathbf{w}}_t, \xi_t^k)\|^2 + \sum_{k=1}^{N}\sum_{j\neq k} p_j p_k \langle\nabla F_k(\overline{\mathbf{w}}_t), \nabla F_j(\overline{\mathbf{w}}_t)\rangle$$

$$= \sum_{k=1}^{N} p_k^2 \mathbb{E}_t\|\nabla F_k(\overline{\mathbf{w}}_t, \xi_t^k)\|^2 + \sum_{k=1}^{N}\sum_{j=1}^{N} p_j p_k \langle\nabla F_k(\overline{\mathbf{w}}_t), \nabla F_j(\overline{\mathbf{w}}_t)\rangle - \sum_{k=1}^{N} p_k^2\|\nabla F_k(\overline{\mathbf{w}}_t)\|^2$$

$$\leq \sum_{k=1}^{N} p_k^2 \mathbb{E}_t\|\nabla F_k(\overline{\mathbf{w}}_t, \xi_t^k)\|^2 + \|\sum_{k} p_k \nabla F_k(\overline{\mathbf{w}}_t)\|^2 - \frac{1}{N}\nu_{\min}\|\sum_{k} p_k \nabla F_k(\overline{\mathbf{w}}_t)\|^2$$

$$= \sum_{k=1}^{N} p_k^2 \mathbb{E}_t\|\nabla F_k(\overline{\mathbf{w}}_t, \xi_t^k)\|^2 + (1 - \frac{1}{N}\nu_{\min})\|\nabla F(\overline{\mathbf{w}}_t)\|^2$$

and so following Ma et al. (2018) if we let $\alpha_t = \min\{\frac{qN}{2l\nu_{\max}}, \frac{1-q}{2L(1 - \frac{1}{N}\nu_{\min})}\}$ for a $q \in [0, 1]$ to be optimized later, we have

$$\mathbb{E}_t(F(\overline{\mathbf{w}}_t) - \alpha_t\|\sum_{k=1}^{N} p_k \nabla F_k(\overline{\mathbf{w}}_t, \xi_t^k)\|^2)$$

$$\geq \mathbb{E}_t \sum_{k=1}^{N} p_k F_k(\overline{\mathbf{w}}_t) - \alpha_t\left[\sum_{k=1}^{N} p_k^2 \mathbb{E}_t\|\nabla F_k(\overline{\mathbf{w}}_t, \xi_t^k)\|^2 + (1 - \frac{1}{N}\nu_{\min})\|\nabla F(\overline{\mathbf{w}}_t)\|^2\right]$$

$$\geq \mathbb{E}_t \sum_{k=1}^{N} p_k (q F_k(\overline{\mathbf{w}}_t, \xi_t^k) - \alpha_t \frac{1}{N} \nu_{\max} \|\nabla F_k(\overline{\mathbf{w}}_t, \xi_t^k)\|^2) + ((1-q)F(\overline{\mathbf{w}}_t) - \alpha_t(1 - \frac{1}{N}\nu_{\min})\|\nabla F(\overline{\mathbf{w}}_t)\|^2)$$

$$\geq q\mathbb{E}_t \sum_{k=1}^{N} p_k (F_k(\overline{\mathbf{w}}_t, \xi_t^k) - \frac{1}{2l}\|\nabla F_k(\overline{\mathbf{w}}_t, \xi_t^k)\|^2) + (1-q)(F(\overline{\mathbf{w}}_t) - \frac{1}{2L}\|\nabla F(\overline{\mathbf{w}}_t)\|^2)$$

$$\geq 0$$

again using $\mathbf{w}^*$ optimizes $F_k(\mathbf{w}, \xi_t^k)$ with $F_k(\mathbf{w}^*, \xi_t^k) = 0$.

Maximizing $\alpha_t = \min\{\frac{qN}{2l\nu_{\max}}, \frac{1-q}{2L(1-\frac{1}{N}\nu_{\min})}\}$ over $q \in [0, 1]$, we see that $q = \frac{l\nu_{\max}}{l\nu_{\max}+L(N-\nu_{\min})}$ results in the fastest convergence, and this translates to $\alpha_t = \frac{1}{2}\frac{N}{l\nu_{\max}+L(N-\nu_{\min})}$. Next we claim that $\alpha_{t-1} = c\frac{1}{2}\frac{N}{l\nu_{\max}+L(N-\nu_{\min})}$ also guarantees

$$\mathbb{E}(1 - \frac{l\alpha_{t-1}(2l^2\alpha_t^2 + \alpha_t L)}{1 - \alpha_t \mu})F(\overline{\mathbf{w}}_{t-1}) - \frac{\alpha_{t-1}}{2}\|\sum_{k=1}^{N} p_k \nabla F_k(\overline{\mathbf{w}}_{t-1}, \xi_{t-1}^k)\|^2 \geq 0$$

Note that by scaling $\alpha_{t-1}$ by a constant $c \leq 1$ if necessary, we can guarantee $\frac{l\alpha_{t-1}(2l^2\alpha_t^2 + \alpha_t L)}{1 - \alpha_t \mu} \leq \frac{1}{2}$, and so the condition is equivalent to

$$F(\overline{\mathbf{w}}_{t-1}) - \alpha_{t-1}\|\sum_{k=1}^{N} p_k \nabla F_k(\overline{\mathbf{w}}_{t-1}, \xi_{t-1}^k)\|^2 \geq 0$$

which was shown to hold with $\alpha_{t-1} \leq \frac{1}{2}\frac{N}{l\nu_{\max}+L(N-\nu_{\min})}$.

For the proof of general $E \geq 2$, we use the following two identities:

$$\|\mathbf{g}_t\|^2 \leq 2\|\sum_{k=1}^{N} p_k \nabla F_k(\overline{\mathbf{w}}_t, \xi_t^k)\|^2 + 2\sum_{k=1}^{N} p_k l^2 \|\overline{\mathbf{w}}_t - \mathbf{w}_t^k\|^2$$

$$\mathbb{E}\sum_{k=1}^{N} p_k \|\overline{\mathbf{w}}_t - \mathbf{w}_t^k\|^2 \leq \mathbb{E}2(1 + 2l^2\alpha_{t-1}^2)\sum_{k=1}^{N} p_k \|\overline{\mathbf{w}}_{t-1} - \mathbf{w}_{t-1}^k\|^2 + 8\alpha_{t-1}^2 lF(\overline{\mathbf{w}}_{t-1}) - 2\alpha_{t-1}^2\|\mathbf{g}_{t-1}\|^2$$

where the first inequality has been established before. To establish the second inequality, note that

$$\sum_{k=1}^{N} p_k \|\overline{\mathbf{w}}_t - \mathbf{w}_t^k\|^2 = \sum_{k=1}^{N} p_k \|\overline{\mathbf{w}}_{t-1} - \alpha_{t-1}\mathbf{g}_{t-1} - \mathbf{w}_{t-1}^k + \alpha_{t-1}\mathbf{g}_{t-1,k}\|^2$$

$$\leq 2\sum_{k=1}^{N} p_k \left(\|\overline{\mathbf{w}}_{t-1} - \mathbf{w}_{t-1}^k\|^2 + \|\alpha_{t-1}\mathbf{g}_{t-1} - \alpha_{t-1}\mathbf{g}_{t-1,k}\|^2\right)$$

and

$$\sum_k p_k \|\mathbf{g}_{t-1,k} - \mathbf{g}_{t-1}\|^2 = \sum_k p_k(\|\mathbf{g}_{t-1,k}\|^2 - \|\mathbf{g}_{t-1}\|^2)$$

$$= \sum_k p_k \|\nabla F_k(\overline{\mathbf{w}}_{t-1}, \xi_{t-1}^k) + \nabla F_k(\mathbf{w}_{t-1}^k, \xi_{t-1}^k) - \nabla F_k(\overline{\mathbf{w}}_{t-1}, \xi_{t-1}^k)\|^2 - \|\mathbf{g}_{t-1}\|^2$$

$$\leq 2\sum_k p_k \left(\|\nabla F_k(\overline{\mathbf{w}}_{t-1}, \xi_{t-1}^k)\|^2 + l^2\|\mathbf{w}_{t-1}^k - \overline{\mathbf{w}}_{t-1}\|^2\right) - \|\mathbf{g}_{t-1}\|^2$$

so that using the $l$-smoothness of $\ell$,

$$\mathbb{E}\sum_{k=1}^{N} p_k \|\overline{\mathbf{w}}_t - \mathbf{w}_t^k\|^2$$

$$\leq \mathbb{E}2(1 + 2l^2\alpha_{t-1}^2)\sum_{k=1}^{N} p_k \|\overline{\mathbf{w}}_{t-1} - \mathbf{w}_{t-1}^k\|^2 + 4\alpha_{t-1}^2\sum_k p_k \|\nabla F_k(\overline{\mathbf{w}}_{t-1}, \xi_{t-1}^k)\|^2 - 2\alpha_{t-1}^2\|\mathbf{g}_{t-1}\|^2$$

$$\leq \mathbb{E}2(1 + 2l^2\alpha_{t-1}^2)\sum_{k=1}^{N} p_k\|\overline{\mathbf{w}}_{t-1} - \mathbf{w}_{t-1}^k\|^2 + 4\alpha_{t-1}^2 2l\sum_{k} p_k(F_k(\overline{\mathbf{w}}_{t-1}, \xi_{t-1}^k) - F_k(\mathbf{w}^*, \xi_{t-1}^k)) - 2\alpha_{t-1}^2\|\mathbf{g}_{t-1}\|^2$$

$$= \mathbb{E}2(1 + 2l^2\alpha_{t-1}^2)\sum_{k=1}^{N} p_k\|\overline{\mathbf{w}}_{t-1} - \mathbf{w}_{t-1}^k\|^2 + 8\alpha_{t-1}^2 lF(\overline{\mathbf{w}}_{t-1}) - 2\alpha_{t-1}^2\|\mathbf{g}_{t-1}\|^2$$

Using the first inequality, we have

$$\mathbb{E}\|\overline{\mathbf{w}}_{t+1} - \mathbf{w}^*\|^2 \leq \mathbb{E}(1 - \alpha_t\mu)\|\overline{\mathbf{w}}_t - \mathbf{w}^*\|^2$$

$$- 2\alpha_t F(\overline{\mathbf{w}}_t) + 2\alpha_t^2\|\sum_{k=1}^{N} p_k\nabla F_k(\overline{\mathbf{w}}_t, \xi_t^k)\|^2$$

$$+ (2\alpha_t^2 l^2 + \alpha_t L)\sum_{k=1}^{N} p_k\|\overline{\mathbf{w}}_t - \mathbf{w}_t^k\|^2$$

and we choose $\alpha_t$ and $\alpha_{t-1}$ such that $\mathbb{E}(F(\overline{\mathbf{w}}_t) - \alpha_t\|\sum_{k=1}^{N} p_k\nabla F_k(\overline{\mathbf{w}}_t, \xi_t^k)\|^2) \geq 0$ and $(2\alpha_t^2 l^2 + \alpha_t L) \leq (1 - \alpha_t\mu)(2\alpha_{t-1}^2 l^2 + \alpha_{t-1}L)/3$. This gives

$$\mathbb{E}\|\overline{\mathbf{w}}_{t+1} - \mathbf{w}^*\|^2 \leq \mathbb{E}(1 - \alpha_t\mu)[(1 - \alpha_{t-1}\mu)\|\overline{\mathbf{w}}_{t-1} - \mathbf{w}^*\|^2 - 2\alpha_{t-1}F(\overline{\mathbf{w}}_{t-1}) + 2\alpha_{t-1}^2\|\sum_{k=1}^{N} p_k\nabla F_k(\overline{\mathbf{w}}_{t-1}, \xi_{t-1}^k)\|^2$$

$$+ (2\alpha_{t-1}^2 l^2 + \alpha_{t-1}L)(\sum_{k=1}^{N} p_k\|\overline{\mathbf{w}}_{t-1} - \mathbf{w}_{t-1}^k\|^2 + \sum_{k=1}^{N} p_k\|\overline{\mathbf{w}}_t - \mathbf{w}_t^k\|^2)/3]$$

Using the second inequality

$$\sum_{k=1}^{N} p_k\|\overline{\mathbf{w}}_t - \mathbf{w}_t^k\|^2 \leq \mathbb{E}2(1 + 2l^2\alpha_{t-1}^2)\sum_{k=1}^{N} p_k\|\overline{\mathbf{w}}_{t-1} - \mathbf{w}_{t-1}^k\|^2 + 8\alpha_{t-1}^2 lF(\overline{\mathbf{w}}_{t-1}) - 2\alpha_{t-1}^2\|\mathbf{g}_{t-1}\|^2$$

and that $2(1 + 2l^2\alpha_{t-1}^2) \leq 3$, $2\alpha_{t-1}^2 l^2 + \alpha_{t-1}L \leq 1$, we have

$$\mathbb{E}\|\overline{\mathbf{w}}_{t+1} - \mathbf{w}^*\|^2 \leq \mathbb{E}(1 - \alpha_t\mu)[(1 - \alpha_{t-1}\mu)\|\overline{\mathbf{w}}_{t-1} - \mathbf{w}^*\|^2$$

$$- 2\alpha_{t-1}F(\overline{\mathbf{w}}_{t-1}) + 2\alpha_{t-1}^2\|\sum_{k=1}^{N} p_k\nabla F_k(\overline{\mathbf{w}}_{t-1}, \xi_{t-1}^k)\|^2 + 8\alpha_{t-1}^2 lF(\overline{\mathbf{w}}_{t-1})$$

$$+ (2\alpha_{t-1}^2 l^2 + \alpha_{t-1}L)(2\sum_{k=1}^{N} p_k\|\overline{\mathbf{w}}_{t-1} - \mathbf{w}_{t-1}^k\|^2)]$$

and if $\alpha_{t-1}$ is chosen such that

$$(F(\overline{\mathbf{w}}_{t-1}) - 4\alpha_{t-1}lF(\overline{\mathbf{w}}_{t-1})) - \alpha_{t-1}\|\sum_{k=1}^{N} p_k\nabla F_k(\overline{\mathbf{w}}_{t-1}, \xi_{t-1}^k)\|^2 \geq 0$$

and

$$(2\alpha_{t-1}^2 l^2 + \alpha_{t-1}L)(1 - \alpha_{t-1}\mu) \leq (2\alpha_{t-2}^2 l^2 + \alpha_{t-2}L)/3$$

we again have

$$\mathbb{E}\|\overline{\mathbf{w}}_{t+1} - \mathbf{w}^*\|^2 \leq \mathbb{E}(1 - \alpha_t\mu)(1 - \alpha_{t-1}\mu)[\|\overline{\mathbf{w}}_{t-1} - \mathbf{w}^*\|^2 + (2\alpha_{t-2}^2 l^2 + \alpha_{t-2}L)\cdot(2\sum_{k=1}^{N} p_k\|\overline{\mathbf{w}}_{t-1} - \mathbf{w}_{t-1}^k\|^2)/3]$$

Applying the above derivation iteratively $\tau < E$ times, we have

$$\mathbb{E}\|\overline{\mathbf{w}}_{t+1} - \mathbf{w}^*\|^2 \leq \mathbb{E}(1 - \alpha_t\mu)\cdots(1 - \alpha_{t-\tau+1}\mu)[(1 - \alpha_{t-\tau}\mu)\|\overline{\mathbf{w}}_{t-\tau} - \mathbf{w}^*\|^2$$

$$- 2\alpha_{t-\tau} F(\overline{\mathbf{w}}_{t-\tau}) + 2\alpha_{t-\tau}^2 \| \sum_{k=1}^{N} p_k \nabla F_k(\overline{\mathbf{w}}_{t-\tau}, \xi_{t-\tau}^k) \|^2 + 8\tau \alpha_{t-\tau}^2 l F(\overline{\mathbf{w}}_{t-\tau})$$

$$+ (2\alpha_{t-\tau}^2 l^2 + \alpha_{t-\tau} L)((\tau+1) \sum_{k=1}^{N} p_k \| \overline{\mathbf{w}}_{t-\tau} - \mathbf{w}_{t-\tau}^k \|^2)]$$

as long as the step sizes $\alpha_{t-\tau}$ are chosen such that the following inequalities hold

$$(2\alpha_{t-\tau}^2 l^2 + \alpha_{t-\tau} L)(1 - \alpha_{t-\tau}\mu) \leq (2\alpha_{t-\tau-1}^2 l^2 + \alpha_{t-\tau-1} L)/3$$

$$2(1 + 2l^2 \alpha_{t-\tau}^2) \leq 3$$

$$2\alpha_{t-\tau}^2 l^2 + \alpha_{t-\tau} L \leq 1$$

$$(F(\overline{\mathbf{w}}_{t-\tau}) - 4\tau \alpha_{t-\tau} l F(\overline{\mathbf{w}}_{t-\tau})) - \alpha_{t-\tau} \| \sum_{k=1}^{N} p_k \nabla F_k(\overline{\mathbf{w}}_{t-\tau}, \xi_{t-\tau}^k) \|^2 \geq 0$$

We can check that setting $\alpha_{t-\tau} = c\frac{1}{\tau+1} \frac{N}{l\nu_{\max} + L(N - \nu_{\min})}$ for some small constant $c$ satisfies the requirements.

Since communication is done every $E$ iterations, $\overline{\mathbf{w}}_{t_0} = \mathbf{w}_{t_0}^k$ for some $t_0 > t - E$, from which we can conclude that

$$\mathbb{E}\|\overline{\mathbf{w}}_t - \mathbf{w}^*\|^2 \leq ( \prod_{\tau=1}^{t-t_0-1} (1 - \mu\alpha_{t-\tau})) \|\mathbf{w}_{t_0} - \mathbf{w}^*\|^2$$

$$\leq (1 - c\frac{\mu}{E} \frac{N}{l\nu_{\max} + L(N - \nu_{\min})})^{t-t_0} \|\mathbf{w}_{t_0} - \mathbf{w}^*\|^2$$

and applying this inequality to iterations between each communication round,

$$\mathbb{E}\|\overline{\mathbf{w}}_t - \mathbf{w}^*\|^2 \leq (1 - c\frac{\mu}{E} \frac{N}{l\nu_{\max} + L(N - \nu_{\min})})^{t} \|\mathbf{w}_0 - \mathbf{w}^*\|^2$$

$$= O(\exp(\frac{\mu}{E} \frac{N}{l\nu_{\max} + L(N - \nu_{\min})} t)) \|\mathbf{w}_0 - \mathbf{w}^*\|^2$$

With partial participation, we note that

$$\mathbb{E}\|\overline{\mathbf{w}}_{t+1} - \mathbf{w}^*\|^2 = \mathbb{E}\|\overline{\mathbf{w}}_{t+1} - \overline{\mathbf{v}}_{t+1} + \overline{\mathbf{v}}_{t+1} - \mathbf{w}^*\|^2$$

$$= \mathbb{E}\|\overline{\mathbf{w}}_{t+1} - \overline{\mathbf{v}}_{t+1}\|^2 + \mathbb{E}\|\overline{\mathbf{v}}_{t+1} - \mathbf{w}^*\|^2$$

$$= \frac{1}{K} \sum_k p_k \mathbb{E}\|\mathbf{w}_{t+1}^k - \overline{\mathbf{w}}_{t+1}\|^2 + \mathbb{E}\|\overline{\mathbf{v}}_{t+1} - \mathbf{w}^*\|^2$$

and so the recursive identity becomes

$$\mathbb{E}\|\overline{\mathbf{w}}_{t+1} - \mathbf{w}^*\|^2 \leq \mathbb{E}(1 - \alpha_t\mu)\cdots(1 - \alpha_{t-\tau+1}\mu)[(1 - \alpha_{t-\tau}\mu)\|\overline{\mathbf{w}}_{t-\tau} - \mathbf{w}^*\|^2$$

$$- 2\alpha_{t-\tau} F(\overline{\mathbf{w}}_{t-\tau}) + 2\alpha_{t-\tau}^2 \| \sum_{k=1}^{N} p_k \nabla F_k(\overline{\mathbf{w}}_{t-\tau}, \xi_{t-\tau}^k) \|^2 + 8\tau \alpha_{t-\tau}^2 l F(\overline{\mathbf{w}}_{t-\tau})$$

$$+ (2\alpha_{t-\tau}^2 l^2 + \alpha_{t-\tau} L + \frac{1}{K})((\tau+1) \sum_{k=1}^{N} p_k \| \overline{\mathbf{w}}_{t-\tau} - \mathbf{w}_{t-\tau}^k \|^2)]$$

which requires

$$(2\alpha_{t-\tau}^2 l^2 + \alpha_{t-\tau} L + \frac{1}{K})(1 - \alpha_{t-\tau}\mu) \leq (2\alpha_{t-\tau-1}^2 l^2 + \alpha_{t-\tau-1} L + \frac{1}{K})/3$$

$$2(1 + 2l^2 \alpha_{t-\tau}^2) \leq 3$$

$$2\alpha_{t-\tau}^2 l^2 + \alpha_{t-\tau} L + \frac{1}{K} \leq 1$$

$$(F(\overline{\mathbf{w}}_{t-\tau}) - 4\tau\alpha_{t-\tau}lF(\overline{\mathbf{w}}_{t-\tau})) - \alpha_{t-\tau}\|\sum_{k=1}^{N} p_k \nabla F_k(\overline{\mathbf{w}}_{t-\tau}, \xi_{t-\tau}^k)\|^2 \geq 0$$

to hold. Again setting $\alpha_{t-\tau} = c\frac{1}{\tau+1}\frac{N}{l\nu_{\max}+L(N-\nu_{\min})}$ for a possibly different constant from before satisfies the requirements.

Finally, using the $L$-smoothness of $F$,

$$F(\overline{\mathbf{w}}_T) - F(\mathbf{w}^*) \leq \frac{L}{2}\mathbb{E}\|\overline{\mathbf{w}}_T - \mathbf{w}^*\|^2 = O(L\exp(-\frac{\mu}{E}\frac{N}{l\nu_{\max}+L(N-\nu_{\min})}T))\|\mathbf{w}_0 - \mathbf{w}^*\|^2$$

$\square$

### H.2 GEOMETRIC CONVERGENCE OF FEDAVG FOR OVERPARAMETERIZED LINEAR REGRESSION

We first provide details on quantities used in the proof of results on linear regression in Section G. The local device objectives are now given by the sum of squares $F_k(\mathbf{w}) = \frac{1}{2n_k}\sum_{j=1}^{n_k}(\mathbf{w}^T\mathbf{x}_k^j - \mathbf{z}_k^j)^2$, and there exists $\mathbf{w}^*$ such that $F(\mathbf{w}^*) \equiv 0$. Define the local Hessian matrix as $\mathbf{H}^k := \frac{1}{n_k}\sum_{j=1}^{n_k}\mathbf{x}_k^j(\mathbf{x}_k^j)^T$, and the stochastic Hessian matrix as $\tilde{\mathbf{H}}_t^k := \xi_t^k(\xi_t^k)^T$, where $\xi_t^k$ is the stochastic sample on the $k$th device at time $t$. Define $l$ to be the smallest positive number such that $\mathbb{E}\|\xi_t^k\|^2\xi_t^k(\xi_t^k)^T \preceq l\mathbf{H}^k$ for all $k$. Note that $l \leq \max_{k,j}\|\mathbf{x}_k^j\|^2$. Let $L$ and $\mu$ be lower and upper bounds of non-zero eigenvalues of $\mathbf{H}^k$. Define $\kappa_1 := l/\mu$ and $\kappa := L/\mu$.

Following Liu & Belkin (2020); Jain et al. (2017), we define the statistical condition number $\tilde{\kappa}$ as the smallest positive real number such that $\mathbb{E}\sum_k p_k\tilde{\mathbf{H}}_t^k\mathbf{H}^{-1}\tilde{\mathbf{H}}_t^k \leq \tilde{\kappa}\mathbf{H}$. The condition numbers $\kappa_1$ and $\tilde{\kappa}$ are important in the characterization of convergence rates for FedAvg algorithms. Note that $\kappa_1 > \kappa$ and $\kappa_1 > \tilde{\kappa}$.

Let $\mathbf{H} = \sum_k p_k\mathbf{H}^k$. In general $\mathbf{H}$ has zero eigenvalues. However, because the null space of $\mathbf{H}$ and range of $\mathbf{H}$ are orthogonal, in our subsequence analysis it suffices to project $\overline{\mathbf{w}}_t - \mathbf{w}^*$ onto the range of $\mathbf{H}$, thus we may restrict to the non-zero eigenvalue of $\mathbf{H}$.

A useful observation is that we can use $\mathbf{w}^{*T}\mathbf{x}_k^j - \mathbf{z}_k^j \equiv 0$ to rewrite the local objectives as $F_k(\mathbf{w}) = \frac{1}{2}\langle\mathbf{w} - \mathbf{w}^*, \mathbf{H}^k(\mathbf{w} - \mathbf{w}^*)\rangle \equiv \frac{1}{2}\|\mathbf{w} - \mathbf{w}^*\|_{\mathbf{H}^k}^2$:

$$F_k(\mathbf{w}) = \frac{1}{2n_k}\sum_{j=1}^{n_k}(\mathbf{w}^T\mathbf{x}_{k,j} - \mathbf{z}_{k,j} - (\mathbf{w}^{*T}\mathbf{x}_{k,j} - \mathbf{z}_{k,j}))^2 = \frac{1}{2n_k}\sum_{j=1}^{n_k}((\mathbf{w} - \mathbf{w}^*)^T\mathbf{x}_{k,j})^2$$

$$= \frac{1}{2}\langle\mathbf{w} - \mathbf{w}^*, \mathbf{H}^k(\mathbf{w} - \mathbf{w}^*)\rangle = \frac{1}{2}\|\mathbf{w} - \mathbf{w}^*\|_{\mathbf{H}^k}^2$$

so that $F(\mathbf{w}) = \frac{1}{2}\|\mathbf{w} - \mathbf{w}^*\|_H^2$.

Finally, note that $\mathbb{E}\tilde{\mathbf{H}}_t^k = \frac{1}{n_k}\sum_{j=1}^{n_k}\mathbf{x}_k^j(\mathbf{x}_k^j)^T = \mathbf{H}^k$ and $\mathbf{g}_{t,k} = \nabla F_k(\mathbf{w}_t^k, \xi_t^k) = \tilde{\mathbf{H}}_t^k(\mathbf{w}_t^k - \mathbf{w}^*)$ while $\mathbf{g}_t = \sum_{k=1}^N p_k \nabla F_k(\mathbf{w}_t^k, \xi_t^k) = \sum_{k=1}^N p_k\tilde{\mathbf{H}}_t^k(\mathbf{w}_t^k - \mathbf{w}^*)$ and $\overline{\mathbf{g}}_t = \sum_{k=1}^N p_k\mathbf{H}^k(\mathbf{w}_t^k - \mathbf{w}^*)$

**Theorem 6.** *For the overparamterized linear regression problem, FedAvg with communication every $E$ iterations with constant step size $\overline{\alpha} = \mathcal{O}(\frac{1}{E}\frac{N}{l\nu_{\max}+\mu(N-\nu_{\min})})$ has geometric convergence:*

$$\mathbb{E}F(\overline{\mathbf{w}}_T) \leq \mathcal{O}\left(L\exp(-\frac{NT}{E(\nu_{\max}\kappa_1 + (N-\nu_{\min}))})\|\mathbf{w}_0 - \mathbf{w}^*\|^2\right).$$

*Proof.* We again show the result first when $E = 2$ and $t - 1$ is a communication round. We have

$$\|\overline{\mathbf{w}}_{t+1} - \mathbf{w}^*\|^2 = \|(\overline{\mathbf{w}}_t - \alpha_t\mathbf{g}_t) - \mathbf{w}^*\|^2$$
$$= \|\overline{\mathbf{w}}_t - \mathbf{w}^*\|^2 - 2\alpha_t\langle\overline{\mathbf{w}}_t - \mathbf{w}^*, \mathbf{g}_t\rangle + \alpha_t^2\|\mathbf{g}_t\|^2$$

and

$$-2\alpha_t\mathbb{E}_t\langle\overline{\mathbf{w}}_t - \mathbf{w}^*, \mathbf{g}_t\rangle$$

$$= -2\alpha_t \sum_{k=1}^{N} p_k \langle \overline{\mathbf{w}}_t - \mathbf{w}^*, \nabla F_k(\mathbf{w}_t^k) \rangle$$

$$= -2\alpha_t \sum_{k=1}^{N} p_k \langle \overline{\mathbf{w}}_t - \mathbf{w}_t^k, \nabla F_k(\mathbf{w}_t^k) \rangle - 2\alpha_t \sum_{k=1}^{N} p_k \langle \mathbf{w}_t^k - \mathbf{w}^*, \nabla F_k(\mathbf{w}_t^k) \rangle$$

$$= -2\alpha_t \sum_{k=1}^{N} p_k \langle \overline{\mathbf{w}}_t - \mathbf{w}_t^k, \nabla F_k(\mathbf{w}_t^k) \rangle - 2\alpha_t \sum_{k=1}^{N} p_k \langle \mathbf{w}_t^k - \mathbf{w}^*, \mathbf{H}^k(\mathbf{w}_t^k - \mathbf{w}^*) \rangle$$

$$= -2\alpha_t \sum_{k=1}^{N} p_k \langle \overline{\mathbf{w}}_t - \mathbf{w}_t^k, \nabla F_k(\mathbf{w}_t^k) \rangle - 4\alpha_t \sum_{k=1}^{N} p_k F_k(\mathbf{w}_t^k)$$

$$\leq 2\alpha_t \sum_{k=1}^{N} p_k (F_k(\mathbf{w}_t^k) - F_k(\overline{\mathbf{w}}_t) + \frac{L}{2} \|\overline{\mathbf{w}}_t - \mathbf{w}_t^k\|^2) - 4\alpha_t \sum_{k=1}^{N} p_k F_k(\mathbf{w}_t^k)$$

$$= \alpha_t L \sum_{k=1}^{N} p_k \|\overline{\mathbf{w}}_t - \mathbf{w}_t^k\|^2 - 2\alpha_t \sum_{k=1}^{N} p_k F_k(\overline{\mathbf{w}}_t) - 2\alpha_t \sum_{k=1}^{N} p_k F_k(\mathbf{w}_t^k)$$

$$= \alpha_t L \sum_{k=1}^{N} p_k \|\overline{\mathbf{w}}_t - \mathbf{w}_t^k\|^2 - \alpha_t \sum_{k=1}^{N} p_k \langle (\overline{\mathbf{w}}_t - \mathbf{w}^*), \mathbf{H}^k(\overline{\mathbf{w}}_t - \mathbf{w}^*) \rangle - 2\alpha_t \sum_{k=1}^{N} p_k F_k(\mathbf{w}_t^k)$$

and

$$\|\mathbf{g}_t\|^2 = \| \sum_{k=1}^{N} p_k \tilde{\mathbf{H}}_t^k(\mathbf{w}_t^k - \mathbf{w}^*) \|^2$$

$$= \| \sum_{k=1}^{N} p_k \tilde{\mathbf{H}}_t^k(\overline{\mathbf{w}}_t - \mathbf{w}^*) + \sum_{k=1}^{N} p_k \tilde{\mathbf{H}}_t^k(\mathbf{w}_t^k - \overline{\mathbf{w}}_t) \|^2$$

$$\leq 2\| \sum_{k=1}^{N} p_k \tilde{\mathbf{H}}_t^k(\overline{\mathbf{w}}_t - \mathbf{w}^*) \|^2 + 2\| \sum_{k=1}^{N} p_k \tilde{\mathbf{H}}_t^k(\mathbf{w}_t^k - \overline{\mathbf{w}}_t) \|^2$$

which gives

$$\mathbb{E}\|\overline{\mathbf{w}}_{t+1} - \mathbf{w}^*\|^2 \leq \mathbb{E}\|\overline{\mathbf{w}}_t - \mathbf{w}^*\|^2 - \alpha_t \sum_{k=1}^{N} p_k \langle \overline{\mathbf{w}}_t - \mathbf{w}^*, \mathbf{H}^k \overline{\mathbf{w}}_t - \mathbf{w}^* \rangle + 2\alpha_t^2 \| \sum_{k=1}^{N} p_k \tilde{\mathbf{H}}_t^k(\overline{\mathbf{w}}_t - \mathbf{w}^*) \|^2$$

$$+ \alpha_t L \sum_{k=1}^{N} p_k \|\overline{\mathbf{w}}_t - \mathbf{w}_t^k\|^2 + 2\alpha_t^2 \| \sum_{k=1}^{N} p_k \tilde{\mathbf{H}}_t^k(\mathbf{w}_t^k - \overline{\mathbf{w}}_t) \|^2 - 2\alpha_t \sum_{k=1}^{N} p_k F_k(\mathbf{w}_t^k)$$

following Ma et al. (2018) we first prove that

$$\mathbb{E}\|\overline{\mathbf{w}}_t - \mathbf{w}^*\|^2 - \alpha_t \sum_{k=1}^{N} p_k \langle (\overline{\mathbf{w}}_t - \mathbf{w}^*), \mathbf{H}^k(\overline{\mathbf{w}}_t - \mathbf{w}^*) \rangle + 2\alpha_t^2 \| \sum_{k=1}^{N} p_k \tilde{\mathbf{H}}_t^k(\overline{\mathbf{w}}_t - \mathbf{w}^*) \|^2$$

$$\leq (1 - \frac{N}{8(\nu_{\max}\kappa_1 + (N - \nu_{\min}))})\mathbb{E}\|\overline{\mathbf{w}}_t - \mathbf{w}^*\|^2$$

with appropriately chosen $\alpha_t$. Compared to the rate $O(\frac{\mu N}{l\nu_{\max} + L(N - \nu_{\min})}) = O(\frac{N}{\nu_{\max}\kappa_1 + (N - \nu_{\min})\kappa})$ for general strongly convex and smooth objectives, this is an improvement as linear speedup is now available for a larger range of $N$.

We have

$$\mathbb{E}_t \| \sum_{k=1}^{N} p_k \tilde{\mathbf{H}}_t^k(\overline{\mathbf{w}}_t - \mathbf{w}^*) \|^2$$

$$= \mathbb{E}_t \langle \sum_{k=1}^{N} p_k \tilde{\mathbf{H}}_t^k(\overline{\mathbf{w}}_t - \mathbf{w}^*), \sum_{k=1}^{N} p_k \tilde{\mathbf{H}}_t^k(\overline{\mathbf{w}}_t - \mathbf{w}^*) \rangle$$

$$= \sum_{k=1}^{N} p_k^2 \mathbb{E}_t \|\tilde{\mathbf{H}}_t^k(\overline{\mathbf{w}}_t - \mathbf{w}^*)\|^2 + \sum_{k=1}^{N} \sum_{j \neq k} p_j p_k \mathbb{E}_t \langle \tilde{\mathbf{H}}_t^k(\overline{\mathbf{w}}_t - \mathbf{w}^*), \tilde{\mathbf{H}}_t^j(\overline{\mathbf{w}}_t - \mathbf{w}^*) \rangle$$

$$= \sum_{k=1}^{N} p_k^2 \mathbb{E}_t \|\tilde{\mathbf{H}}_t^k(\overline{\mathbf{w}}_t - \mathbf{w}^*)\|^2 + \sum_{k=1}^{N} \sum_{j \neq k} p_j p_k \mathbb{E}_t \langle \mathbf{H}^k(\overline{\mathbf{w}}_t - \mathbf{w}^*), \mathbf{H}^j(\overline{\mathbf{w}}_t - \mathbf{w}^*) \rangle$$

$$= \sum_{k=1}^{N} p_k^2 \mathbb{E}_t \|\tilde{\mathbf{H}}_t^k(\overline{\mathbf{w}}_t - \mathbf{w}^*)\|^2 + \sum_{k=1}^{N} \sum_{j=1}^{N} p_j p_k \mathbb{E}_t \langle \mathbf{H}^k(\overline{\mathbf{w}}_t - \mathbf{w}^*), \mathbf{H}^j(\overline{\mathbf{w}}_t - \mathbf{w}^*) \rangle - \sum_{k=1}^{N} p_k^2 \|\mathbf{H}^k(\overline{\mathbf{w}}_t - \mathbf{w}^*)\|^2$$

$$= \sum_{k=1}^{N} p_k^2 \mathbb{E}_t \|\tilde{\mathbf{H}}_t^k(\overline{\mathbf{w}}_t - \mathbf{w}^*)\|^2 + \|\sum_k p_k \mathbf{H}^k(\overline{\mathbf{w}}_t - \mathbf{w}^*)\|^2 - \sum_{k=1}^{N} p_k^2 \|\mathbf{H}^k(\overline{\mathbf{w}}_t - \mathbf{w}^*)\|^2$$

$$\leq \sum_{k=1}^{N} p_k^2 \mathbb{E}_t \|\tilde{\mathbf{H}}_t^k(\overline{\mathbf{w}}_t - \mathbf{w}^*)\|^2 + \|\sum_k p_k \mathbf{H}^k(\overline{\mathbf{w}}_t - \mathbf{w}^*)\|^2 - \frac{1}{N} \nu_{\min} \|\sum_k p_k \mathbf{H}^k(\overline{\mathbf{w}}_t - \mathbf{w}^*)\|^2$$

$$\leq \frac{1}{N} \nu_{\max} \sum_{k=1}^{N} p_k \mathbb{E}_t \|\tilde{\mathbf{H}}_t^k(\overline{\mathbf{w}}_t - \mathbf{w}^*)\|^2 + (1 - \frac{1}{N} \nu_{\min}) \|\sum_k p_k \mathbf{H}^k(\overline{\mathbf{w}}_t - \mathbf{w}^*)\|^2$$

$$\leq \frac{1}{N} \nu_{\max} l \sum_{k=1}^{N} p_k \langle (\overline{\mathbf{w}}_t - \mathbf{w}^*), \mathbf{H}^k(\overline{\mathbf{w}}_t - \mathbf{w}^*) \rangle + (1 - \frac{1}{N} \nu_{\min}) \|\sum_k p_k \mathbf{H}^k(\overline{\mathbf{w}}_t - \mathbf{w}^*)\|^2$$

$$= \frac{1}{N} \nu_{\max} l \langle (\overline{\mathbf{w}}_t - \mathbf{w}^*), \mathbf{H}(\overline{\mathbf{w}}_t - \mathbf{w}^*) \rangle + (1 - \frac{1}{N} \nu_{\min}) \langle \overline{\mathbf{w}}_t - \mathbf{w}^*, \mathbf{H}^2(\overline{\mathbf{w}}_t - \mathbf{w}^*) \rangle$$

using $\|\tilde{\mathbf{H}}_t^k\| \leq l$.

Now we have

$$\mathbb{E}\|\overline{\mathbf{w}}_t - \mathbf{w}^*\|^2 - \alpha_t \sum_{k=1}^{N} p_k \langle (\overline{\mathbf{w}}_t - \mathbf{w}^*), \mathbf{H}^k(\overline{\mathbf{w}}_t - \mathbf{w}^*) \rangle + 2\alpha_t^2 \|\sum_{k=1}^{N} p_k \tilde{\mathbf{H}}_t^k(\overline{\mathbf{w}}_t - \mathbf{w}^*)\|^2 =$$

$$\langle \overline{\mathbf{w}}_t - \mathbf{w}^*, (I - \alpha_t \mathbf{H} + 2\alpha_t^2 (\frac{\nu_{\max} l}{N} \mathbf{H} + \frac{N - \nu_{\min}}{N} \mathbf{H}^2))(\overline{\mathbf{w}}_t - \mathbf{w}^*) \rangle$$

and it remains to bound the maximum eigenvalue of

$$(I - \alpha_t \mathbf{H} + 2\alpha_t^2 (\frac{\nu_{\max} l}{N} \mathbf{H} + \frac{N - \nu_{\min}}{N} \mathbf{H}^2))$$

and we bound this following Ma et al. (2018). If we choose $\alpha_t < \frac{N}{2(\nu_{\max} l + (N - \nu_{\min})L)}$, then

$$-\alpha_t \mathbf{H} + 2\alpha_t^2 (\frac{\nu_{\max} l}{N} \mathbf{H} + \frac{N - \nu_{\min}}{N} \mathbf{H}^2) \prec 0$$

and the convergence rate is given by the maximum of $1 - \alpha_t \lambda + 2\alpha_t^2 (\frac{\nu_{\max} l}{N} \lambda + \frac{N - \nu_{\min}}{N} \lambda^2)$ maximized over the non-zero eigenvalues $\lambda$ of $\mathbf{H}$. To select the step size $\alpha_t$ that gives the smallest upper bound, we then minimize over $\alpha_t$, resulting in

$$\min_{\alpha_t < \frac{N}{2(\nu_{\max} l + (N - \nu_{\min})L)}} \max_{\lambda > 0 : \exists v, \mathbf{H}v = \lambda v} \left\{ 1 - \alpha_t \lambda + 2\alpha_t^2 (\frac{\nu_{\max} l}{N} \lambda + \frac{N - \nu_{\min}}{N} \lambda^2) \right\}$$

Since the objective is quadratic in $\lambda$, the maximum is achieved at either the largest eigenvalue $\lambda_{\max}$ of $\mathbf{H}$ or the smallest non-zero eigenvalue $\lambda_{\min}$ of $\mathbf{H}$.

When $N \leq \frac{4\nu_{\max} l}{L - \lambda_{\min}} + 4\nu_{\min}$, i.e. when $N = O(l/\lambda_{\min}) = O(\kappa_1)$, the optimal objective value is achieved at $\lambda_{\min}$ and the optimal step size is given by $\alpha_t = \frac{N}{4(\nu_{\max} l + (N - \nu_{\min})\lambda_{\min})}$. The optimal convergence rate (i.e. the optimal objective value) is equal to $1 - \frac{1}{8} \frac{N\lambda_{\min}}{(\nu_{\max} l + (N - \nu_{\min})\lambda_{\min})} = 1 - \frac{1}{8} \frac{N}{(\nu_{\max} \kappa_1 + (N - \nu_{\min}))}$. This implies that when $N = O(\kappa_1)$, the optimal convergence rate has a linear speedup in $N$. When $N$ is larger, this step size is no longer optimal, but we still have $1 - \frac{1}{8} \frac{N}{(\nu_{\max} \kappa_1 + (N - \nu_{\min}))}$ as an upper bound on the convergence rate.

Now we have proved

$$\mathbb{E}\|\overline{\mathbf{w}}_{t+1} - \mathbf{w}^*\|^2 \leq (1 - \frac{1}{8}\frac{N}{(\nu_{\max}\kappa_1 + (N - \nu_{\min}))})\mathbb{E}\|\overline{\mathbf{w}}_t - \mathbf{w}^*\|^2$$

$$+ \alpha_t L \sum_{k=1}^{N} p_k \|\overline{\mathbf{w}}_t - \mathbf{w}_t^k\|^2 + 2\alpha_t^2 \|\sum_{k=1}^{N} p_k \tilde{\mathbf{H}}_t^k(\mathbf{w}_t^k - \overline{\mathbf{w}}_t)\|^2 - 2\alpha_t \sum_{k=1}^{N} p_k F_k(\mathbf{w}_t^k)$$

Next we bound terms in the second line using a similar argument as the general case. We have

$$2\alpha_t^2 \|\sum_{k=1}^{N} p_k \tilde{\mathbf{H}}_t^k(\mathbf{w}_t^k - \overline{\mathbf{w}}_t)\|^2 \leq 2\alpha_t^2 l^2 \sum_{k=1}^{N} p_k \|\overline{\mathbf{w}}_t - \mathbf{w}_t^k\|^2$$

and

$$\mathbb{E}\sum_{k=1}^{N} p_k \|\overline{\mathbf{w}}_t - \mathbf{w}_t^k\|^2 \leq \mathbb{E}2(1 + 2l^2\alpha_{t-1}^2)\sum_{k=1}^{N} p_k \|\overline{\mathbf{w}}_{t-1} - \mathbf{w}_{t-1}^k\|^2 + 8\alpha_{t-1}^2 lF(\overline{\mathbf{w}}_{t-1})$$

$$= 4\alpha_{t-1}^2 l\langle\overline{\mathbf{w}}_{t-1} - \mathbf{w}^*, \mathbf{H}(\overline{\mathbf{w}}_{t-1} - \mathbf{w}^*)\rangle$$

and if $\alpha_t, \alpha_{t-1}$ satisfy

$$\alpha_t L + 2\alpha_t^2 \leq (1 - \frac{1}{8}\frac{N}{(\nu_{\max}\kappa_1 + (N - \nu_{\min}))})(\alpha_{t-1}L + 2\alpha_{t-1}^2)/3$$

$$2(1 + 2l^2\alpha_{t-1}^2) \leq 3$$

$$\alpha_t L + 2\alpha_t^2 \leq 1$$

we have

$$\mathbb{E}\|\overline{\mathbf{w}}_{t+1} - \mathbf{w}^*\|^2$$

$$\leq (1 - \frac{1}{8}\frac{N}{(\nu_{\max}\kappa_1 + (N - \nu_{\min}))})[\mathbb{E}\|\overline{\mathbf{w}}_{t-1} - \mathbf{w}^*\|^2 - \alpha_t\langle\overline{\mathbf{w}}_{t-1} - \mathbf{w}^*, \mathbf{H}\overline{\mathbf{w}}_{t-1} - \mathbf{w}^*\rangle + 2\alpha_t^2\|\sum_{k=1}^{N} p_k\tilde{\mathbf{H}}_t^k(\overline{\mathbf{w}}_t - \mathbf{w}^*)\|^2$$

$$+ (\alpha_{t-1}L + 2\alpha_{t-1}^2) \cdot 2\sum_{k=1}^{N} p_k\|\overline{\mathbf{w}}_{t-1} - \mathbf{w}_{t-1}^k\|^2 + 4\alpha_{t-1}^2 l\langle\overline{\mathbf{w}}_{t-1} - \mathbf{w}^*, \mathbf{H}(\overline{\mathbf{w}}_{t-1} - \mathbf{w}^*)\rangle]$$

and again by choosing $\alpha_{t-1} = c\frac{N}{8(\nu_{\max}l + (N - \nu_{\min})\lambda_{\min})}$ for a small constant $c$, we can guarantee that

$$\mathbb{E}\|\overline{\mathbf{w}}_{t-1} - \mathbf{w}^*\|^2 - \alpha_{t-1}\langle\overline{\mathbf{w}}_{t-1} - \mathbf{w}^*, \mathbf{H}\overline{\mathbf{w}}_{t-1} - \mathbf{w}^*\rangle$$

$$+ 2\alpha_{t-1}^2\|\sum_{k=1}^{N} p_k\tilde{\mathbf{H}}_{t-1}^k(\overline{\mathbf{w}}_{t-1} - \mathbf{w}^*)\|^2 + 4\alpha_{t-1}^2 l\langle\overline{\mathbf{w}}_{t-1} - \mathbf{w}^*, \mathbf{H}(\overline{\mathbf{w}}_{t-1} - \mathbf{w}^*)\rangle$$

$$\leq (1 - c\frac{N}{16(\nu_{\max}l + (N - \nu_{\min})\lambda_{\min})})\mathbb{E}\|\overline{\mathbf{w}}_{t-1} - \mathbf{w}^*\|^2$$

For general $E$, we have the recursive relation

$$\mathbb{E}\|\overline{\mathbf{w}}_{t+1} - \mathbf{w}^*\|^2 \leq \mathbb{E}(1 - c\frac{1}{8}\frac{N}{(\nu_{\max}\kappa_1 + (N - \nu_{\min}))}) \cdots (1 - c\frac{1}{8\tau}\frac{N}{(\nu_{\max}\kappa_1 + (N - \nu_{\min}))})[\|\overline{\mathbf{w}}_{t-\tau} - \mathbf{w}^*\|^2$$

$$- \alpha_{t-\tau}\langle\overline{\mathbf{w}}_{t-\tau} - \mathbf{w}^*, \mathbf{H}\overline{\mathbf{w}}_{t-\tau} - \mathbf{w}^*\rangle + 2\alpha_{t-\tau}^2\|\sum_{k=1}^{N} p_k\tilde{\mathbf{H}}_{t-\tau}^k(\overline{\mathbf{w}}_{t-\tau} - \mathbf{w}^*)\|^2$$

$$+ 4\tau\alpha_{t-1}^2 l\langle\overline{\mathbf{w}}_{t-1} - \mathbf{w}^*, \mathbf{H}(\overline{\mathbf{w}}_{t-1} - \mathbf{w}^*)\rangle$$

$$+ (2\alpha_{t-\tau}^2 l^2 + \alpha_{t-\tau}L)((\tau + 1)\sum_{k=1}^{N} p_k\|\overline{\mathbf{w}}_{t-\tau} - \mathbf{w}_{t-\tau}^k\|^2)]$$

as long as the step sizes are chosen $\alpha_{t-\tau} = c\frac{N}{4\tau(\nu_{\max}l + (N - \nu_{\min})\lambda_{\min})}$ such that the following inequalities hold

$$(2\alpha_{t-\tau}^2 l^2 + \alpha_{t-\tau}L) \leq (1 - \alpha_{t-\tau}\mu)(2\alpha_{t-\tau-1}^2 l^2 + \alpha_{t-\tau-1}L)/3$$

$$2(1 + 2l^2\alpha_{t-\tau}^2) \le 3$$
$$2\alpha_{t-\tau}^2 l^2 + \alpha_{t-\tau}L \le 1$$

and

$$\|\overline{\mathbf{w}}_{t-\tau} - \mathbf{w}^*\|^2 - \alpha_{t-\tau}\langle\overline{\mathbf{w}}_{t-\tau} - \mathbf{w}^*, \mathbf{H}\overline{\mathbf{w}}_{t-\tau} - \mathbf{w}^*\rangle$$

$$+ 2\alpha_{t-\tau}^2\|\sum_{k=1}^N p_k\tilde{\mathbf{H}}_{t-\tau}^k(\overline{\mathbf{w}}_{t-\tau} - \mathbf{w}^*)\|^2 + 4\tau\alpha_{t-1}^2 l\langle\overline{\mathbf{w}}_{t-1} - \mathbf{w}^*, \mathbf{H}(\overline{\mathbf{w}}_{t-1} - \mathbf{w}^*)\rangle$$

$$\le (1 - c\frac{N}{8(\tau+1)(\nu_{\max}\kappa_1 + (N - \nu_{\min}))})\mathbb{E}\|\overline{\mathbf{w}}_{t-\tau} - \mathbf{w}^*\|^2$$

which gives

$$\mathbb{E}\|\overline{\mathbf{w}}_t - \mathbf{w}^*\|^2 \le (1 - c\frac{1}{8E}\frac{N}{(\nu_{\max}\kappa_1 + (N - \nu_{\min}))})^t\|\mathbf{w}_0 - \mathbf{w}^*\|^2$$

$$= O(\exp(-\frac{1}{E}\frac{N}{(\nu_{\max}\kappa_1 + (N - \nu_{\min}))}t))\|\mathbf{w}_0 - \mathbf{w}^*\|^2$$

and with partial participation, the same bound holds with a possibly different choice of $c$. $\qquad\square$

### H.3 GEOMETRIC CONVERGENCE OF FEDMASS FOR OVERPARAMETERIZED LINEAR REGRESSION

**Theorem 7.** *For the overparamterized linear regression problem, FedMaSS with communication every $E$ iterations and constant step sizes $\overline{\eta}_1 = \mathcal{O}(\frac{1}{E}\frac{N}{l\nu_{\max} + \mu(N - \nu_{\min})}), \overline{\eta}_2 = \frac{\overline{\eta}_1(1 - \frac{1}{\tilde{\kappa}})}{1 + \frac{1}{\sqrt{\kappa_1\tilde{\kappa}}}}, \overline{\gamma} = \frac{1 - \frac{1}{\sqrt{\kappa_1\tilde{\kappa}}}}{1 + \frac{1}{\sqrt{\kappa_1\tilde{\kappa}}}}$ has geometric convergence:*

$$\mathbb{E}F(\overline{\mathbf{w}}_T) \le \mathcal{O}\left(L\exp(-\frac{NT}{E(\nu_{\max}\sqrt{\kappa_1\tilde{\kappa}} + (N - \nu_{\min}))})\|\mathbf{w}_0 - \mathbf{w}^*\|^2\right).$$

*Proof.* The proof is based on results in Liu & Belkin (2020) which originally proposed the MaSS algorithm. Note that the update can equivalently be written as

$$\mathbf{v}_{t+1}^k = (1 - \alpha^k)\mathbf{v}_t^k + \alpha^k\mathbf{u}_t^k - \delta^k\mathbf{g}_{t,k}$$

$$\mathbf{w}_{t+1}^k = \begin{cases} \mathbf{u}_t^k - \eta^k\mathbf{g}_{t,k} & \text{if } t+1 \notin \mathcal{I}_E \\ \sum_{k=1}^N p_k[\mathbf{u}_t^k - \eta^k\mathbf{g}_{t,k}] & \text{if } t+1 \in \mathcal{I}_E \end{cases}$$

$$\mathbf{u}_{t+1}^k = \frac{\alpha^k}{1 + \alpha^k}\mathbf{v}_{t+1}^k + \frac{1}{1 + \alpha^k}\mathbf{w}_{t+1}^k$$

where there is a bijection between the parameters $\frac{1 - \alpha^k}{1 + \alpha^k} = \gamma^k, \eta^k = \eta_1^k, \frac{\eta^k - \alpha^k\delta^k}{1 + \alpha^k} = \eta_2^k$, and we further introduce an auxiliary parameter $\mathbf{v}_t^k$, which is initialized at $\mathbf{v}_0^k$. We also note that when $\delta^k = \frac{\eta^k}{\alpha^k}$, the update reduces to the Nesterov accelerated SGD. This version of the FedAvg algorithm with local MaSS updates is used for analyzing the geometric convergence.

As before, define the virtual sequences $\overline{\mathbf{w}}_t = \sum_{k=1}^N p_k\mathbf{w}_t^k, \overline{\mathbf{v}}_t = \sum_{k=1}^N p_k\mathbf{v}_t^k, \overline{\mathbf{u}}_t = \sum_{k=1}^N p_k\mathbf{u}_t^k$, and $\overline{\mathbf{g}}_t = \sum_{k=1}^N p_k\mathbb{E}\mathbf{g}_{t,k}$. We have $\mathbb{E}\mathbf{g}_t = \overline{\mathbf{g}}_t$ and $\overline{\mathbf{w}}_{t+1} = \overline{\mathbf{u}}_t - \eta_t\mathbf{g}_t, \overline{\mathbf{v}}_{t+1} = (1 - \alpha^k)\overline{\mathbf{v}}_t + \alpha^k\overline{\mathbf{w}}_t - \delta^k\mathbf{g}_t$, and $\overline{\mathbf{u}}_{t+1} = \frac{\alpha^k}{1 + \alpha^k}\overline{\mathbf{v}}_{t+1} + \frac{1}{1 + \alpha^k}\overline{\mathbf{w}}_{t+1}$.

We first prove the theorem with $E = 2$ and $t - 1$ being a communication round. We have

$$\|\overline{\mathbf{v}}_{t+1} - \mathbf{w}^*\|_{\mathbf{H}^{-1}}^2$$

$$= \|(1 - \alpha)\overline{\mathbf{v}}_t + \alpha\overline{\mathbf{u}}_t - \delta\sum_k p_k\tilde{\mathbf{H}}_t^k(\mathbf{u}_t^k - \mathbf{w}^*) - \mathbf{w}^*\|_{\mathbf{H}^{-1}}^2$$

$$= \|(1 - \alpha)\overline{\mathbf{v}}_t + \alpha\overline{\mathbf{u}}_t - \mathbf{w}^*\|_{\mathbf{H}^{-1}}^2 + \delta^2\|\sum_k p_k\tilde{\mathbf{H}}_t^k(\mathbf{u}_t^k - \mathbf{w}^*)\|_{\mathbf{H}^{-1}}^2$$

$$- 2\delta\langle\sum_k p_k\tilde{\mathbf{H}}_t^k(\mathbf{u}_t^k - \mathbf{w}^*), (1-\alpha)\overline{\mathbf{v}}_t + \alpha\overline{\mathbf{u}}_t - \mathbf{w}^*\rangle_{\mathbf{H}^{-1}}$$

$$\leq \underbrace{\|(1-\alpha)\overline{\mathbf{v}}_t + \alpha\overline{\mathbf{u}}_t - \mathbf{w}^*\|_{\mathbf{H}^{-1}}^2}_{A} + \underbrace{2\delta^2\|\sum_k p_k\tilde{\mathbf{H}}_t^k(\overline{\mathbf{u}}_t - \mathbf{w}^*)\|_{\mathbf{H}^{-1}}^2}_{B} + 2\delta^2\|\sum_k p_k\tilde{\mathbf{H}}_t^k(\overline{\mathbf{u}}_t - \mathbf{u}_t^k)\|_{\mathbf{H}^{-1}}^2$$

$$\underbrace{- 2\delta\langle\sum_k p_k\tilde{\mathbf{H}}_t^k(\overline{\mathbf{u}}_t - \mathbf{w}^*), (1-\alpha)\overline{\mathbf{v}}_t + \alpha\overline{\mathbf{u}}_t - \mathbf{w}^*\rangle_{\mathbf{H}^{-1}}}_{C}$$

$$- 2\delta\langle\sum_k p_k\tilde{\mathbf{H}}_t^k(\mathbf{u}_t^k - \overline{\mathbf{u}}_t), (1-\alpha)\overline{\mathbf{v}}_t + \alpha\overline{\mathbf{u}}_t - \mathbf{w}^*\rangle_{\mathbf{H}^{-1}}$$

Following the proof in Liu & Belkin (2020),

$$\mathbb{E}A \leq \mathbb{E}(1-\alpha)\|\overline{\mathbf{v}}_t - \mathbf{w}^*\|_{\mathbf{H}^{-1}}^2 + \alpha\|\overline{\mathbf{u}}_t - \mathbf{w}^*\|_{\mathbf{H}^{-1}}^2$$

$$\leq \mathbb{E}(1-\alpha)\|\overline{\mathbf{v}}_t - \mathbf{w}^*\|_{\mathbf{H}^{-1}}^2 + \frac{\alpha}{\mu}\|\overline{\mathbf{u}}_t - \mathbf{w}^*\|^2$$

using the convexity of the norm $\|\cdot\|_{\mathbf{H}^{-1}}$ and that $\mu$ is the smallest non-zero eigenvalue of $H$.

Now

$$\mathbb{E}B \leq 2\delta^2(\nu_{\max}\frac{1}{N}\tilde{\kappa} + \frac{N - \nu_{\min}}{N})\|(\overline{\mathbf{u}}_t - \mathbf{w}^*)\|_H^2$$

using the folowing bound:

$$\mathbb{E}\left(\sum_k p_k\tilde{\mathbf{H}}_t^k\right)\mathbf{H}^{-1}\left(\sum_k p_k\tilde{\mathbf{H}}_t^k\right) = \mathbb{E}\sum_k p_k^2\tilde{\mathbf{H}}_t^k\mathbf{H}^{-1}\tilde{\mathbf{H}}_t^k + \sum_{k\neq j}p_kp_j\tilde{\mathbf{H}}_t^k\mathbf{H}^{-1}\tilde{\mathbf{H}}_t^j$$

$$\preceq \nu_{\max}\frac{1}{N}\mathbb{E}\sum_k p_k\tilde{\mathbf{H}}_t^k\mathbf{H}^{-1}\tilde{\mathbf{H}}_t^k + \sum_{k\neq j}p_kp_j\mathbf{H}^k\mathbf{H}^{-1}\mathbf{H}^j$$

$$= \nu_{\max}\frac{1}{N}\mathbb{E}\sum_k p_k\tilde{\mathbf{H}}_t^k\mathbf{H}^{-1}\tilde{\mathbf{H}}_t^k + \sum_{k,j}p_kp_j\mathbf{H}^k\mathbf{H}^{-1}\mathbf{H}^j - \sum_k p_k^2\mathbf{H}^k\mathbf{H}^{-1}\mathbf{H}^k$$

$$\preceq \nu_{\max}\frac{1}{N}\mathbb{E}\sum_k p_k\tilde{\mathbf{H}}_t^k\mathbf{H}^{-1}\tilde{\mathbf{H}}_t^k + \mathbf{H} - \frac{1}{N}\nu_{\min}\sum_k p_k\mathbf{H}^k\mathbf{H}^{-1}\mathbf{H}^k$$

$$\preceq \nu_{\max}\frac{1}{N}\mathbb{E}\sum_k p_k\tilde{\mathbf{H}}_t^k\mathbf{H}^{-1}\tilde{\mathbf{H}}_t^k + \mathbf{H} - \frac{1}{N}\nu_{\min}(\sum_k p_k\mathbf{H}^k)\mathbf{H}^{-1}(\sum_k p_k\mathbf{H}^k)$$

$$= \nu_{\max}\frac{1}{N}\mathbb{E}\sum_k p_k\tilde{\mathbf{H}}_t^k\mathbf{H}^{-1}\tilde{\mathbf{H}}_t^k + \frac{N - \nu_{\min}}{N}\mathbf{H}$$

$$\preceq \nu_{\max}\frac{1}{N}\tilde{\kappa}\mathbf{H} + \frac{N - \nu_{\min}}{N}\mathbf{H}$$

where we have used $\mathbb{E}\sum_k p_k\tilde{\mathbf{H}}_t^k\mathbf{H}^{-1}\tilde{\mathbf{H}}_t^k \leq \tilde{\kappa}\mathbf{H}$ by definition of $\tilde{\kappa}$ and the operator convexity of the mapping $W \to W\mathbf{H}^{-1}W$.

Finally,

$$\mathbb{E}C = -\mathbb{E}2\delta\langle\sum_k p_k\tilde{\mathbf{H}}_t^k(\overline{\mathbf{u}}_t - \mathbf{w}^*), (1-\alpha)\overline{\mathbf{v}}_t + \alpha\overline{\mathbf{u}}_t - \mathbf{w}^*\rangle_{\mathbf{H}^{-1}}$$

$$= -2\delta\langle\sum_k p_k\mathbf{H}^k(\overline{\mathbf{u}}_t - \mathbf{w}^*), (1-\alpha)\overline{\mathbf{v}}_t + \alpha\overline{\mathbf{u}}_t - \mathbf{w}^*\rangle_{\mathbf{H}^{-1}}$$

$$= -2\delta\langle(\overline{\mathbf{u}}_t - \mathbf{w}^*), (1-\alpha)\overline{\mathbf{v}}_t + \alpha\overline{\mathbf{u}}_t - \mathbf{w}^*\rangle$$

$$= -2\delta\langle(\overline{\mathbf{u}}_t - \mathbf{w}^*), \overline{\mathbf{u}}_t - \mathbf{w}^* + \frac{1-\alpha}{\alpha}(\overline{\mathbf{u}}_t - \overline{\mathbf{w}}_t)\rangle$$

$$= -2\delta\|\overline{\mathbf{u}}_t - \mathbf{w}^*\|^2 + \frac{1-\alpha}{\alpha}\delta(\|\overline{\mathbf{w}}_t - \mathbf{w}^*\|^2 - \|\overline{\mathbf{u}}_t - \mathbf{w}^*\|^2 - \|\overline{\mathbf{w}}_t - \overline{\mathbf{u}}_t\|^2)$$

$$\leq \frac{1-\alpha}{\alpha}\delta\|\overline{\mathbf{w}}_t - \mathbf{w}^*\|^2 - \frac{1-\alpha}{\alpha}\delta\|\overline{\mathbf{u}}_t - \mathbf{w}^*\|^2$$

where we have used

$$
\begin{aligned}
(1-\alpha)&\overline{\mathbf{v}}_t + \alpha\overline{\mathbf{u}}_t \\
&= (1-\alpha)((1+\alpha)\overline{\mathbf{u}}_t - \overline{\mathbf{w}}_t)/\alpha + \alpha\overline{\mathbf{u}}_t \\
&= \frac{1}{\alpha}\overline{\mathbf{u}}_t - \frac{1-\alpha}{\alpha}\overline{\mathbf{w}}_t
\end{aligned}
$$

and the identity that $-2\langle \mathbf{a}, \mathbf{b}\rangle = \|\mathbf{a}\|^2 + \|\mathbf{b}\|^2 - \|\mathbf{a}+\mathbf{b}\|^2$.

It follows that

$$
\begin{aligned}
\mathbb{E}\|&\overline{\mathbf{v}}_{t+1} - \mathbf{w}^*\|_{\mathbf{H}^{-1}}^2 \\
&\leq (1-\alpha)\|\overline{\mathbf{v}}_t - \mathbf{w}^*\|_{\mathbf{H}^{-1}}^2 + \frac{1-\alpha}{\alpha}\delta\|\overline{\mathbf{w}}_t - \mathbf{w}^*\|^2 \\
&+ (\frac{\alpha}{\mu} - \frac{1-\alpha}{\alpha}\delta)\|\overline{\mathbf{u}}_t - \mathbf{w}^*\|^2 + 2\delta^2(\nu_{\max}\frac{1}{N}\tilde{\kappa} + \frac{N-\nu_{\min}}{N})\|(\overline{\mathbf{u}}_t - \mathbf{w}^*)\|_H^2 \\
&+ 2\delta^2\|\sum_k p_k\tilde{\mathbf{H}}_t^k(\overline{\mathbf{u}}_t - \mathbf{u}_t^k)\|_{\mathbf{H}^{-1}}^2 \\
&- 2\delta\langle\sum_k p_k\tilde{\mathbf{H}}_t^k(\mathbf{u}_t^k - \overline{\mathbf{u}}_t), (1-\alpha)\overline{\mathbf{v}}_t + \alpha\overline{\mathbf{u}}_t - \mathbf{w}^*\rangle_{\mathbf{H}^{-1}}
\end{aligned}
$$

On the other hand,

$$
\begin{aligned}
\mathbb{E}\|\overline{\mathbf{w}}_{t+1} - \mathbf{w}^*\|^2 &= \mathbb{E}\|\overline{\mathbf{u}}_t - \mathbf{w}^* - \eta\sum_k p_k\tilde{\mathbf{H}}_t^k(\overline{\mathbf{u}}_t - \mathbf{w}^*)\|^2 \\
&= \mathbb{E}\|\overline{\mathbf{u}}_t - \mathbf{w}^*\|^2 - 2\eta\|\overline{\mathbf{u}}_t - \mathbf{w}^*\|_H^2 + \eta^2\|\sum_k p_k\tilde{\mathbf{H}}_t^k(\overline{\mathbf{u}}_t - \mathbf{w}^*)\|^2 \\
&\leq \mathbb{E}\|\overline{\mathbf{u}}_t - \mathbf{w}^*\|^2 - 2\eta\|\overline{\mathbf{u}}_t - \mathbf{w}^*\|_H^2 + \eta^2(\nu_{\max}\frac{1}{N}\ell + L\frac{N-\nu_{\min}}{N})\|\overline{\mathbf{u}}_t - \mathbf{w}^*\|^2
\end{aligned}
$$

where we use the following bound:

$$
\begin{aligned}
\mathbb{E}&\left(\sum_k p_k\tilde{\mathbf{H}}_t^k\right)\left(\sum_k p_k\tilde{\mathbf{H}}_t^k\right) \\
&= \mathbb{E}\sum_k p_k^2\tilde{\mathbf{H}}_t^k\tilde{\mathbf{H}}_t^k + \sum_{k\neq j} p_k p_j\tilde{\mathbf{H}}_t^k\tilde{\mathbf{H}}_t^j \\
&\preceq \nu_{\max}\frac{1}{N}\mathbb{E}\sum_k p_k\tilde{\mathbf{H}}_t^k\tilde{\mathbf{H}}_t^k + \sum_{k\neq j} p_k p_j\mathbf{H}^k\mathbf{H}^j \\
&= \nu_{\max}\frac{1}{N}\mathbb{E}\sum_k p_k\tilde{\mathbf{H}}_t^k\tilde{\mathbf{H}}_t^k + \sum_{k,j} p_k p_j\mathbf{H}^k\mathbf{H}^j - \sum_k p_k^2\mathbf{H}^k\mathbf{H}^k \\
&\preceq \nu_{\max}\frac{1}{N}\mathbb{E}\sum_k p_k\tilde{\mathbf{H}}_t^k\tilde{\mathbf{H}}_t^k + \mathbf{H}^2 - \frac{1}{N}\nu_{\min}\sum_k p_k\mathbf{H}^k\mathbf{H}^k \\
&\preceq \nu_{\max}\frac{1}{N}\mathbb{E}\sum_k p_k\tilde{\mathbf{H}}_t^k\tilde{\mathbf{H}}_t^k + \mathbf{H}^2 - \frac{1}{N}\nu_{\min}(\sum_k p_k\mathbf{H}^k)(\sum_k p_k\mathbf{H}^k) \\
&= \nu_{\max}\frac{1}{N}\mathbb{E}\sum_k p_k\tilde{\mathbf{H}}_t^k\tilde{\mathbf{H}}_t^k + \frac{N-\nu_{\min}}{N}\mathbf{H}^2 \\
&\preceq \nu_{\max}\frac{1}{N}\ell\mathbf{H} + L\frac{N-\nu_{\min}}{N}\mathbf{H}
\end{aligned}
$$

again using that $W \to W^2$ is operator convex and that $\mathbb{E}\tilde{\mathbf{H}}_t^k\tilde{\mathbf{H}}_t^k \preceq \ell\mathbf{H}^k$ by definition of $\ell$.

Combining the bounds for $\mathbb{E}\|\overline{\mathbf{w}}_{t+1} - \mathbf{w}^*\|^2$ and $\mathbb{E}\|\overline{\mathbf{v}}_{t+1} - \mathbf{w}^*\|^2_{\mathbf{H}^{-1}}$,

$$
\mathbb{E}\frac{\delta}{\alpha}\|\overline{\mathbf{w}}_{t+1} - \mathbf{w}^*\|^2 + \|\overline{\mathbf{v}}_{t+1} - \mathbf{w}^*\|^2_{\mathbf{H}^{-1}}
$$

$$
\leq (1-\alpha)\|\overline{\mathbf{v}}_t - \mathbf{w}^*\|^2_{\mathbf{H}^{-1}} + \frac{1-\alpha}{\alpha}\delta\|\overline{\mathbf{w}}_t - \mathbf{w}^*\|^2 + (\frac{\alpha}{\mu} - \delta)\|\overline{\mathbf{u}}_t - \mathbf{w}^*\|^2
$$

$$
+ (2\delta^2(\nu_{\max}\frac{1}{N}\tilde{\kappa} + \frac{N - \nu_{\min}}{N}) - 2\eta\delta/\alpha + \eta^2\delta(\nu_{\max}\frac{1}{N}l + L\frac{N - \nu_{\min}}{N})/\alpha)\|\overline{\mathbf{u}}_t - \mathbf{w}^*\|^2
$$

$$
+ 2\delta^2\|\sum_k p_k\tilde{\mathbf{H}}^k_t(\overline{\mathbf{u}}_t - \mathbf{u}^k_t)\|^2_{\mathbf{H}^{-1}}
$$

$$
+ \delta L \sum_k p_k\|(\overline{\mathbf{u}}_t - \mathbf{u}^k_t)\|^2_{\mathbf{H}^{-1}}
$$

Following Liu & Belkin (2020) if we choose step sizes so that

$$
\frac{\alpha}{\mu} - \delta \leq 0
$$

$$
2\delta^2(\nu_{\max}\frac{1}{N}\tilde{\kappa} + \frac{N - \nu_{\min}}{N}) - 2\eta\delta/\alpha + \eta^2\delta(\nu_{\max}\frac{1}{N}l + L\frac{N - \nu_{\min}}{N})/\alpha \leq 0
$$

or equivalently

$$
\alpha/\delta \leq \mu
$$

$$
2\alpha\delta(\nu_{\max}\frac{1}{N}\tilde{\kappa} + \frac{N - \nu_{\min}}{N}) + \eta(\eta(\nu_{\max}\frac{1}{N}l + L\frac{N - \nu_{\min}}{N}) - 2) \leq 0
$$

the second and third terms are negative. To optimize the step sizes, note that the two inequalities imply

$$
\alpha^2 \leq \eta(2 - \eta(\nu_{\max}\frac{1}{N}l + L\frac{N - \nu_{\min}}{N}))\mu/2(\nu_{\max}\frac{1}{N}\tilde{\kappa} + \frac{N - \nu_{\min}}{N})
$$

and maximizing the right hand side with respect to $\eta$, which is quadratic, we see that $\eta \equiv 1/(\nu_{\max}\frac{1}{N}l + L\frac{N - \nu_{\min}}{N})$ maximizes the right hand side, with

$$
\alpha \equiv \frac{1}{\sqrt{2(\nu_{\max}\frac{1}{N}\kappa_1 + \kappa\frac{N - \nu_{\min}}{N})(\nu_{\max}\frac{1}{N}\tilde{\kappa} + \frac{N - \nu_{\min}}{N})}}
$$

$$
\delta \equiv \frac{\alpha}{\mu} = \frac{\eta}{\alpha(\nu_{\max}\frac{1}{N}\tilde{\kappa} + \frac{N - \nu_{\min}}{N})}
$$

Note that $\alpha = \frac{1}{\sqrt{2(\nu_{\max}\frac{1}{N}\kappa_1 + \kappa\frac{N - \nu_{\min}}{N})(\nu_{\max}\frac{1}{N}\tilde{\kappa} + \frac{N - \nu_{\min}}{N})}} = O(\frac{N}{\sqrt{\kappa_1\tilde{\kappa}}})$ when $N = O(\min\{\tilde{\kappa}, \kappa_1/\kappa\})$.

Finally, to deal with the terms $2\delta^2\|\sum_k p_k\tilde{\mathbf{H}}^k_t(\overline{\mathbf{u}}_t - \mathbf{u}^k_t)\|^2_{\mathbf{H}^{-1}} + \delta L\sum_k p_k\|(\overline{\mathbf{u}}_t - \mathbf{u}^k_t)\|^2_{\mathbf{H}^{-1}}$, we can use Jensen

$$
2\delta^2\|\sum_k p_k\tilde{\mathbf{H}}^k_t(\overline{\mathbf{u}}_t - \mathbf{u}^k_t)\|^2_{\mathbf{H}^{-1}} + \delta L\sum_k p_k\|(\overline{\mathbf{u}}_t - \mathbf{u}^k_t)\|^2_{\mathbf{H}^{-1}}
$$

$$
\leq (2\delta^2 l^2 + \delta L)\sum_k p_k\|\overline{\mathbf{u}}_t - \mathbf{u}^k_t\|^2_{\mathbf{H}^{-1}}
$$

$$
= (2\delta^2 l^2 + \delta L)\sum_k p_k\|\frac{\alpha}{1+\alpha}\overline{\mathbf{v}}_t + \frac{1}{1+\alpha}\overline{\mathbf{w}}_t - (\frac{\alpha}{1+\alpha}v^k_t + \frac{1}{1+\alpha}w^k_t)\|^2_{\mathbf{H}^{-1}}
$$

$$
\leq (2\delta^2 l^2 + \delta L)(2(\frac{\alpha}{1+\alpha})^2\delta^2 + 2(\frac{1}{1+\alpha})^2\eta^2)\sum_k p_k\|\tilde{\mathbf{H}}^k_{t-1}(\overline{\mathbf{u}}_{t-1} - \mathbf{w}^*)\|^2
$$

$$
\leq (2\delta^2 l^2 + \delta L)(2(\frac{\alpha}{1+\alpha})^2\delta^2 + 2(\frac{1}{1+\alpha})^2\eta^2)l^2\|(\overline{\mathbf{u}}_{t-1} - \mathbf{w}^*)\|^2
$$

which can be combined with the terms with $\|(\overline{\mathbf{u}}_{t-1} - \mathbf{w}^*)\|^2$ in the recursive expansion of $\mathbb{E}\frac{\delta}{\alpha}\|\overline{\mathbf{w}}_t - \mathbf{w}^*\|^2 + \|\overline{\mathbf{v}}_t - \mathbf{w}^*\|_{\mathbf{H}^{-1}}^2$:

$$\mathbb{E}\frac{\delta}{\alpha}\|\overline{\mathbf{w}}_t - \mathbf{w}^*\|^2 + \|\overline{\mathbf{v}}_t - \mathbf{w}^*\|_{\mathbf{H}^{-1}}^2$$

$$\leq (1-\alpha)\|\overline{\mathbf{v}}_{t-1} - \mathbf{w}^*\|_{\mathbf{H}^{-1}}^2 + \frac{1-\alpha}{\alpha}\delta\|\overline{\mathbf{w}}_{t-1} - \mathbf{w}^*\|^2 + (\frac{\alpha}{\mu} - \delta)\|\overline{\mathbf{u}}_{t-1} - \mathbf{w}^*\|^2$$

$$+ (2\delta^2(\nu_{\max}\frac{1}{N}\tilde{\kappa} + \frac{N - \nu_{\min}}{N}) - 2\eta\delta/\alpha + \eta^2\delta(\nu_{\max}\frac{1}{N}l + L\frac{N - \nu_{\min}}{N})/\alpha)\|\overline{\mathbf{u}}_{t-1} - \mathbf{w}^*\|^2$$

and the step sizes can be chosen so that the resulting coefficients are negative. Therefore, we have shown that

$$\mathbb{E}\|\overline{\mathbf{w}}_{t+1} - \mathbf{w}^*\|^2 \leq (1-\alpha)^2\|\overline{\mathbf{w}}_{t-1} - \mathbf{w}^*\|^2$$

where $\alpha = \frac{1}{\sqrt{2(\nu_{\max}\frac{1}{N}\kappa_1 + \kappa\frac{N - \nu_{\min}}{N})(\nu_{\max}\frac{1}{N}\tilde{\kappa} + \frac{N - \nu_{\min}}{N})}} = O(\frac{N}{\nu_{\max}\sqrt{\kappa_1\tilde{\kappa}} + N - \nu_{\min}})$ when $N = O(\min\{\tilde{\kappa}, \kappa_1/\kappa\})$.

For general $E > 1$, choosing $\eta = c/E(\nu_{\max}\frac{1}{N}l + L\frac{N - \nu_{\min}}{N})$ for some small constant $c$ results in $\alpha = O(\frac{1}{E\sqrt{(\nu_{\max}\frac{1}{N}\kappa_1 + \kappa\frac{N - \nu_{\min}}{N})(\nu_{\max}\frac{1}{N}\tilde{\kappa} + \frac{N - \nu_{\min}}{N})}})$ and this guarantees that

$$\mathbb{E}\|\overline{\mathbf{w}}_t - \mathbf{w}^*\|^2 \leq (1-\alpha)^t\|\mathbf{w}_0 - \mathbf{w}^*\|^2$$

for all $t$.

$\square$

# I DETAILS ON EXPERIMENTS AND ADDITIONAL RESULTS

We describe the precise procedure to reproduce the results in this paper. As we mentioned in Section 5, we empirically verified the linear speed up on various convex settings for both FedAvg and its accelerated variants. For all the results, we set random seeds as $0, 1, 2$ and report the best convergence rate across the three folds. For each run, we initialize $\mathbf{w}_0 = \mathbf{0}$ and measure the number of iteration to reach the target accuracy $\epsilon$. We use the small-scale dataset w8a Platt (1998), which consists of $n = 49749$ samples with feature dimension $d = 300$. The label is either positive one or negative one. The dataset has sparse binary features in $\{0, 1\}$. Each sample has 11.15 non-zero feature values out of 300 features on average. We set the batch size equal to four across all experiments. In the next following subsections, we introduce parameter searching in each objective separately.

## I.1 STRONGLY CONVEX OBJECTIVES

We first consider the strongly convex objective function, where we use a regularized binary logistic regression with regularization $\lambda = 1/n \approx 2e - 5$. We evenly distributed on $1, 2, 4, 8, 16, 32$ devices and report the number of iterations/rounds needed to converge to $\epsilon-$accuracy, where $\epsilon = 0.005$. The optimal objective function value $f^*$ is set as $f^* = 0.126433176216545$. This is determined numerically and we follow the setting in Stich (2019). The learning rate is decayed as the $\eta_t = \min(\eta_0, \frac{nc}{1+t})$, where we extensively search the best learning rate $c \in \{2^{-1}c_0, 2^{-2}c_0, c_0, 2c_0, 2^2c_0\}$. In this case, we search the initial learning rate $\eta_0 \in \{1, 32\}$ and $c_0 = 1/8$.

## I.2 CONVEX SMOOTH OBJECTIVES

We also use binary logistic regression without regularization. The setting is almost same as its regularized counter part. We also evenly distributed all the samples on $1, 2, 4, 8, 16, 32$ devices. The figure shows the number of iterations needed to converge to $\epsilon-$accuracy, where $\epsilon = 0.02$. The optiaml objective function value is set as $f^* = 0.11379089057514849$, determined numerically. The learning rate is decayed as the $\eta_t = \min(\eta_0, \frac{nc}{1+t})$, where we extensively search the best learning rate $c \in \{2^{-1}c_0, 2^{-2}c_0, c_0, 2c_0, 2^2c_0\}$. In this case, we search the initial learning rate $\eta_0 \in \{1, 32\}$ and $c_0 = 1/8$.

### I.3 LINEAR REGRESSION

For linear regression, we use the same feature vectors from w8a dataset and generate ground truth $[\mathbf{w}^*, b^*]$ from a multivariate normal distribution with zero mean and standard deviation one. Then we generate label based on $y_i = \mathbf{x}_i^t \mathbf{w}^* + b^*$. This procedure will ensure we satisfy the over-parameterized setting as required in our theorems. We also evenly distributed all the samples on $1, 2, 4, 8, 16, 32$ devices. The figure shows the number of iterations needed to converge to $\epsilon-$accuracy, where $\epsilon = 0.02$. The optiaml objective function value is $f^* = 0$. The learning rate is decayed as the $\eta_t = \min(\eta_0, \frac{nc}{1+t})$, where we extensively search the best learning rate $c \in \{2^{-1}c_0, 2^{-2}c_0, c_0, 2c_0, 2^2c_0\}$. In this case, we search the initial learning rate $\eta_0 \in \{0.1, 0.12\}$ and $c_0 = 1/256$.

### I.4 PARTIAL PARTICIPATION

To examine the linear speedup of FedAvg in partial participation setting, we evenly distributed data on $4, 8, 16, 32, 64, 128$ devices and uniformly sample $50\%$ devices without replacement. All other hyperparameters are the same as previous sections.

### I.5 NESTEROV ACCELERATED FEDAVG

The experiments of Nesterov accelerated FedAvg (the update formula is given as follows) uses the same setting as previous three sections for vanilia FedAvg.

$$\mathbf{y}_{t+1}^k = \mathbf{w}_t^k - \alpha_t \mathbf{g}_{t,k}$$

$$\mathbf{w}_{t+1}^k = \begin{cases} \mathbf{y}_{t+1}^k + \beta_t(\mathbf{y}_{t+1}^k - \mathbf{y}_t^k) & \text{if } t+1 \notin \mathcal{I}_E \\ \sum_{k \in \mathcal{S}_{t+1}} \left(\mathbf{y}_{t+1}^k + \beta_t(\mathbf{y}_{t+1}^k - \mathbf{y}_t^k)\right) & \text{if } t+1 \in \mathcal{I}_E \end{cases}$$

We set $\beta_t = 0.1$ and search $\alpha_t$ in the same way as $\eta_t$ in FedAvg.

### I.6 THE IMPACT OF $E$.

In this subsection, we further examine how does the number of local steps ($E$) affect convergence. As shown in Figure 2, the number of iterations increases as $E$ increase, which slow down the convergence in terms of gradient computation. However, it can save communication costs as the number of rounds decreased when the $E$ increases. This showcase that we need a proper choice of $E$ to trade-off the communication cost and convergence speed.

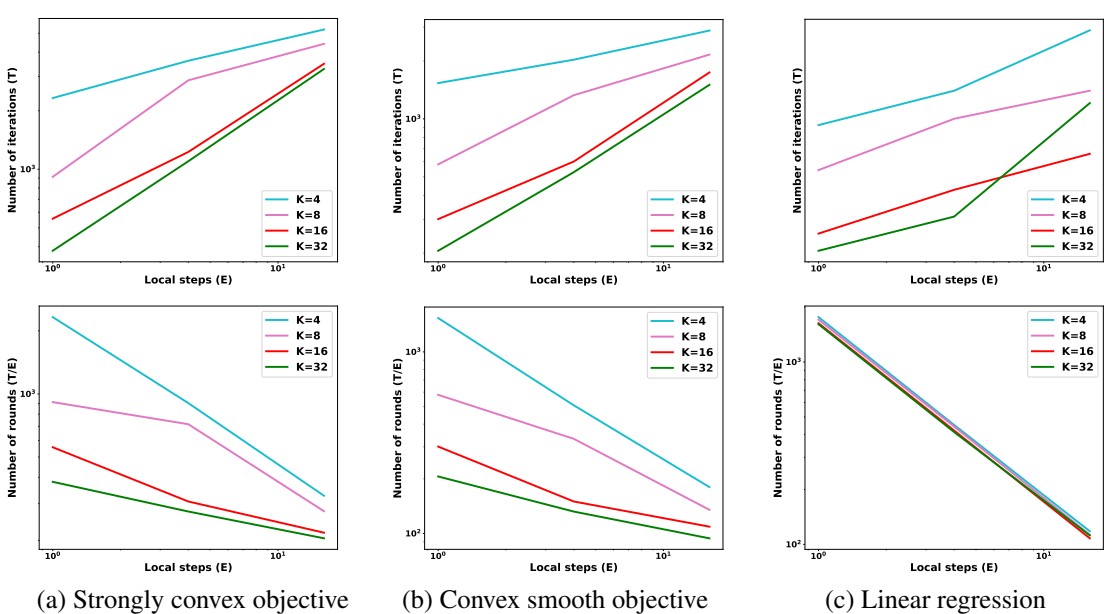

(a) Strongly convex objective     (b) Convex smooth objective     (c) Linear regression

Figure 2: The convergence of FedAvg w.r.t the number of local steps $E$.

