# OpenReview forum: "Federated Learning's Blessing: FedAvg has Linear Speedup"
_ICLR.cc/2021/Conference — Reject_

### Official Review · AnonReviewer4 · 2020-10-28
**Reivew**

**Rating:** 5
**Confidence:** 3

**Review:**

This paper shows a linear speedup in FedAvg w.r.t. number of devices, mainly theoretically, while most prior works ignore it. The main convergence results are given for three cases: a) Strongly Convex+Smooth, b) Convex+Smoth, and c) Strongly/x convex+Smooth+0 training loss can be achieved. The paper is well-written and motivated with good discussions of the algorithm and the related works.

Even though, I still have doubts about this paper:
1).  In the over-parameterized case, this paper shows a linear convergence rate $O(exp(-NT/E\kappa))$. Could you explain how large $\kappa$ would be when width goes infinity (polynomially or exponentially)? Or $\kappa$ is bounded by a universal constant? While I go through Appendix G, I find the proof in the over-parameterized case relies on Strongly Convex+ Smooth and further Assumption that 0 loss can be achieved. However, I don't reckon this is the over-parameterization case, as the over-parameterized functions are, as far as I know, always Non-convex. In my opinion, there is no necessity to connect FedAvg with Over-Parameterization in such a way.

Overall, I think this paper is qualified for acceptance except for the part of the over-parameterized case. I rate my score 5 at first, but consider raising it if the authors dispel my doubts. I'm looking forward to the discussion.

---

> ### Author Response · Authors · 2020-11-13
> **Thank you for the constructive feedback!**
>
> Thank you very much for your valuable and constructive feedback! Our point-to-point responses are provided below.
>
> **1. Condition numbers in the bounds in the overparameterized setting.**
> The condition numbers $\kappa_1$ and $\tilde{\kappa}$ are defined in Section H.2. These quantities are defined in terms of the Hessian and stochastic Hessian matrices at each device and the global Hessian matrix, and are fixed and finite quantities given a particular problem instance. We use the fact that $\tilde{\kappa}\leq \kappa_1$. As the number of participating devices goes to infinity, the condition numbers converge to $\kappa:=L/\mu$, where $L$ and $\mu$ are the Lipschitz and smoothness constants of the global optimization objective. This is observed in e.g. [2]
>
> **2. Relevance of the overparameterized setting.**
> Our results in the overparameterized setting indeed assume strong convexity. While it is true that many overparameterized deep learning systems are non-convex, here by overparameterization we only mean that a global optimizer exists that yields 0 global loss, which does not restrict the problem to be non-convex. This type of overparameterization setting with strong convexity was studied in the works [1,2,3] in the non-FL setting in order to develop provable accelerations over SGD, which motivated us to study extensions to the FL setting. Our main results highlight the geometric convergence with linear speedup in the overparameterized setting, which improves on the general strongly convex and convex settings, and provide a provable acceleration of momentum-based stochastic FedAvg algorithm over the vanilla FedAvg. While overparameterization is an important setting in practice, we agree that the applicability of our results is limited by the strong convexity assumption, and that our presentation of this topic in the main text is limited by constraints on space. We can remove the results in this section and include them in an extended version or a separate work since they are not as closely connected to the non-overparameterized setting, and we believe that they warrant a more thorough treatment and discussion to highlight their importance, which is limited by the current draft.
>
> [1]Prateek Jain, Sham M Kakade, Rahul Kidambi, Praneeth Netrapalli, and Aaron Sidford. Accelerating stochastic gradient descent. InProc. STAT,volume 1050, page 26, 2017.
> [2]Chaoyue Liu and Mikhail Belkin. Accelerating sgd with momentum for over-parameterized learning.ICLR, 2020.
> [3]Siyuan Ma, Raef Bassily, and Mikhail Belkin. The power of interpolation:Understanding the effectiveness of sgd in modern over-parametrized learning.ICML, 2018.

---

### Official Review · AnonReviewer3 · 2020-10-28
**Good work but can be improved**

**Rating:** 6
**Confidence:** 4

**Review:**

This paper gives convergence analysis for FedAvg and its accelerated version under data heterogeneity and system heterogeneity. The main improvement comes from a more careful analysis for one-step descent where the authors make use of the term $\alpha_{t} \sum_{k=1}^{N} p_{k}\left[F_{k}\left(\mathbf{w}^{*}\right)-F_{k}\left(\overline{\mathbf{w}}_{t}\right)\right]$ that is ignored by previous work Li (2020b). The paper focuses on how FedAvg’s convergence scales with the number of participating devices. It improves previous analysis for FedAvg under more federated settings and shows that FedAvg has linear speedup for any number of participating devices.

Pros:
1. The paper is well written and easy to follow. It is very appreciated that the authors analyze both strongly convex and general convex cases. All proof seems correct (though I didn’t check very carefully).
2. They summarize existing convergence results for FL algorithms in Table 2 of Appendix B, which helps comparison and I appreciate a lot, though some are missed.

Cons and discussions:
To extend the results of full device participation to partial device participation, one can use the technique from Li (2020b), which is also pointed out by Haddadpour & Mahdavi (2019). So I will focus on the innovative part.
1. The descanting lemmas are quite novel. In particular, both Lemma 6 and Lemma 3 are descending lemmas for FedAvg, except that the former is for (general) convex while the latter is for strongly convex. However, based on their bounds, it seems that the result of Lemma 6 is much stronger than Lemma 3 since $\alpha_{t}\left(F\left(\overline{\mathbf{w}}_{t}\right)-F\left(\mathbf{w}^{*}\right)\right)$ is always non-negative. This is counterintuitive since Lemma 6 requires a much weaker condition (where strong convexity is not required) but has stronger results. I feel that Lemma 3 seems redundant.
2. There are other papers related to the discussed topic [2-3]. I think it is better to cite them since they give a better understanding of FedAvg for full device participation. The state-of-the-art analysis of Local SGD is given in [2-3]. However, when the case is reduced to full device participation, it is unlikely for Theorem 1-2 to cover them, which implies the analysis is somewhat weak.
3. The motivation to formulate Nesterov-accelerated FedAvg seems unclear. From theorems and experiments, it seems that the accelerated FedAvg has no advantage over FedAvg even in the full participation setting. This is strange since intuitively the accelerated version should converge much faster and thus have less communication cost. By contrast, FedAC[1] could achieve a faster convergence without the assumption of overparameterization, so it strikes me that the convergence analysis for accelerated FedAvg is quite weak. In original Nesterov full gradient descent, $\beta_t$ can be set as $\frac{\sqrt{L}-\sqrt{\mu}}{\sqrt{L}+\sqrt{\mu}}$ (a constant) but here it is set as $O(\frac{1}{t})$ (which decays, see Lemma 7). Is it the reason for slow convergence?
4. Since Theorem 3 and 4 have the same error bounds as Theorem 1 and 2, it is better to recite the bounds rather than restate them for a quicker understanding. Besides, the overparameterized convergence result is only mentioned in the ‘contribution’ part, while in the main body, it is just a passing. I think the author should talk more about FedMaSS and less about Nesterov-accelerated FedAvg considering the latter algorithm is poorly guaranteed.

[1] Yuan, Honglin, and Tengyu Ma. "Federated Accelerated Stochastic Gradient Descent." arXiv preprint arXiv:2006.08950 (2020).
[2] Woodworth, Blake, et al. "Is Local SGD Better than Minibatch SGD?." arXiv preprint arXiv:2002.07839 (2020).
[3] Woodworth, Blake, Kumar Kshitij Patel, and Nathan Srebro. "Minibatch vs Local SGD for Heterogeneous Distributed Learning." arXiv preprint arXiv:2006.04735 (2020).

---

> ### Author Response · Authors · 2020-11-13
> **Thank you for the constructive feedback!**
>
> Thank you very much for your valuable and constructive feedback! Our point-to-point responses are provided below.
>
> **Response to point 1. Difference between Lemma 3 and Lemma 6.**
> While Lemma 3 and Lemma 6 have a similar form, the crucial difference between the two is that the bound in Lemma 3, under strong convexity, contains a contraction factor of $(1-\mu \alpha_t)$ in front of $E\|\bar w_t-w^*\|^2$, which is not present in the bound in Lemma 6 for convex problems. This contraction factor is what allows one to obtain an $\mathcal{O}(1/KT)$ rate for strongly convex problems, compared to the $\mathcal{O}(1/\sqrt{KT})$ rate for convex problems. The term $\alpha_{t}(F({\bar w}_{t})-F(w^{\ast}))$ actually also appears in the bound in Lemma 3, but is bounded in a different manner, which constitutes the main difference between the one-step progress bounds for the strongly convex and convex cases. Thus even though they look similar, their proofs are different and we have highlighted the differences with more emphasis in the paper, e.g. on page 19.
>
>
> **Response to point 2. Comparison with [1] and [2].**
> Thank you for pointing out our omission of these excellent references, and we have incorporated them in the updated version. We verified that even though the algorithm they study is also referred to as FedAvg, the setting in the two works do not allow either heterogeneous data or partial participation, in contrast to the FL setting considered in this paper. Thus even in the full participation case, our results are not directly comparable as we assume that the data that is available to each client is different, which departs from the classical distributed optimization setting. We have included comparison with these works in the updated version, but since Table 2 only includes papers that study data heterogeneous settings, we do not put these two works there, although can definitely change that if the reviewer thinks it is necessary.
>
> **Response to point 3 and 4. Non-acceleration of stochastic Nesterov FedAvg.**
> Indeed as the reviewer points out, full Nesterov-accelerated gradient descent with constant step size provably accelerates over vanilla GD. However, this breaks down with stochastic methods, and Nesterov SGD does not accelerate over SGD even in the non-FL setting. See e.g. [1,2,5]. Thus the best we can hope for in the FL setting for FedAvg with stochastic Nesterov updates is the same convergence rate with linear speedup as FedAvg with SGD, and [4] proves that this is the case for non-convex problems in terms of convergence to stationary points. No results existed in convex and strongly convex settings, so our results fill this gap.
>
> While optimal theoretical results for Nesterov FedAvg are limited to the same rates as FedAvg with vanilla SGD, in practice FedAvg with Nesterov is observed to perform better empirically. In fact, many previous works such as [3] on FedAvg with vanilla SGD uses Nesterov or other momentum versions in their experiments to achieve target accuracy. We have incorporated this observation in the section on Nesterov FedAvg to better motivate Theorems 3 and 4.
>
> We agree that the algorithm in [5] achieves the same linear speedup rates with a better communication complexity for general convex and strongly convex problems, but their setting does not allow either data heterogeneity or partial participation. Although their proposed accelerated algorithm is interesting from both theoretical and practical perspectives, Nesterov and other momentum-based algorithms are more commonly used in practice in both non-FL and FL settings. Therefore, we believe that our results on the convergence rates of Neseterov FedAvg provide theoretical understanding of the linear speedup behavior in the full FL setting and complement the result in [4] for non-convex problems to complete the picture.
>
> Given this discussion, we believe that our optimal linear speedup results on Nesterov FedAvg are relevant due to the popularity of momentum methods in practice and the lack of theoretical understanding in convex settings. The overparameterized results are to illustrate the geometric convergence and to provide a provable acceleration of the geometric rate. Because of the relevance of the Nesterov results and the fact that their analyses can be unified with those of Theorems 1 and 2 to provide insights on the linear speedup, we decided to emphasize these and delegated the overparameterized results to the appendix.
>
> In the updated version, we have emphasized that the bounds are the same for Nesterov and vanilla SGD, but can simplify it further. In particular, by ''reciting'' the bound do you mean e.g. referring to Theorem 1 in Theorem 3 directly?

---

> > ### Author Response · Authors · 2020-11-13
> > **References**
> >
> > [1]Rahul Kidambi, Praneeth Netrapalli, Prateek Jain, and Sham Kakade. On the insufficiency of existing momentum schemes for stochastic optimization.In2018 Information Theory and Applications Workshop (ITA), pages 1–9.IEEE, 2018.
> > [2]Chaoyue Liu and Mikhail Belkin. Accelerating sgd with momentum for over-parameterized learning.ICLR, 2020.
> > [3]Sebastian U Stich. Local sgd converges fast and communicates little.ICLR,2019.
> > [4]Hao Yu, Rong Jin, and Sen Yang. On the linear speedup analysis of communication efficient momentum sgd for distributed non-convex optimization.ICML, 2019.
> > [5]Honglin Yuan and Tengyu Ma. Federated accelerated stochastic gradient descent. Advances in Neural Information Processing Systems, 33, 2020.

---

### Official Review · AnonReviewer2 · 2020-10-28
**A theoretical study on federated averaging for convex and strongly convex loss**

**Rating:** 5
**Confidence:** 4

**Review:**

The paper provided convergence analyses for federated averaging and a momentum-based variant for convex and strongly convex problems, with a focus on the effect of the number of participating devices.

Pros:
This paper fills a gap in the theory literature. While the convergence of federated averaging is already studied for convex/strongly convex/nonconvex problems in literature, the effect of client sampling is usually not taken into consideration. This paper considered the effect of unbiased client sampling in the convergence analysis.

Cons:
1. The results that a larger number of devices can accelerate training is quite expected and standard in distributed optimization literature. There are no new insights from the paper except for the choice of E implied by Theorem 2. The insight is that E can be O(T^{1/4}/N^{3/4}) when all devices are participating in every iteration, however, E should be O(1) if client sampling is used. This an important insight and it is rather counter-intuitive. This is not discussed in depth in the paper and it is not sure whether it is an artifact of the analysis or such a phenomenon exists in practice. Possible improvements could be discussing how the requirement of E being O(1) arises in the analysis and looking for problem instances where E being O(1) is required.

2. Another concern is this paper seems to be incremental, the analysis for federated averaging with client sampling should not be difficult given the existing theoretical frameworks for strongly convex/convex/nonconvex problems. If the analysis is non-trivial, I encourage the authors to discuss the difficult parts in the main paper.

---

> ### Author Response · Authors · 2020-11-13
> **Thank you for the constructive feedback!**
>
> Thank you very much for your valuable and constructive feedback! Our point-to-point responses are provided below.
>
>
> **1. Choice of $E$ in partial participation.**
>
> The choice of $E$ in the partial participation case is tight due to the sampling variance of our sampling schemes. We have incorporated this observation on page 6 of the main text in the discussion under communication complexity, where we remark that the requirement $E=\mathcal{O}(1)$ cannot be removed for our particular sampling schemes, as the dependence on $E$ of the sampling variance $\mathcal{O}(E^2/T^2)$ is tight, and also refer to Proposition 1 in the Appendix, where we provide a problem instance that demonstrates the tightness. We can definitely place further emphasis on this point in the next round of updates if the reviewer thinks that is necessary.
>
> **2. Significance of results in full FL setting.**
>
> We agree that there has been a line of works in distributed optimization that demonstrate the linear speedup phenomenon in simpler settings without heterogeneous data or partial participation, and we have discussed and compared with these works in the introduction and after our main theorems. However, what distinguishes the full FL setting from these distributed optimization settings is the presence of both types of heterogeneities, and prior works were not able to demonstrate the linear speedup phenomenon with arbitrary number of participating devices in the full FL setting, so our work is the first to do so. Moreover, we provide a unified analysis of the results for convex and strongly convex problems, for both SGD-based and momentum-based FedAvg, and highlight the common elements and differences in their analysis, which also contributes to the understanding of linear speedup behavior of these algorithms under both data heterogeneity and system heterogeneity.

---

> > ### Comment · AnonReviewer2 · 2020-11-15
> > **Where does the  term in theorem 2 come from? My question was on Theorem 2 not on Theorem 1.**
> >
> > Thank you for the quick reply. I agree with the authors that the bound on the variance term is tight, however, the term that prohibits using larger E than O(1) in Theorem 2 is the term $E^2G^2/\sqrt{KT}$. Where does this term come from, is this term also variance? In my original review, my question about E is on Theorem 2 but the authors' answer focuses on Theorem 1. My concern on E is still not addressed.

---

> > > ### Author Response · Authors · 2020-11-15
> > > **Updated**
> > >
> > > Thanks for your response! The $E^2 G^2/\sqrt{KT}$ term in the bound in Theorem 2 indeed also comes from the sampling variance, in a way that is similar to the term $\frac{\kappa E^2 G^2 \mu}{KT}$ in Theorem 1. We apologize that we did not directly address the concern about $E^2 G^2/\sqrt{KT}$, and have incorporated it in the discussion following Theorem 2, as well as the updated proof for Theorem 2, where we spell out how the sampling variance $\mathbb{E}\|\mathbf{\overline{w}}_t-\mathbf{\overline{v}}_t\|^2$ becomes $E^2 G^2/\sqrt{KT}$ in the final bound. Essentially, Proposition 1 demonstrates that the sampling variance $\mathbb{E}\|\mathbf{\overline{w}}_t-\mathbf{\overline{v}}_t\|^2=\mathcal{O}(\alpha_t^2 E^2)$ cannot be made independent of $E$, which is true independent of whether we are in the strongly convex or convex case, since the sampling scheme does not depend on the convexity in any way. Due to this dependence on $E$, the extra term $E^2 G^2/\sqrt{KT}$ in the bound in Theorem 2 in the partial participation case cannot be made independent of $E$. Since it is the leading term, this restricts $E=\mathcal{O}(1)$ for partial participation.

---

### Official Review · AnonReviewer1 · 2020-10-29
**Concern about the dependence on E**

**Rating:** 6
**Confidence:** 3

**Review:**

##########################################################################

Summary:

This work studies federated learning (FL) and analyzes FedAvg, the most widely used FL algorithm. The contribution is mainly theoretical. This paper contributes better convergers rates than prior work, arguably.


##########################################################################

Reasons for score:

This is a borderline paper. I am fine with either acceptance and rejection. While the new theories are somehow interesting, my main concern is that the convergence rates are not better than the standard SGD. I elaborate on this point in below.



##########################################################################

Pros:

This paper has theoretical contributions. This paper contributes new theories for FedAvg, which is a very popular algorithm, under various settings.


##########################################################################

Cons:

The point of FedAvg is that setting $E$ (number of local updates) greater than one makes the convergence faster. This is why FedAvg is more useful than distributed SGD. However, the bounds in this paper are not very reasonable: as $E$ increases, the convergence gets slower. If it is the case, then why not using distributed SGD? I am not sure if I missed anything. Please address my concern.



##########################################################################

After reading the authors' feedback, I decided to increase my rating from 5 to 6. The authors convinced me that setting $E>1$ can reduce the number of communications.

---

> ### Author Response · Authors · 2020-11-13
> **Thank you for the constructive feedback!**
>
> Thank you very much for your valuable and constructive feedback! Our point-to-point responses are provided below.
>
> **1. Dependence on $E$ and comparison to distributed SGD.**
>
> In our notation, $E$ is the number of local updates between consecutive communication rounds, $T$ is the total number of iterations or equivalently the total number of SGD updates, and $T/E$ is the total number of communication rounds.  The fact that an increased $E$ can lead to worse *iteration complexity* holds true universally in both distributed optimization and federated learning settings, as the decrease in communication frequency among local clients results in slower convergence to the global optimizer. The benefit of increasing $E$ is to reduce the communication cost, as it decreases the number of communication rounds $T/E$ required to achieve an iteration complexity, which can then translate to faster *wall clock time* when communication is costly. For example, our results imply that the number of communication rounds $T/E$ can be as small as $\mathcal{O}(\sqrt{NT})$ in the full participation case without degrading the $\mathcal{O}(1/NT)$ linear speedup rate, whereas distributed SGD needs $T$ communications (since it averages at every iteration) to achieve the same $\mathcal{O}(1/NT)$ convergence rate. So it is in this sense that our algorithm is faster compared to distributed SGD.
>
> We apologize that this point was not made clear in the text, and have clarified it further in the main text of the updated version, e.g. in the discussions under communication complexity on page 6, as well as Table 2 in the appendix, where we provide the largest possible number of local updates $E$ that guarantees the linear speedup convergence.

---

### Decision · Program_Chairs · 2021-01-07
**Final Decision**

**Decision:**

Reject

**Comment:**

This work provides theoretical analysis for FedAvg, contributing better convergence rates than prior work. Moreover, the paper shows that setting E > 1 can reduce the number of communications.  The contribution is incremental.